# Cloud-scale modelling of the impact of deep convection on the fate of oceanic bromoform in the troposphere: a case study over the west coast of Borneo

Paul. D. Hamer[1,2], Virginie Marécal[2], Ryan Hossaini[3], Michel Pirre[4,*], Gisèle Krysztofiak[4], Franziska Ziska[5], Andreas Engel[6], Stephan Sala[6], Timo Keber[6], Harald Bönisch[6], Elliot Atlas[7], Kirsten Krüger[8], Martyn Chipperfield[9], Valery Catoire[4], Azizan A. Samah[10,11], Marecel Dorf[12], Phang Siew Moi[11], Hans Schlager[13], Klaus Pfeilsticker[14]

[1]Norwegian Institute for Air Research (NILU), Kjeller, Norway
[2]Centre National de Recherches Météorologiques, Université de Toulouse, Météo-France, CNRS, Toulouse, France
[3]Lancaster Environment Centre, Lancaster University, Lancaster, LA1 4YQ, UK
[4]Laboratoire de Physique et Chimie de l'Environnement et de l'Espace, CNRS and University of Orléans, UMR7328, Orléans, France
[5]GEOMAR Helmholtz Centre for Ocean Research Kiel, Kiel, Germany
[6]Institute for Atmospheric and Environmental Sciences, University of Frankfurt, Altenhöferallee 1, 60438 Frankfurt, Germany
[7]University of Miami, 4600 Rickenbacker Causeway, Miami, FL 33149
[8]Department of Geosciences, University of Oslo, Postboks 1022, Blindern, 0315 OSLO
[9]The Institute for Climate & Atmospheric, Science, School of Earth and Environment, University of Leeds, Leeds, UK
[10]National Antarctic Research Centre, University of Malaya, Kuala Lumpur 50603, Malaysia
[11]Institute of Ocean & Earth Sciences, University of Malaya, 50603 Kuala Lumpur, Malaysia
[12]Max Planck Institute for Chemistry, Department of Atmospheric Chemistry, Mainz, Germany
[13]Deutsches Zentrum für Luft- und Raumfahrt (DLR), Institut für Physik der Atmosphäre, Atmosphärische Spurenstoffe, Münchner Straße 20, 82234 Oberpfaffenhofen-Wessling, Germany
[14]Institute of Environmental Physics, Ruprecht-Karls-Universität Heidelberg, Im Neuenheimer Feld 229, D-69120 Heidelberg, Germany
*Retired

*Correspondence to*: Paul D. Hamer (paul.hamer@nilu.no)

**Abstract.**

This paper presents a modelling study on the fate of $CHBr_3$ and its product gases in the troposphere within the context of tropical deep convection. A cloud-scale case study was conducted along the west coast of Borneo where several deep convective systems triggered on the afternoon and early evening of November 19[th] 2011. These systems were sampled by the Falcon aircraft during the field campaign of the SHIVA project and analysed using a simulation with the cloud-resolving meteorological model C-CATT-BRAMS at $2 \times 2$ km resolution that represents the emissions, transport by large scale flow, convection, photochemistry, and washout of $CHBr_3$ and its product gases (PGs). We find that simulated $CHBr_3$ mixing ratios and the observed values in the boundary layer and the outflow of the convective systems agree. However, the model underestimates the background $CHBr_3$ mixing ratios in the upper troposphere, which suggests a missing source at the regional scale. An analysis of the simulated chemical speciation of bromine within and around each simulated convective

system during the mature convective stage reveals that >85% of the bromine derived from $CHBr_3$ and its PGs is transported vertically to the point of convective detrainment in the form of $CHBr_3$ and that the remaining small fraction is in the form of organic PGs, principally insoluble brominated carbonyls produced from the photo-oxidation of $CHBr_3$. The model simulates that within the boundary layer and free troposphere, the inorganic PGs are only present in soluble forms, i.e., HBr, HOBr, and $BrONO_2$, and consequently, within the convective clouds, the inorganic PGs are almost entirely removed by wet scavenging. We find that HBr is the most abundant PG in background lower tropospheric air and that this prevalence of HBr is a result of the relatively low background tropospheric ozone levels at the regional scale. Contrary to a previous study in a different environment, for the conditions in the simulation, the insoluble $Br_2$ species is hardly formed within the convective systems and therefore plays no significant role in the vertical transport of bromine. This likely results from the relatively small quantities of simulated inorganic bromine involved, the presence of HBr in large excess compared to HOBr and BrO, and the relatively efficient removal of soluble compounds within the convective column.

## 1 Introduction

Organic brominated compounds cause stratospheric ozone loss (Engel and Rigby, 2018). A compilation of model and observational evidence shows that both longer lived (e.g., methyl bromide ($CH_3Br$)) organic bromine compounds and so-called very short-lived species (VSLS) are required to explain the ranges of total $Br_y$ within the stratosphere (strat-$Br_y$) of $5\pm2$ pptv (Engel and Rigby, 2018). Recent observation campaigns (Andrews et al., 2016; Navarro et al., 2015; Wales et al., 2018) show some minor variations but broadly agree with this compiled range.

Recent studies using global chemistry transport models (CTM) and chemistry climate models (CCMs) estimate the VSLS contribution to strat-$Br_y$ to range from 2 - 8 pptv (Liang et al., 2010; Hossaini et al., 2012; Hossaini et al., 2016; Aschmann and Sinnhuber, 2013; Liang et al., 2014, Wales et al. 2018; Tegtmeier et al., 2020) which is broadly consistent with the ranges compiled in Carpenter et al. (2014) and Engel and Rigby (2018). The model estimates differ due to the considered VSLS and which assumptions are made for surface emissions, chemistry, and washout in the troposphere.

Brominated VSLS are primarily of biogenic and oceanic origin produced by macroalgae (Leedham et al., 2012) and phytoplankton. Observations indicate that VSLS emissions are larger towards coasts compared to the open ocean (e.g., Quack and Wallace, 2003, Carpenter et al., 2009). Bromoform ($CHBr_3$), with 3 Br atoms per molecule, has the largest emissions among the different brominated VSLS (Engel and Rigby, 2018). For these reasons we focus on $CHBr_3$ in the present study.

Global estimates of $CHBr_3$ emissions range between 120 and 820 Gg/yr (Liang et al., 2010; Warwick et al., 2006; Butler et al. 2007, Ordoñez et al., 2012; Pyle et al., 2011; Ziska et al., 2013, Engel and Rigby, 2018). Most current inventories show that emissions are predominantly distributed in the tropics, but there is considerable uncertainty regarding the precise spatial and temporal distribution of emissions at the regional (Ashfold et al., 2014 Fiehn et al 2017, 2018) and the global scale (Hossaini et al. 2013). Furthermore, recent work suggests that the extratropical zones may also be important source regions

(Keber et al., 2020).

Tropical deep convection is the primary mechanism by which emissions of short-lived tropospheric trace gases and aerosols are transported to the upper troposphere. If convective outflow detrains above the level of zero radiative heating (LZRH), it undergoes net radiative heating and eventual buoyancy-driven slow ascent to the stratosphere. For clear sky conditions, the LZRH is approximately 15 km in the tropics, and can be as low as 11 km for air masses within clouds resulting from

convective outflow (Corti et al., 2005, 2006). Tropical deep convection can also loft air masses directly into the stratosphere through a process called convective over-shooting, but this process is less frequent (e.g., Liu and Zipser 2005, Luo et al. 2008).

After emission, $CHBr_3$ undergoes oxidation in the troposphere during transport either via reaction with the hydroxyl radical (OH) or via photolysis (approximate lifetime of 16 days, Burkholder et al., 2018). The oxidation products are organic and

inorganic product gases (PGs) (Hossaini et al., 2010; Krysztofiak et al., 2012). The most important organic PGs are the brominated organic peroxides, $CBr_3O_2H$, $CHBr_2O_2H$; and the brominated carbonyl species, $CBr_2O$ and $CHBrO$. The inorganic PGs consist of the bromine radical (Br), molecular bromine ($Br_2$), bromine oxide (BrO), hypobromous acid (HOBr), hydrogen bromine (HBr), and bromine nitrate ($BrONO_2$). These PGs have a range of solubilities, and thus washout within convective systems is expected to exert a strong control on the vertical transport of bromine to the upper troposphere

(Hossaini et al., 2010).

Since $CHBr_3$ transport by deep convection occurs at the local scale and involves complex chemistry, the analysis of the detailed $CHBr_3$ and PG processes occurring within deep convection and in its vicinity requires fine scale modelling at the kilometer-resolution with detailed chemistry (e.g., Barth et al., 2001; Marécal et al., 2006). These processes are the convective-scale transport and mixing, the full bromoform degradation scheme in the gaseous phase, the speciation of the

resulting PGs into organic and inorganic forms, the partitioning of PGs across the gas-aqueous phases due to their solubilities and interactions with hydrometeors during formation, mature and decaying convective stages, hydrolysis of $BrONO_2$ within cloud and rain droplets, and aqueous phase chemistry of dissolved gases in cloud and rain droplets. This knowledge, gained from studies at the convective scale, may then improve the representation of the fate of chemical species in global models. Because of their coarse resolution, current state-of-the-art global 3-dimensional models use sub-grid scale

parameterizations of deep convection and are not able to resolve all convective events. These parameterizations are a known source of uncertainty for tracer transport, including $CHBr_3$, from the boundary layer to the upper troposphere (e.g., Hoyle et al. 2011, Liang et al., 2014; Hossaini et al., 2016; Butler et al., 2018). Also, because global models need to compromise between complexity and computing resources, they include simplifications of $CHBr_3$ chemical processes and their interactions with hydrometeors. For instance, in Hossaini et al. (2012), $CHBr_3$ degradation is assumed to release $Br_y$

immediately. The Liang et al. (2014) study is based on a stratospheric model and uses a OH climatology in the troposphere. Aschmann and Sinnhuber (2013) represent in their stratospheric model the partitioning between inorganic species and HBr uptake on ice but no organic products and no explicit tropospheric chemistry is included. Only two detailed model studies examining VSLS degradation chemistry (both gas and aqueous phase), which were idealised cases, have been carried out at

the convective scale (Krysztofiak et al., 2012; Marécal et al., 2012). Krysztofiak et al. (2012) focused on developing and optimising a photochemical mechanism for $CHBr_3$ degradation for use within models, and they estimated Henry's Law coefficients for some of the organic bromine species included within the optimised mechanism. Marécal et al. (2012) implemented the $CHBr_3$ photochemical scheme of Hossaini et al. (2010) in addition to aqueous phase uptake and chemistry based on the Henry's Law coefficients from Krysztofiak et al. (2012). Using idealised simulations of a tropical convective cloud, Marécal et al. (2012) explored the $CHBr_3$ chemistry at the cloud scale and highlighted the importance of aqueous phase processes for understanding source gas and PG chemistry and transport.

The previous studies of VSLS chemistry and transport at the convective scale were only idealised cases that used a set of simplifying assumptions (e.g., no emissions, constant vertical profiles for initial conditions, and no synoptic scale meteorological forcing) and artificial perturbations to the modelled atmosphere to induce their simulated convection. Thus, these cases were not realistic and it would not have been relevant to compare them to observations. We wish to expand upon that previous work, e.g., Marécal et al. (2012), by carrying out a real-world case study. In this paper we present a study based on high-resolution cloud resolving modelling of the transport, chemistry, and washout of $CHBr_3$ and its derivatives within convective clouds along the west coast of Borneo. Specifically, we aim to look at the bromine compound speciation in the areas of convective entrainment and detrainment and both within and outside of several convective systems. The modelling is supported using aircraft observations of $CHBr_3$, which we use to establish the credibility of the simulated chemical processes. The case study corresponds to tropical deep convection reaching the upper troposphere that is far more common than overshooting convection. Because part of the air masses within clouds resulting from convective outflow clouds in the tropical upper troposphere reach the stratosphere by radiative ascent, this paper is relevant for global stratospheric studies, in particular with global models that are only representing this type of pathway and not overshooting convection.

Section 2 gives a description of the measurement campaign over the west coast of Borneo, it provides an overview of the case study, and it includes a description of the meteorological situation based on observations. Section 3 describes the model used, its new developments, and the simulation setup. Section 4 presents our modelling results, which includes: i) an evaluation of the simulated meteorology with respect to the situation described in Sect. 2; ii) a detailed analysis of the transport, chemistry and washout of bromoform and its PGs; and iii) a discussion of the chemistry. Section 5 discusses the limitations, Sect. 6 presents a summary and our conclusions.

## 2 The SHIVA Campaign and Case Study Overview

The EU funded SHIVA ('Stratospheric Ozone: Halogen Impacts in a Varying Atmosphere') project (http://shiva.iup.uni-heidelberg.de/) was designed to address uncertainties in our understanding of VSLS, their contribution to stratospheric bromine, and their impact on stratospheric ozone. A measurement campaign within SHIVA was carried out in November 2011 that focused on the South East Asia Maritime Continent (SEA-MC) to better understand the emissions and the transport of oceanic VSLS, including $CHBr_3$, to the upper troposphere and stratosphere. This region was selected for two reasons.

First, it represents globally the most important region for deep convection (Liu and Zipser, 2005) and second, the SEA-MC was believed to be an important region for VSLS emissions due to its many coastlines and its location in the tropics. The campaign primarily relied on measurements of chemical species onboard the Deutsches Zentrum für Luft- und Raumfahrt (DLR) Falcon aircraft based in Miri, Sarawak, Malaysia, which were complemented by ship and ground-based observations (Pfeilsticker et al., 2013). By sampling convective outflows under the influence of high $CHBr_3$ coastal emission zones, several of the SHIVA flights were particularly well designed to document the impact of deep convection on the bromoform distribution in the upper troposphere. We selected the case study from the SHIVA campaign on the afternoon of 19 November 2011. Figure 1 shows hourly maps of brightness temperature contours measured by the 11 µm channel IR108 on board the MTSAT-2 satellite. Note that the brightness temperatures contours in Fig. 1 are chosen to highlight only cloud tops in the upper troposphere. The brightness temperature imagery illustrates that several deep convective systems initiated inland along the West coast of Borneo where $CHBr_3$ emissions are expected to be strong (Ziska et al., 2013). The Falcon aircraft sampled two of these convective systems, which we henceforth refer to as Obs_Conv1 and Obs_Conv2 and are shown in the blue box and pink box in Fig. 1, respectively.

The temporal evolution of both convective systems develops over several hours (05 UTC to 12 UTC) and is shown in Figs 1 (a) through (h), which indicates that the two follow a similar development scenario. They were both initiated early afternoon inland close to the west coast of Borneo from offshore low-level winds encountering the steep topography of the island. These low levels winds come from the north/north-westerly large-scale flow as indicated by ECMWF analysis (not shown), possibly combined with local diurnal variations of the sea breeze (Johnson and Priegnitz, 1981). The initial convective cells developed vertically and then horizontally to form an anvil from its outflow (also named stratiform part of the convective system) in the upper troposphere driven off the coast by the easterly/south-easterly upper tropospheric flow.

Obs_Conv1 was already well developed at 05 UTC (13h local time: 13 LT) and was located around 5.5°N and 116°E at this time. Obs_Conv1 produced a large anvil on its west flank that started weakening after 10 UTC (18 LT). The anvil of Obs_Conv1 was well sampled by the Falcon aircraft during its mature stage at altitudes between 11 and 13 km from 8.05 UTC to 9.35 UTC (16.05 LT to 17.35 LT). The trajectory of the aircraft is plotted in Fig. 1 in the 09 UTC panel and shows the intersection of the Falcon and the anvil cloud. The other convective cell, Obs_Conv2, initiated at 06 UTC (14 LT) and was located at about 4.3°N and 114.4°E. It later produced an anvil of convective outflow that developed and moved north-westward similarly to Obs_Conv1. It lasted several hours and started to decay from 10 UTC (18 LT).

The Falcon flight on the afternoon of November 19, 2011, was aimed at sampling the outflow of the Obs_Conv1 system. Krysztofiak et al. (2018) identified from humidity data and webcam images the times when the aircraft flew within the convective outflow (i.e., in cloudy conditions) and when it was in cloud-free conditions. This information has been used here to show in Fig. 1 (09 UTC panel) where the flight sampled cloud-free air or cloudy air. The Falcon aircraft sampled the Obs_Conv1 system multiple times but also flew within the Obs_Conv2 system at ~12.5 km altitude around 9.20 UTC (visible at 5.2° N and 114.8° E) and in cloud-free conditions below Obs_Conv2 on its way back to Miri around 9.50 UTC at an altitude of ~6 km.

Since CHBr$_3$ emissions and its marine boundary layer (BL) mixing ratios are large close to Borneo's west coast (Ziska et al., 2013; Fuhlbrügge et al., 2016), CHBr$_3$ was transported from the BL aloft by the Obs_Conv_4.35N and Obs_Conv_3.75N systems as confirmed by observations of elevated CHBr$_3$ mixing ratios relative to the background conditions during the flight (Sala et al., 2014; Krysztofiak et al., 2018).

## 3 Model Simulations

### 3.1 Model Description

    We use the Chemistry-Coupled Aerosol and Tracer Transport model to the Brazilian developments on the Regional Atmospheric Modeling System (C-CATT-BRAMS) (Longo et al. 2013), which is a version of the CATT-BRAMS model (Freitas et al., 2009) coupled on-line with a chemistry model. This system is capable of resolving meteorological processes and the resultant tracer transport and chemistry. C-CATT-BRAMS has its original heritage in the Regional Atmospheric

Modeling System version 6 (RAMS) (Walko et al., 2000). RAMS is a fully compressible non-hydrostatic model consistent with Tripoli and Cotton (1982). RAMS can run in a nested grid configuration and includes various physical parameterizations to simulate sub-grid scale meteorological processes for turbulence, shallow cumulus convection, deep convection, surface-air exchanges, cloud microphysics, and radiation. Note that for km-scale simulations, convection is resolved explicitly and thus subgrid-scale convective parameterizations are not needed. BRAMS builds upon RAMS with the

inclusion of several modifications that serve to improve the model performance within the tropics. For example, BRAMS includes an ensemble implementation of the deep and shallow cumulus convection schemes, a soil moisture initialisation using model prognostication combined with a remote sensing rainfall product, and more realistic surface characteristics for vegetation type derived from the MODIS (Moderate-resolution Imaging Spectroradiometer) NDVI (Normalized Difference Vegetation Index) product (Freitas et al., 2009).

The model represents microphysical processes using the single-moment bulk parameterization (Walko et al., 1995) whereby rain, cloud, pristine ice, snow, aggregates, graupel and hail are considered. The radiation scheme used in the model calculates the effects of clouds, hydrometeors, and aerosols upon radiation (Toon et al., 1989). The model considers turbulent mixing using the turbulent kinetic energy (mean kinetic energy per unit mass for eddies in turbulent flow) as a prognostic variable (Mellor and Yamada, 1982).

The chemistry scheme used in C-CATT-BRAMS simulates gas and aqueous phase chemistry, photochemistry, uptake described by Henry's Law, and hydrolysis. Marécal et al. (2012) and Krysztofiak et al. (2012) provide a detailed overview of the equations describing the chemistry solved by the model. To summarise, the model calculates chemical loss and production rates, and it computes chemical species concentrations in the gas phase, in cloud particles, and in rain droplets. To this end, the chemistry scheme couples with the microphysical scheme which explicitly resolves cloud and precipitation

processes (Marécal et al., 2012). In practice, the model considers, within the bulk microphysical scheme, the effects on the chemical species of condensation, evaporation, water vapour deposition, and sedimentation. In addition, the reversible

exchange of gases between the gas and aqueous phases (cloud and liquid hydrometeors) is estimated using Henry's Law and accommodation constants. Once within the condensed phase, the model includes the transfer of chemical species from within cloud particles to the different types of hydrometeors during coalescence and riming, and within the individual types of hydrometeor. In brief, it assumes Henry's law with cloud water and uses the production of precipitation to determine how much of the soluble trace gas is removed by wet deposition. We use retention coefficients to describe the proportion of a chemical compound that is retained in the condensed phase upon the transition from one type of hydrometeor to another. This approach represents liquid-to-ice processes like riming that are the dominant process for the formation of ice hydrometeors in convective clouds. We simplify the treatment of retention for the formation of ice precipitates by assuming a retention coefficient of 1 (i.e., the entirety of the compound) for all chemical species dissolved in liquid precipitate that undergo freezing. This is a frequently used assumption within washout schemes in global and regional scale chemical models. The uptake of bromine species onto ice hydrometeors is not represented as it was found in Marécal et al. (2012) to not have an important effect on bromine removal.

The photolysis rates are computed on-line in the model using the Fast-TUV (Tropospheric Ultraviolet and Visible) radiative model (Tie et al., 2003). This is done in such a way as to consider the effects of clouds on photolysis rate in an interactive way.

The $BrONO_2$ hydrolysis reaction within cloud particles and rain droplets has been added to the chemical scheme. Its mathematical implementation is described by Marécal et al. (2012). The reaction scheme considers the reaction within cloud particles and rain droplets separately using the mean mass radius and mean mixing ratios for cloud particles and rain droplets from the bulk micro-physical scheme, the thermal velocity, the gas phase diffusivity, and the accommodation coefficient of $BrONO_2$.

Note that bromine-chlorine reactions have not been included in the chemistry scheme since Marécal et al. (2012) showed that it has only a small impact on the production of $Br_x$ even under the most favourable conditions possible.

## 3.2 New Model Developments

Several important changes have been applied to the model to simulate chemical and physical processes associated with $CHBr_3$ degradation chemistry and transport. A photochemical mechanism for the degradation of $CHBr_3$ was developed, tested, and optimised for use in C-CATT-BRAMS (Krysztofiak et al., 2012). The development of the new mechanism also included the estimation of the most favoured branching ratios for the halogenated peroxy reactions with the hydro-peroxy radical ($XRO_2 + HO_2$) (see Table 1 of Krysztofiak et al., 2012 for more details) using ab initio calculations of the standard reaction enthalpies. Reaction rates were either estimated from the analogous chlorine compounds or via a generalised expression. In addition, to properly simulate the uptake and washout of PGs into cloud particles and rain droplets, Henry's Law coefficients had to be estimated using predictive methods: the bond contribution method (Meylan and Howard, 1991) and the molecular connectivity index (Nirmalaklandan and Speece, 1988) for the brominated organic peroxides, $CBr_3O_2H$,

CHBr$_2$O$_2$H, and the brominated carbonyl species, CBr$_2$O and CHBrO. Krysztofiak et al. (2012) discusses the validity of these estimates. The Henry's Law constants of bromine species are shown in Table 1. Note that the information on BrONO$_2$ is not included in Table 1 since it undergoes rapid hydrolysis in water and thus its removal is uptake limited (see Marécal et al. (2012) for details).

**Table 1: Values used in the determination of the Henry's Law constants $H_X$[1] and effective Henry's constants $H'_X$[2] of bromine species X used in the model.**

| Species | $H_{298}$ (mol l$^{-1}$ atm$^{-1}$) | $a_H$ (K) | $K_{A\,298}$ (mol l$^{-1}$) | $H_X$ [1] at cloud base (20 °C) | $H'_X$ [2] at cloud base (20 °C) and cloud/rain pH of 5 |
|---|---|---|---|---|---|
| HBr | 0.71[5] | 10200[5] | $1 \times 10^9$ [4] | 1.27 | $1.27 \times 10^{14}$ |
| HOBr | $6.1 \times 10^3$ [3] | 5900[4] | 0 | $8.55 \times 10^3$ | - |
| Br$_2$ | 0.76[5] | 4177[5] | 0 | 0.97 | - |
| Br | 1.7[3] | 5200[3] | 0 | 2.29 | - |
| CBr$_3$OOH | $1.96 \times 10^5$ [6] | 5200[7] | 0 | $2.63 \times 10^5$ | - |
| CHBr$_2$OOH | $2.25 \times 10^4$ [6] | 5200[7] | 0 | $3.03 \times 10^4$ | - |
| CHBrO | 74[6] | 5800[8] | 0 | $1.02 \times 10^2$ | - |
| CBr$_2$O | 21.5[6] | 5600[8] | 0 | $2.96 \times 10^1$ | - |
| CHBr$_3$ | $3.4 \times 10^{-2}$ [3] | 1800[3] | 0 | $3.77 \times 10^{-2}$ | - |

$^1 H_X = H_{298} exp\left(a_H \left(\frac{1}{T} - \frac{1}{298}\right)\right)$

$^2 H'_X = H_X \times (1 + \frac{K_{A\,298}}{[H^+]})$

[3]Sander (1999); [4]As for HOCl (Sander, 1999); [5]Yang et al. (2005); [6]Krysztofiak et al. (2012); [7]As for CH$_3$OOH (Sander, 1999); [8]Mean temperature dependency of RCHO and RR'CO (Sander, 1999)

These new developments coupled with a simple tropospheric chemistry scheme including carbon monoxide (CO), methane (CH$_4$), ozone (O$_3$), oxidised nitrogen (NO$_y$), and hydrogen oxide radicals (HO$_x$) (Barth et al., 2007) were successfully
implemented by Marécal et al. (2012). Building on the new mechanism implemented in Marécal et al. (2012), non-methane hydrocarbon (NMHC) chemistry was added to the chemical mechanism in order to provide a more realistic description of the chemistry for the SHIVA real case study. The NMHC mechanism is a reduced version of the Regional Atmospheric Chemistry Mechanism (Stockwell et al., 1997) called the Regional Lumped Atmospheric Chemical Scheme (ReLACS, Crassier et al., 2000). We term this modified version of the ReLACS scheme RELASH. RELASH includes 60 chemical
species, 166 chemical reactions, treats NMHCs with up to 8 carbon atoms via lumping scheme, and is designed to describe tropospheric chemistry only. The full chemical mechanism is described in S1 of the supplement.

### 3.3 Model Configuration

The model simulation was run for 3 days from 12:00 UTC November 17 to 12:00 UTC November 20 2011. We used a nested grid configuration with three grids. The coarsest and largest grid covers from 90°E to 135°E and from 14°S to 23°N and uses a spatial resolution of $50 \times 50$ km; the next coarsest grid covers from 106°E to 123°E and from 2°S to 12°N at a resolution of $10 \times 10$ km; and the finest scale grid covers from 112.7°E to 117.4°E and from 3.3°N to 7.6°N and has a spatial resolution of $2 \times 2$ km. This horizontal spatial configuration allows the finest grid to completely include the two convective systems (Obs_Conv1 and Obs_Conv2) and the region covered by Falcon flight on the afternoon of November 19[th]. The finer domain and its associated model orography are plotted in Fig. 2. This illustrates well the abrupt topography on the west side of Borneo Island leading to the development of deep convection in sea-breeze conditions. The model has 53 vertical levels with varying vertical separation using finer resolution within the BL. The top of the model reaches to 26.6 km. The model meteorology was initialised and forced along the coarse grid boundaries of C-CATT-BRAMS using 6 hourly European Centre for Medium-Range Weather Forecasts (ECMWF) analysis fields for vector wind components, temperature, geopotential height, and specific humidity. We used the ECMWF operational analysis at $0.5° \times 0.5°$ resolution.

Within the coarsest two model grids we enabled the parameterisations for shallow cumulus convection and for deep convection. We used the deep convection parameterisation from Grell and Dévényi (2002) as implemented in CATT-BRAMS in (Freitas et al., 2009). We allowed the model to resolve clouds and convective processes directly for the finest resolution grid. The topography used in the model has a 10 km resolution within the coarsest two grids and a 1 km resolution within the finest grid. Sea surface temperatures (SSTs) were initialised using the satellite observed weekly average SSTs (Reynolds et al., 2002).

To describe $CHBr_3$ emissions, we have implemented the emission inventory of Ziska et al. (2013). This is a bottom-up inventory based on the atmospheric and oceanic measurements of the HalOcAt (Halocarbons in the Ocean and Atmosphere) database project (https://halocat.geomar.de/). Using SHIVA flight measurements and the TOMCAT CTM, Hossaini et al. (2013) showed that this inventory performs best for bromoform in the Maritime Continent region compared to the inventories of Liang et al. (2010), Warwick et al. (2006) updated by Pyle et al. (2011), and Ordóñez et al. (2012). The Ziska et al. (2013) emissions have a $1° \times 1°$ resolution. A diurnal variability linked to solar zenith angle is applied to these emissions such that they peak at solar noon. They are shown in Fig. 3 for the largest domain used in the C-CATT-BRAMS simulation. Note that the emissions are large on the west coast of Borneo Island where convection develops on the afternoon of November 19[th], 2011.

Various chemical species were initialised and forced along the coarse grid boundaries using 6 hourly output from the TOMCAT CTM (Chipperfield, 2006). The chemical species were: $CHBr_3$, $O_3$, hydrogen peroxide, nitrogen oxide, nitrogen dioxide ($NO_2$), nitric acid, pernitric acid, CO, methane, ethane, propane, isoprene, HCHO, methaldehyde, ethaldehyde, acetone, peroxy actyl nitrate, peroxy propyl nitrate, methyl hydroperoxide, ethyl hydroperoxide, $Br_2$, BrO, HOBr, HBr, and $BrONO_2$, the bromo carbonyls $Br_2C(=O)$ and $HBrC(=O)$, and bromo peroxides ($CHBr_2OOH$ and $CBr_3COOH$). The

TOMCAT simulation was run using the Ziska et al. (2013) emissions to ensure the consistency between the C-CATT-BRAMS simulation and its chemical boundary conditions from TOMCAT. For some of these species we had to perform lumping, splitting, and scaling by reactivity in order to achieve consistency with the chemical mechanism used in C-CATT-BRAMS.

## 4 Results

Consistent with the objectives of this paper, the results shown and discussed in this section are only those of the finest resolution grid (2 ×2 km, see Fig. 2) since it gives a detailed description of the meteorology and chemical composition within and in the vicinity of deep convection developing over the west coast of Borneo.

### 4.1 Meteorology

Simulations with limited area models with horizontal resolutions of the order of 1 km, as in the present study, are largely
used to study in detail the development of convective systems since they provide an explicit representation of their dynamical and thermodynamic processes. At this resolution, there is no need to use sub-grid-scale parameterizations for convection. But even at this fine resolution, modelling tropical deep convective systems remains a challenge when one wants to reproduce the exact time, intensity and structure (extent of convective cloud component versus stratiform cloud component) compared to observations. This is particularly true in maritime conditions because of the uncertainties in the
representation of hydrometeor properties and processes and because of their sensitivity to large scale meteorological conditions (e.g., Varble et al. 2014 and references therein). We do not attempt to make a detailed comparison of the model simulations with the particular convective systems sampled by the Falcon (Obs_Conv1 and Obs_Conv2). Instead, we now evaluate if the observed, general features of the development of the deep convective systems (described in Sect. 2 and Fig. 1) are well captured by the model.

To show the evolution of the modelled convective systems, we plot in Fig. 4 the model-derived brightness temperatures at 11 mm (IR108 channel wavelength) estimated from cloud top pressures from 05 UTC to 12 UTC on November 19[th], 2011 using RTTOV v12.3 (Radiative Transfer of TOVS, https://nwpsaf.eu/site/software/rttov/, last access: 23 June 2021, Saunders et al. 2018). Figure 4 shows that the model simulates three deep convective systems that develop during the afternoon. These systems (called hereafter Mod_Conv_4.35N, Mod_Conv_3.75N and Mod_Conv_5.4N) can be identified and followed in
time by coloured rectangles. Consistent with the observations, the analysis of the simulated meteorological fields show that all three systems are triggered inland close to the coast in the early afternoon from the large-scale low-level winds (north/north-westerlies) enhanced by local sea-breeze that encounters the fairly steep topography of west Borneo (Fig. 2). This process is illustrated in Fig. 5a, which shows the simulated low-level wind direction and intensity at 06 UTC (14 LT, a time when convection is at an early stage) and the associated temperature field that exhibits a sea-land positive gradient.
After 06 UTC the deep convective systems move offshore towards the west/north-west driven by the upper tropospheric

winds, and they develop an anvil from their outflow. Upper tropospheric winds at 09 UTC (17 LT) are shown in Fig. 5b. Compared to the evolution of the observed brightness temperatures (Fig. 1), the vertical extension of the convective part of the systems during the mature stage tends to decrease a bit too rapidly in the model (Fig. 4) leading to a less extended anvil, likely due to a too rapid removal of precipitation related to uncertainties in the microphysical parameters. However, the

values of brightness temperatures in the convective systems, that are a proxy of the cloud top height, are similar in the model and the observations showing that the model provides a good estimate of the height of the convective systems that developed on the west coast of Borneo on the studied day.

**Table 2: Characteristics of the observed and simulated deep convective systems. The cloud top heights for the observed convective systems are based on Hamada and Nishi (2010) and Iwasaki et al. (2010). The outflow refers to the stratiform part of the convective system.**

| | Location where the convective system first reaches ~14.5 km altitude | Time when the convective system first reaches ~14.5 km altitude | Time when the outflow starts to dissipate | Estimated maximum top altitude of the outflow |
|---|---|---|---|---|
| Obs_Conv1 | 5.5°N-116.0°E | ~5 UTC (~13 LT) | After 10 UTC (18 LT) | 14.5 ±0.5 km at 9UTC (17 LT) 15.5 ±0.5 km at 10UTC (18 LT) |
| Obs_Conv2 | 4.3°N-114.4°E | ~7 UTC (~15 LT) | After 11 UTC (19 LT) | 15.5 ±0.5 km at 10UTC (18 LT) 14.5 ±0.5 km at 11UTC (19 LT) |
| Mod_Conv_4.35N | 4.3°N-114.2°E | ~5 UTC (~13 LT) | At least 8 UTC (16 LT) (out of the domain) | ~15.5 km at 7UTC (15 LT) ~15.5 km at 8UTC (16 LT) |
| Mod_Conv_3.75N | 3.75°N-113.9°E | ~8 UTC (~16 LT) | At least 11 UTC (19 LT) (out of the domain) | ~14.5km at 10UTC (18 LT) ~14.5km at 11UTC (19 LT) |
| Mod_Conv_5.4N | 5.4 °N-115.8 °E | ~8 UTC (~16 LT) | After 11 UTC (19 LT) | ~14.5km at 10UTC (18 LT) ~14 km at 11UTC (19 LT) |

Other quantitative characteristics of the observed (Obs_Conv1, Obs_Conv2) and modelled (Mod_Conv_4.35N,

Mod_Conv_3.75N, Mod_Conv_5.4N) convective systems are compared in Table 2. For Obs_Conv1 and Obs_Conv2, we use the estimates of the observed cloud top heights derived from brightness temperatures (Hamada and Nishi, 2010; Iwasaki et al., 2010) only where the uncertainty is ~0.5 km or lower. For the model, we use a set of cross sections from the 3D-fields to estimate the model cloud tops. Table 2 shows a general agreement on altitudes between the observations and the model. Regarding the timing, the two observed convective systems originating on the west coast of Borneo do not reach the upper

troposphere at the same time (5UTC for Obs_Conv1 and 7 UTC for Obs_Conv2) and the duration before they start to

dissipate also varies (5h for Obs_Conv1 and 4h for Obs_Conv2). In the model, the three convective systems also show variations of these two parameters which are close to those observed. The time when convection first reaches the upper troposphere in the model is only off by 1h maximum (08 UTC) compared to observations (07 UTC) and still occurs in the afternoon. Regarding the dissipation time, there is an uncertainty because two of the systems leave the model domain. However, the model simulates anvils that last at least 3h before they start to dissipate.

Overall, from collating the information from Fig.1, Fig. 4, and Table 2, we find that the simulation represents the general characteristics of the observed convective systems well, in particular:

- the origin of their development from the interaction of the large-scale flow and the steep orography of the west coast of Borneo combined to local effects,
- the location of the initial convective cell about 30 km inland on the west coast of Borneo,
- the development of an anvil (the stratiform part) from the convective outflow during the afternoon moves off-shore,
- the cloud top height of the outflow,
- the transport of the convective systems north-westwards and westwards,
- the duration of the system of several hours and decay during early evening.

The main discrepancy is that the condensed water in the simulated convective part of the systems tends to precipitate a bit too efficiently compared to observations. Nevertheless, in its early stages, the convective part of the system, as evidenced by condensed water, reaches altitudes greater than 14.5 km. Achieving this altitude gives a strong indication that the intensity of the main updraft transporting bromoform into the upper troposphere is predicted well by the model.

In conclusion, the model is able to simulate the general meteorology of the observed convective systems, at least within the constraints and uncertainties of km-scale modelling of convection (Varble et al. 2014).

The three convective systems we examine detrain into altitudes ranging between 11 and 15 km. According to Corti et al. (2005) and Corti et al. (2006), the LZRH can be as low as 11 km for air masses within ice clouds due to the effects of their radiative properties. Ice clouds are present in the anvils of all three of the simulated systems, and they could cause a shift in the radiative balance and sufficient heating to lower the altitude of the LZRH. This would imply that the simulated air masses could gain positive buoyancy sufficient to reach the stratosphere over long enough time and large enough spatial scales. Thus, the study of the chemistry and washout within these systems could have relevance for the transport of $CHBr_3$ and its PGs to the stratosphere.

**4.2 Comparison of the Measured and Modelled Bromoform Statistics and Convective Transport Efficiency**

Before discussing the results of the simulated chemistry in detail (section 4.3), we evaluate if the simulation gives reasonable results for $CHBr_3$ concentrations and for convective transport efficiency compared to the aircraft observations.

We firstly use statistical characteristics for this comparison. We choose this approach for two reasons. First, because of differences in location and timing between the observed and simulated convection events, and, second, because of spatial

uncertainties in the emission inventory used in the simulation. This approach allows a clearer comparison of the observations and simulation by removing effects arising from inherent temporal and spatial uncertainties.

In order to compare the convective transport efficiency between the observed and simulated systems we follow the approach proposed by Cohan et al. (1999) and used by Bertram et al. (2007). To estimate the air fraction, $f$, originating from the boundary layer (BL) and transported by convection we use the relationship from Cohan et al. (1999):

$$[X]_{UTconv} = f \cdot [X]_{BL} + (1 - f) \cdot [X]_{UTnoconv} \qquad (1)$$

where the mean mixing ratios in the boundary layer, the upper troposphere within the convective systems, and the upper
troposphere in the vicinity but outside the convective systems are represented by $[X]_{BL}$, $[X]_{UTconv}$, and $[X]_{UTnoconv}$, respectively. $f$ ranges from 0 to 1 with large values corresponding to an efficient convective transport of air masses from the boundary layer to the upper troposphere. This formulation of $f$ is chosen because it was recently applied to the SHIVA aircraft data (Krysztofiak et al., 2018). It relies on the assumption of a low variability of background concentrations with altitude, which is fulfilled for $CHBr_3$ in our case study (not shown). Previous studies based on observations, and reported in
Krysztofiak et al (2018), provides estimates of $f$ in the range 0.17 to 0.36.

Table 3 shows the modelled and observed mixing ratios of $CHBr_3$ that are used to calculate the $f$ fraction. The mixing ratios are divided into three subsets corresponding to the boundary layer ($[X]_{BL}$), the upper troposphere within the convective systems ($[X]_{UTconv}$) and in the upper troposphere in the vicinity but outside the convective systems ($[X]_{UTnoconv}$). The details on how the estimates from the observations and from the model were determined are given in Supplement S2.

**Table 3: Estimates from the model simulations of the $CHBr_3$ mixing ratios (all in pptv) in the boundary layer $[X]_{BL}$, in the UT outside convection $[X]_{UTnoconv}$ and in the UT within convection $[X]_{UTconv}$. $f$ is the air fraction originating from the boundary layer and transported by convection. Details on the method used are given in Supplement S2. (\*) corresponds to the SHIVA estimate https://www.klm.no/enfrom 4 flights and boat data from different days including the flight on the afternoon of November 19, 2011, using measurements of different species ($CHBr_3$, CO, $CH_4$, and $CH_3I$) (Krysztofiak et al., 2018). The error listed for $f$ is calculated by propagating the standard deviation errors on each [X] term used to calculate it. The equations to explain the propagation of error are given in Supplement S3.**

| | $[X]_{BL}$ (mean ± 1σ) | $[X]_{UTnoconv}$ (mean ± 1σ) | $[X]_{UTconv}$ (mean ± 1σ) | fraction $f$ |
|---|---|---|---|---|
| Mod_Conv_4.35N | 2.11 ± 0.24 | 0.29 ± 0.07 | 0.62 ± 0.18 | 0.18 ± 0.11 |
| Mod_Conv_3.75N | 1.20 ± 0.25 | 0.33 ± 0.13 | 0.62 ± 0.13 | 0.33 ± 0.23 |
| Mod_Conv_5.4N | 1.58 ± 0.37 | 0.34 ± 0.11 | 0.56 ± 0.12 | 0.18 ± 0.14 |
| Obs_Conv1 and Obs_Conv2 from $CHBr_3$ SHIVA observations on the afternoon of November 19th, 2011 | 1.82 ± 0.86 | 0.51 ± 0.04 | 0.73 ± 0.12 | 0.17 ± 0.15 |
| Mean from observations of 4 SHIVA flights* | | | | 0.29 ± 0.25 |


Mod_Conv_4.35N and Mod_Conv_5.4N give values of the fraction of air transported by convection from the BL that are close to the estimates of $f$ based on CHBr$_3$ observations gathered on November 19$^{th}$, 2011.

A higher $f$ fraction is calculated for Mod_Conv_3.75N meaning that this system was more efficient for transport of CHBr$_3$ from the BL to the UT. However, this high fraction $f$ is consistent with the average value calculated from all SHIVA aircraft data (0.29 ± 0.25) determined by Krysztofiak et al. (2018) using carbon monoxide measurements from the SPIRIT instrument (Catoire et al. 2017) and the GHOST CHBr$_3$ measurements onboard the Falcon (Sala et al., 2014). The GHOST instrument is a gas-chromatograph mass spectrometer and had an error of ±17.7% that was primarily driven by uncertainties in the gas standard (Sala et al., 2014). Table 3 shows an overall good consistency (both in terms of magnitude and simulated variability) between the model results, the findings of Krysztofiak et al., (2018), and the SHIVA measurements concerning $f$. Furthermore, Fuhlbrügge et al. (2016) used a trajectory model and found similar values of $f$ at 10-13 km of between 30-40% for November 19$^{th}$ for the west coast Borneo region. It is also worth noting that Mod_Conv_3.75N has a higher uncertainty and agrees with the fraction f of Obs_Conv1 and Obs_Conv2 within the combined uncertainties.

We now examine the magnitude of the CHBr$_3$ mixing ratios in the BL and in the UT in both inside and outside of the convection using Table 3 and Fig. 6. The box and whisker plots in Fig. 6 giving the median (i.e., 50$^{th}$ percentile) and the 5$^{th}$, 25$^{th}$, 75$^{th}$ and 95$^{th}$ percentiles provide complementary statistical information to Table 3 on the variability of the observed and simulated CHBr$_3$ mixing ratios. Differences between the median and the mean is a measure of the skewness of the distribution of points. The 5$^{th}$, 25$^{th}$, 75$^{th}$ and 95$^{th}$ percentiles give additional information characterising the low and high values of the distribution of the bromoform mixing ratios.

In the BL, there is a large spread in the observations because of the very large local variability of the emissions that the model cannot capture due the resolution of the CHBr$_3$ emission inventory we used. Nevertheless, the median BL mixing ratios of all three lie within the 25$^{th}$ and 75$^{th}$ percentile of the observed BL mixing ratios, and the 5$^{th}$-95$^{th}$ percentile range of simulated mixing ratios across all three simulated systems lie within the 5$^{th}$-95$^{th}$ percentile range of observed mixing ratios in the BL. In the BL, the Mod_Conv_4.35N mean and median mixing ratios are higher compared (see Fig. 6 and Table 3) to the observations. Meanwhile, the mean and median mixing ratios in the BL below Mod_Conv_5.4N are a bit lower than that observed, and Mod_Conv_3.75N shows the lowest mean and median there (see Table 3 and Fig. 6). This is likely related to two combined factors: the emissions are weaker in the southern part of Borneo's west coast where Mod_Conv_3.75N takes place (see Fig. 3); and Mod_Conv_4.35N initiates in closer proximity to the coast compared to the other two systems where CHBr$_3$ emissions and BL mixing ratios are higher.

The mean and median UT mixing ratios in the simulation, which largely depend on the chemistry initial conditions from the TOMCAT simulation, are underestimated compared to observations, which leads to lower mixing ratios both within and out of the convection (differences of 0.17 to 0.22 pptv in UTnoconv, and of 0.11 to 0.17 pptv in UTconv). Hossaini et al. (2013) previously showed a comparable 0.08 pptv average negative bias in TOMCAT relative to the SHIVA aircraft measurements of CHBr$_3$ throughout the entire duration of the flight on the afternoon of November 19$^{th}$. TOMCAT's negative bias is

slightly smaller than in our case because our sampling focuses on a smaller spatio-temporal domain (between 8.20 UTC and 9.40 UTC) in proximity to the convective system whereas the TOMCAT negative biases were larger than the 0.08 pptv average for the full flight. The negative bias of the UT mixing ratios is probably linked to an underestimate in the emissions somewhere to the East of Borneo where these background UT air masses originate from in the TOMCAT simulation. This finding is consistent with Fuhlbrügge et al. (2016) who showed that local sources alone cannot account for the observed $CHBr_3$ levels in the UT. Furthermore, Keber et al. (2020) indicate that underestimates in the background tropical UT might arise due to underestimates in extratropical $CHBr_3$ sources.

In the background UT ($UT_{noconv}$), we see small differences in the mean and median mixing ratios in the vicinity of the three simulated convective systems with slightly higher values and variability (see all percentiles in Fig. 6 and $1\sigma$ in Table 3) for Mod_Conv_3.75N and Mod_Conv_5.4N because they developed in locations previously affected by convective transport of bromoform.

The mean and median mixing ratios for all three systems are close with slightly higher values (see Fig. 6 and Table 3) for Mod_Conv_4.35N due to higher BL concentrations and for Mod_Conv_3.75N due to its more efficient transport (highest $f$ in Table 3). Mod_conv_4.35N shows the highest variability (see all percentiles in Fig. 6 and $1\sigma$ in Table 3) within convection ($UT_{conv}$) and the lowest outside ($UT_{noconv}$) indicating that there is less mixing during the detrainment in the UT in this system.

Note that the results presented in this section have little sensitivity to the threshold in ice concentration used to define the sampling of grid points within the convection and outside in its vicinity (see Supplement S4).

As a complement to the statistical comparison, Fig. 7 shows a spatial comparison. We choose Mod_Conv_5.4 for this comparison because it is the closest in time/space to Obs_Conv1. For the flight observations, we only select those gathered in the upper troposphere at ~12-13 km altitude range which corresponds to the sampling of Obs_Conv1. We plot the modelled bromoform mixing ratio at 12.5 km altitude. To account for the shift in time between the model and the observations, we plot the model fields at 10 UTC when the anvil is well developed over the ocean as in the observations around 9 UTC. The observations are also slightly shifted in space in order to match the Mod_Conv_5.4 location. In Fig. 7a we show the model bromoform mixing ratio and in Fig. 7b the same field but adding 0.17 pptv that is the model bias in both UTconv and UTnoconv for Mod_Conv_5.4 with respect to the measurements. Figure 7a illustrates well the model bias linked to the initial conditions. By removing this bias in Fig. 7b, we find a good consistency between the model and the observations. This confirms the findings of the statistical analysis.

In summary, this evaluation shows that the simulation provides reasonable results compared to observations for the transport efficiency, and for $CHBr_3$ concentrations knowing that the UT background values are underestimated in the initial conditions.

**4.3 Cross section analyses of the simulated chemical processes**

We now briefly explain the underlying methodology used to interpret the model results (in the following sections 4.3.1, 4.3.2, and 4.3.3) and to conclude whether the convective transport leads to an enhancement or deficit in the mixing ratios for

CHBr$_3$ and its PGs within the UT. The enhancements or deficits in the CHBr$_3$ and PG mixing ratios in the convective column and UT that we report are based on comparisons of the simulated mixing ratios in each region of the atmosphere involved in the convective system, i.e., from BL, to vertical component of convective system, and to outflow. We use the

simulated mass mixing ratios of condensed water as a metric to define what is within and what is outside of the convective systems and outflow. Mass mixing ratios of 0.5 g kg$^{-1}$ and 0.01 g kg$^{-1}$ are used to define the most intense and outer limits of the convection systems, respectively.

### 4.3.1 Bromoform

In this section, we analyse the cross sections of the simulated chemical fields of CHBr$_3$ within the convective systems. As an

illustration of how CHBr$_3$ evolves during the convection, Fig. 8 shows a vertical cross section of the 5-hour time evolution of CHBr$_3$ mixing ratios in the central part of the Mod_Conv_5.4N system. The cross section is taken as close to the centreline of the convective system as possible. We selected Mod_Conv_5.4N since it corresponds most closely in space to Obs_Conv1. We can see the complete evolution of the convective system from a situation with elevated CHBr$_3$ concentrations in the boundary layer close to the location of the convection (of up to 2.1 pptv) at the very early stage of the

system (07 UTC, Fig. 8(a)). Then the convective column ascends in a relatively vertical fashion (08 UTC, Fig. 8 (b)) and afterwards develops an anvil on its west side (from 09 UTC, Figs 8 (c), (d) and (e)). The concentrations in the anvil are naturally at their highest at the time and location of convective detrainment and reach up to 0.9 pptv at 9 UTC (17 LT) and begin to decrease after one hour (up to 0.75 pptv) as the anvil is advected north westward by the high-altitude winds.

The analysis of the transport, chemical processes, and Br-atom speciation done hereafter is based on vertical cross sections

chosen in the central part of each convective system at the time when the anvil is in its mature stage; the precipitation and vertical transport within the convective column are also near their maximum at this stage. This means that these cross sections are representative of the most intense convective activity which demonstrate the combined effects of intense vertical transport, washout, and development of the anvil. All of the numbers presented in the following sections correspond to those of the cross-sections presented in the figures. Also, to be able to compare the contribution of the different bromine species to

the total Br-atom mixing ratios all the figures hereafter are expressed as Br-atom mixing ratios (henceforth known as pptv Br).

We show the concentrations of CHBr$_3$ in the three convective systems in Figs. 9 (a), (b), and (c) and their percentage contribution to the total Br mixing ratio in Figs 10 (a), (b) and (c). Note that Fig. 9 (c) is identical to Fig. 8 (c) except scaled by the number of bromine atoms, i.e., a factor of 3. Each of the three simulated convective systems exhibit different CHBr$_3$

mixing ratios within their convective columns and within their outflow anvils. This variability is because they each entrained different boundary layer mixing ratios of CHBr$_3$, and they also detrained into UT regions with slightly differing CHBr$_3$ backgrounds and have different transport efficiencies (Table 3).

Despite these variations, we see a consistent result in the Br-atom speciation and mixing ratios of each convective system in Figs. 9 and 10. CHBr$_3$ is elevated above background levels in all of the atmospheric regions dynamically linked to the

boundary layer (surface to 600-800m height) on a timescale well below the lifetime of $CHBr_3$ (i.e., the boundary layer, convective columns, and convective outflow). We can see that the $CHBr_3$ in these air masses have only undergone limited photochemical ageing because $CHBr_3$ accounts for >85% of the total Br mixing ratio (Figs. 10 (a), (b), and (c)) in these atmospheric regions. However, we consistently see lower $CHBr_3$ contributions to the total Br mixing ratio in atmospheric regions above the boundary layer not affected either directly or indirectly by convection. These regions include the low

(from the top of the boundary layer to ~2 km height), mid (from ~2 km height to ~8 km) and upper-troposphere (from ~8 km height to ~13-14 km) where the model typically simulates 0.45-0.9 pptv Br of $CHBr_3$ accounting for 60-70% of the total Br mixing ratio in the absence of convection to any vertical level. We can see some evidence of elevated $CHBr_3$ in the upper troposphere related to the transport of outflow from distant convection, for instance in Fig 9 (c) at around 116.5° E longitude and at 11 to 12 km. Within these air masses the model simulates intermediate mixing ratios and Br-atom contributions

signifying air masses of intermediate $CHBr_3$ ageing and mixing. Note that the sharp changes in bromine mixing ratios that we see above 14 km in Figs. 9 and onwards are due to the vertical transition into the tropical tropopause layer, which is influenced by the stratosphere where we find 0-0.45 pptv Br of $CHBr_3$ accounting for 15-20 % of the total simulated Br mixing ratio there.

### 4.3.2 Inorganic and Organic PGs

Figs. 9 (d), (e), and (f) show that there are relatively low levels of inorganic bromine (Br, $Br_2$, BrO, HOBr, HBr, $BrONO_2$) concentrations in the boundary layer even in the areas not directly affected by convective precipitation with values typically in the range of 0 to 0.4 pptv Br, i.e., <5 % contribution to the total Br mixing ratio (Figs. 10 (d), (e), and (f)). The highest simulated inorganic bromine mixing ratios (0.3-0.4 pptv Br) in the boundary layer occur to the west of the Mod_Conv_5.4N system still only contribute <10 % to the total boundary layer pptv Br (see Fig. 10 (f)). This spatial variability in the

boundary layer inorganic bromine mixing ratios around each convective system arises due to differences in precipitation location and timing over the course of the simulation prior to November 19th, 2011. Precipitation events occurring in the two preceding days deplete the boundary layer of inorganic bromine due to washout (analysis not shown). In the boundary layer, organic PGs (CHBrO, $CBr_2O$, $CHBr_2OOH$, and $CBr_3COOH$) concentrations are up to 0.2 pptv Br (up to 10% contribution to total bromine but very locally) and are formed due to $CHBr_3$ photochemical loss (Figs. 9 (g), (h) and (i) and 10 (g), (h) and

(i)).

Air masses in the convective column itself and convective outflow are almost entirely depleted of inorganic bromine with mixing ratios of <0.1 pptv Br and with contributions to the total Br mixing ratio well below 5%. There, organic compounds are being driven from the low levels up to the upper troposphere in the main ascent and the outflow and show enhanced mixing ratios within the convective column compared to the free troposphere (Figs. 9 (g), (h), (i)). Still, organic PGs have a

contribution to the total bromine only up to ~4%.

In the free troposphere, inorganic and organic bromine concentrations are enhanced between 1 and 4 km to the west of each convective system (Figs. 9 (d), (e), (f), (g), (h), (i) and 10 (d), (e), (f), (g), (h), (i)). There, total inorganic (resp. organic) PGs

peak up to 1 pptv Br (resp. 0.2 pptv Br), which constitutes a portion up to 45% (resp. 5%) of the total Br mixing ratio. Among the three convective systems, Mod_Conv_5.4N exhibits the highest concentrations of the organic PGs.

Above 4 km altitude in convection-free areas, inorganic bromine is mainly in the 0.2-0.4 pptv Br range (15-35% contribution to total Br) that is higher than within convection (Figs. 9 (d), (e), (f) and 10 (d), (e), (f)). There, organic PGs have low concentrations (0.02-0.03 pptv Br) and contributes only to 1-3% (Figs. 9 (g), (h), (i) and 10 (g), (h), (i)).

### 4.3.3 The Impact of PG Solubility

Given that washout is an important process within the convective systems, the relative solubilities of each component we look at is relevant for explaining the concentration levels of the inorganic and organic PGs. Table 1 shows the Henry's Law constants for the PGs. In order of increasing solubility, we first list the inorganic bromine PGs: BrO, Br, $Br_2$, HOBr, HBr, $BrONO_2$; and then the organic PGs: CHBrO, $CBr_2O$, $CHBr_2OOH$, and $CBr_3COOH$. In discussions from this point on, we will classify the inorganic PGs into two groups: soluble inorganic that includes HOBr, HBr, and $BrONO_2$; and insoluble

inorganic, comprising Br, $Br_2$, and BrO. We also classify the bromo-carbonyls (CHBrO and $CBr_2O$) as insoluble organic and the bromo-methyl peroxides ($CHBr_2OOH$ and $CBr_3COOH$) as soluble organic PGs. Note, that except for $Br_2$, $CHBr_3$ is less soluble than its PGs. The soluble inorganic, insoluble inorganic, soluble organic and insoluble organic bromine compounds and their relative contributions are shown in Figs. 11 and 12.

The gas phase mixing ratios of soluble inorganic bromine species are depleted to near zero in the convective columns and in

the immediate area of detrainment for each system (Figs.11 (a) (b) and (c)). Similarly, soluble inorganic species make almost no contribution to the total gas phase Br within each convective system (Figs. 12 (a), (b) and (c)). This depletion of soluble inorganic bromine species occurs even though their boundary layer mixing ratios range between 0.1 and 0.5 pptv Br and the soluble inorganic species form the bulk of the inorganic bromine at all levels in the troposphere outside of convective systems. This strongly implies that the model simulates the near total removal of the soluble inorganic bromine species

within the convective columns. The insoluble inorganic species (Figs. 11 (d), (e), and (f) and 12 (d), (e), and (f)) only make a negligible contribution to the total bromine budget throughout the troposphere, only reaching peak mixing ratios of 0.1 pptv Br and 4% of the total bromine in areas of the lower troposphere not affected by convection. We also see no enhancement of the insoluble inorganic species (which include $Br_2$) within the convection column or in the fresh convective outflow above the levels seen in the rest of the troposphere. There is also only a negligible contribution of insoluble inorganic bromine in

the UT affected by previous convection as illustrated in Figs. 12 (e) and (f) (to the East of both Mod_Conv_3.75N and Mod_Conv_5.4N).

The contribution to the total Br mixing ratio from the soluble organic bromine PGs (Figs. 11 (g), (h) and (i) and 12 (g), (h) and (i)) is also negligible (at a maximum of ~1% in the low troposphere not affected by convection). The bulk of organic PG species are instead in the form of insoluble species (Figs. 11 and 12 (j), (k), and (l)). The enhancements of the organic PGs

we see in Figs. 9 (g), (h), and (i) within the convective system are due to the insoluble bromo-carbonyls, i.e., up to 0.08 pptv

compared to the background free troposphere 0.02 pptv (Figs. 11 (j), (k), and (l) and 12 (j), (k), and (l)). These compounds contribute a maximum of between 75-95% of the PG bromine total within each of the three the convective columns, which is based on figures (not shown) of the relative contribution of the insoluble bromo-carbonyls to the total PG mixing ratio. We see across all three convective systems that the relative contribution of the insoluble bromo-carbonyls is at a maximum at the point PG mixing ratios are at a minimum. This indicates that these compounds survive complete washout during vertical ascent but play a small role in the vertical transport of bromine within the convection systems.

In order to understand further the behaviour of inorganic bromine we need to examine its composition. Figures 13 and 14 show HBr and HOBr mixing ratios and their percentage contribution to the total inorganic bromine. Inorganic bromine is almost entirely composed of HBr, HOBr, and $BrONO_2$ within the background troposphere outside of the convective systems in our simulations. HBr dominates the inorganic Br-atom contribution in the regions of the UT outside of the convective systems (Figs. 13 (a) (b) (c)). HBr provides between 40-65% of the total Br mixing ratio in the regions of the low and mid-troposphere (1.5-4 km height) unaffected directly by precipitation washout (Fig. 14, (a), (b) and (c)) while HOBr and $BrONO_2$ represent small fractions (0-35% and 0-20%, respectively) of the remainder (Figs. 13 (d) (e) and (f), and 14 (d) (e) and (f), $BrONO_2$ not shown). The lower to mid-troposphere (1.5-4 km) is the only tropospheric region where HOBr and $BrONO_2$ have significant mixing ratios of up to 0.3 pptv and 0.25 pptv. Otherwise HBr dominates the total inorganic Br mixing ratio in the BL and UT. Note that the very high HOBr and HBr relative contributions of up to 100 % within the most active part of the convective systems shown in Figs. 14 (d) (e) (f) are not meaningful since the total inorganic bromine mixing ratios are negligible there.

### 4.3.4 Discussion

We now present a discussion of the most important processes that control the overall speciation of the PGs in the background lower troposphere unaffected by convection in our simulation. This is important because this speciation determines the starting mix of chemical compounds present in the surrounding environment prior to convection and then ultimately what is available for entrainment into the convective system. The speciation of PGs in the lower troposphere is relevant for determining their potential transport to the upper troposphere since they individually have different solubilities.

A key finding is that inorganic bromine dominates the PG budget within the background lower troposphere during this case study. However, during convective transport the inorganic PGs present in the low tropospheric background air are almost entirely removed by washout since HBr is by far its most prevalent component in the marine boundary layer and the next two most abundant inorganic PGs (HOBr and $BrONO_2$) are also highly soluble. The regional tropospheric composition present in our simulations is the underlying cause of this prevalence of HBr and this in turn causes the efficient washout of inorganic PGs.

Table 4 shows an example built from mixing ratios at a point in the marine boundary layer at 200 m altitude in proximity to Mod_Conv_5.4N and gives an indication of the relative reaction rates for these conditions. Mixing ratios at this location are selected because the marine boundary layer air masses are those that are entrained within the convective systems and are

therefore most relevant. Note that the relative rates give an indicator of the preferred reaction for a particular species, e.g., for bromine atoms between R1, R2, and R3 shown below. However, the calculated rates cannot be used to quantitatively compare the production and loss of a compound, e.g., production and loss of BrO, because they only give an instantaneous approximate estimate of the rates for single reactions within the complex chemical system solved by the numerical solver.

**Table 4: Rate constants and reaction rates at a point in the marine boundary layer (200 m, 988.7 mb and 298.7 K) within an air mass being advected into the convective updraft of Mod_Conv_5.4N (5.8 N 115.5 E). This was done based on chemical species' mixing ratios at this location for all of the reactions (R1-R9) discussed in Sect. 4.3.4. This was in a region with no cloud or rain. For R8 and R10, the reactions taking place in the aqueous phase, we select some representative rain and cloud water mass mixing ratios (0.5 g kg-1 and 0.1 g kg-1, respectively) to demonstrate the reaction rates in the presence of liquid water condensate. In the case of R10 the hypothetical reaction rate is calculated using the aqueous concentration of HOBr at Henry's Law equilibrium, and due to the extremely high solubility of HBr it is assumed that all of its gas phase mixing ratio is dissolved in solution. For both R8 and R10 the reaction rates in cloud and rain droplets are shown separately, but in the case of R10 they cannot be combined additively.**

| Reaction | Mixing ratio of 1st reactant | Mixing ratio of 2nd reactant | Rate constant | Reaction rate / molecules cm$^{-3}$ s$^{-1}$ |
|---|---|---|---|---|
| R1: $Br + O_3$ | 0.08 ppqv | 13.1 ppbv | $1.17 \times 10^{-12}$ molecules$^{-1}$ cm$^3$ s$^{-1}$ | 678 |
| R2: $Br + HCHO$ | 0.08 ppqv | 3.7 ppbv | $1.17 \times 10^{-12}$ molecules$^{-1}$ cm$^3$ s$^{-1}$ | 191 |
| R3: $Br + HO_2$ | 0.08 ppqv | 37.2 pptv | $1.70 \times 10^{-12}$ molecules$^{-1}$ cm$^3$ s$^{-1}$ | 2.8 |
| R4: $BrO + HO_2$ | 0.5 ppqv | 32.5 pptv | $1.59 \times 10^{-11}$ molecules$^{-1}$ cm$^3$ s$^{-1}$ | 185 |
| R5: $BrO + NO_2$ | 0.5 ppqv | 4.6 pptv | $2.79 \times 10^{-12}$ molecules$^{-1}$ cm$^3$ s$^{-1}$ | 4.0 |
| R6: $BrO + NO$ | 0.5 ppqv | 0.6 pptv | $2.10 \times 10^{-11}$ molecules$^{-1}$ cm$^3$ s$^{-1}$ | 3.8 |
| R7: $BrONO_2$ | 0.8 ppqv | - | $9.45 \times 10^{-4}$ s$^{-1}$ | 18.4 |
| R8: $BrONO_2$ | 0.8 ppqv | - | $3.73 \times 10^{-3}$ s$^{-1}$ (rain: $1.33 \times 10^{-4}$, cloud: $3.60 \times 10^{-3}$) | 72.6 |
| R9: $HOBr + h\nu$ | 3.6 ppqv | - | $1.63 \times 10^{-3}$ s$^{-1}$ | 139 |
| R10: $HOBr + HBr$ | 3.6 ppqv | 0.23 pptv | $5.44 \times 10^{-6}$ molecules$^{-1}$ cm$^3$ s$^{-1}$ (rain)   $2.72 \times 10^{-5}$ molecules$^{-1}$ cm$^3$ s$^{-1}$ (cloud) | 0.22 (rain: 0.18, cloud: 0.04) |

Within the photochemical scheme of C-CATT-BRAMS, bromine atoms can react in the gas phase in one of three ways via R1, R2, or R3 (shown in order of decreasing reaction rate).

$Br + O_3 -> BrO + O_2$ (R1)

$Br + HCHO -> HBr$ (R2)

$$Br + HO_2 \rightarrow HBr \qquad\qquad (R3)$$

Bromine radicals can react with ozone via R1 to form BrO, which is normally the dominant reaction pathway for Br in the troposphere and this is the case here too. In addition, bromine radicals can react with HCHO or $HO_2$ to form HBr, but combined the rate of HBr formation via R2 and R3 is less than a third of the production rate of BrO. However, under relatively low ozone conditions (13.1 ppbv in this example) and in the presence of relatively high $HO_2$ (37.2 pptv) and only low $NO_x$ levels (e.g., 4.6 pptv of $NO_2$ and 0.6 pptv of NO), as we find in our simulations, the reactions between BrO and (in order of decreasing importance) $HO_2$ (R4), $NO_2$ (R5), and NO (R6) are enough to supress BrO mixing ratios to only negligible levels.

$$BrO + HO_2 \rightarrow HOBr \qquad\qquad (R4)$$

$$BrO + NO_2 \rightarrow BrONO_2 \qquad\qquad (R5)$$

$$BrO + NO -> Br + NO_2 \qquad\qquad (R6)$$

Combined R4, R5, and R6 suppress boundary layer BrO mixing ratios down to very low levels (0.5 ppqv) within this example airmass during the daytime. During the night BrO production will shut off, and residual NO as well as $NO_2$ react with BrO resulting in the latter's complete removal.

R4 leads directly to modest HOBr formation. R5 leads to $BrONO_2$ formation but its mixing ratios are also kept low by photolysis in combination with hydrolysis within cloud and rain droplets.

$$BrONO_2 + h\nu \rightarrow Br + NO_3 \qquad\qquad (R7)$$

$$BrONO_{2(aq)} \rightarrow HOBr_{(g)} + HNO_{3(g)} \qquad\qquad (R8)$$

The hydrolysis of aqueous phase $BrONO_2$ R8 is another pathway leading to HOBr formation, which is dependent on the presence of condensed moisture (note that for numerical reasons, in the model, the HOBr and $HNO_3$ from R8 are produced in the model in the gas phase as their partitioning into the aqueous phase is determined by Henry's Law and air-liquid diffusion limited uptake). However, despite modest formation rates of HOBr via R4 and R8 the mixing ratios of HOBr are supressed to relatively low levels during daytime by photolysis.

$$HOBr + h\nu \rightarrow Br + OH \qquad\qquad (R9)$$

Meanwhile, HBr (formed by R2 and R3, albeit relatively slowly) only has a very slow rate of photolysis and it is thus the single inorganic bromine compound with a lifetime long enough to allow accumulation in the gas phase under these conditions. HOBr and $BrONO_2$ are the next most abundant inorganic PGs, but their lifetimes (~10 minutes and ~20 minutes, respectively) are kept low during the daytime by photolysis. If background ozone levels were higher it would allow greater formation rates of BrO and in turn HOBr and $BrONO_2$ since BrO is the start point in the reaction pathways for both species. The maritime continent has been noted previously for having low $O_3$ levels throughout the depth of the troposphere during the winter monsoon resulting from westward transport of moist tropical air across the equatorial Pacific (Rex et al., 2014). The relatively low ozone levels are reproduced in our simulations (daytime values of as low as 10-15 ppbv at 2 km) and arise from the same mechanism across the equatorial Pacific represented in the TOMCAT global model forcing.

The low levels of BrO have another implication. Since BrO is directly involved in the two chemical pathways leading to HOBr formation (R4 and R8), the low levels of BrO can be implicated in the lack of significant quantities of HOBr (Figs. 13 and 14). HBr and HOBr react with one another in the aqueous phase to produce $Br_2$ via reaction R10 extremely rapidly.

$$HOBr_{(aq)} + Br^-_{(aq)} + H^+ \quad \rightarrow \quad Br_{2(aq)} + H_2O \quad\quad\quad\quad (R10)$$

Since this reaction has an effective stoichiometry of 1:1 ($Br^-$ and $H^+$ are both derived from HBr), and our model simulates that HBr is in a vast excess compared to HOBr in the gas phase in the troposphere, HOBr is the limiting reactant for $Br_2$ formation. In the example in Table 4, this excess is more than a factor of $10^3$ in the gas phase and, after accounting for the difference in solubility, a factor of $10^9$ in the aqueous phase. This leads to a relatively low reaction rate despite the speed of the reaction implied by the rate constant. At the levels shown in Table 4 this equates to a lifetime of HOBr in solution of

approximately $2 \times 10^{-2}$ s. This unbalance in the stoichiometry within the aqueous phase is simulated throughout the lower troposphere.

The atmospheric implication of R10 is that two soluble gases (HBr and HOBr) could potentially react rapidly within cloud and rain droplets to form the insoluble gas $Br_2$. R10 is known to lead to the production and release of significant quantities of $Br_2$ in other environments combining both the aqueous and gas phases, e.g., in the polar ice regions (McConnell et al., 1992)

and within volcanic plumes (Oppenheimer et al., 2006). In these examples the gas phase and aqueous phase chemistry combine to form a feedback loop leading to what is known as the "bromine explosion" (Wennberg, 1999). Thus, R10 could have the potential to significantly alter the Br-atom speciation within convective clouds in a short time in such a way as to reduce the PGs overall solubility; this would therefore promote the vertical transport of Br-atoms within convective systems even in the presence of abundant falling hydrometeors that would otherwise washout soluble gases like HBr very rapidly.

Figure 15 shows $Br_2$ mixing ratios in Br pptv. In our simulation, only negligible levels of $Br_2$ form during the most active phase of each of the simulated convective systems (Figs. 15 (a) (b) and (c)). At most we see up to 0.04 pptv of $Br_2$ within Mod_Conv_5.4N (Fig. 15c), and this is only within the lowermost sections of the system and these levels do not propagate vertically into the UT. These peaks in $Br_2$ formation coincide with spatially limited regions with elevated HOBr resulting from slightly higher background levels of BrO at these points.

From this analysis, we should expect more production of $Br_2$ in convective systems in environments with higher levels of background ozone, which would consequently have more HOBr. Indeed, there may be some observational support for this in data collected during the CONTRAST field campaign. Chen et al. (2016) showed that the observed levels of BrO and the sum of HOBr+$Br_2$ were both below the limit of detection (0.6-1.3 pptv and 1.5-3.5 pptv, respectively) in background tropical tropospheric air where ozone levels were relatively low (< 50 ppbv). They found that BrO and HOBr+$Br_2$ were only above

the limit of detection in biomass burning plumes where ozone levels were significantly elevated (> 50 ppbv). Their findings are at least consistent with our expectations of a link between ozone and the bromine speciation between HBr, BrO, HOBr, and $Br_2$.

The residence time of cloud and rain droplets in the atmosphere can impact aqueous phase chemistry in cloud and rain droplets. For other aqueous chemistry systems occurring in cloud and rain droplets the rate of chemical reactions could occur slower than it takes for cloud or rain droplets to fall leading to wet scavenging of the chemical species involved. Bela et al. (2018) report an example of aqueous phase chemistry in convective cloud and rain where the wet scavenging removal of $H_2O_2$ was found to be much faster than its production via aqueous phase chemistry in cloud and rain. By contrast, the rate constant for R10 is approximately $10^7$ times faster than the rate constants involved in the formation of $H_2O_2$ described by Bela et al. (2018), the lifetime of HOBr in solution is approximately $2 \times 10^{-2}$ s as a result. Thus, the residence time of cloud and rain droplets is not a limiting factor for R10.

### 4.3.5 Comparison with Marécal et al. (2012)

While other studies exist of convective scale modelling of ozone and aerosols (Crumeyrolle et al., 2008; Tulet et al., 2002), Marécal et al. (2012) is the only other study to have simulated the transport and photochemistry of bromoform at the convective scale and so we therefore compare our findings with this study. The C-CATT-BRAMS model configuration in Marécal et al. (2012) differs from the setup used in this study in the following ways.

- The model domain was set to be at the same latitude and longitude as Darwin (Australia).
- There were no emissions of $CHBr_3$ or any other VSLS. The only source of $CHBr_3$ was from a 2 km homogenous layer above the surface set as an initial mixing ratio of either 1.6 pptv or 40 pptv in the two scenarios.
- Marécal et al. (2012) used a chemical mechanism that did not include a representation of non-methane hydrocarbon chemistry. This chemistry was added in this study by including the ReLACS chemical mechanism.
- Apart from $CHBr_3$ set in the lowermost 2 km, the model initial and boundary conditions for the chemical species were defined from a single vertical profile from the MOCAGE CTM from over Darwin. The mixing ratios of all PGs were initialized at 0 pptv in all model grids and layers.
- The meteorology in Marécal et al. (2012) involved an initial setup that applied a single vertical profile of meteorological conditions throughout the entire horizontal domain. The vertical profile was defined from a radiosonde profile obtained above Darwin corresponding to November just prior to the main wet season. Convection was artificially forced in the simulation by introducing a perturbation in the lower model layers of increased temperature and humidity.

Marécal et al. (2012) showed in their analysis of the artificially induced tropical deep convective system that most of the Br transported to the UT by convection is in the form of $CHBr_3$, which is consistent with the results here. Nevertheless, in their case using an initial $CHBr_3$ of 1.6 pptv in the BL (which is the most comparable to this study), we can see notable differences in the composition of the convective outflow. Both organic and inorganic PGs are in higher abundance within the upper reaches of each convective system in this study compared to Marécal et al. (2012). In particular, the mixing ratios of all the inorganic PGs are greatly increased by over a factor of ten here compared to Marécal et al. (2012). As a result, Marécal et al. (2012) found a higher relative contribution to the total bromine PG mixing ratio in the UT for organic with

respect to inorganic PGs, i.e., 86% compared to 46% in this present study. The background tropospheric air contains elevated HBr levels in our simulation (Fig. 13 and 14) resulting from photochemical ageing of convectively lofted $CHBr_3$ represented both within the TOMCAT CTM initial and boundary conditions. It is likely that the major differences in simulated inorganic and organic PG mixing ratios within the convective system arise as a result of the idealized $CHBr_3$ and

PG background and initialization used within Marécal et al. (2012) compared to the more realistic simulation of the PG background (specifically HBr) in this study.

The small difference in latitude and atmospheric conditions between Darwin and Borneo is enough to cause longer $CHBr_3$ lifetimes in our simulation with respect to both photolysis (15.2 days compared to ~19 days here) and OH (22.6 days compared to 52.6 days here). Overall, both processes contribute to an increase in $CHBr_3$ lifetime from 9 to ~14 days here

(note both are fully consistent with the compiled ranges from Burkholder et al., 2018). The $CHBr_3$ lifetimes with respect to photolysis and OH are important for defining the relative partitioning of bromine between $CHBr_3$ and its different PGs prior to the convective activity on November 19[th]. The relative importance of photolysis has therefore increased in this study due to these changes, which could lead to an increase in the relative formation rates of inorganic bromine PGs compared to the organic PGs. This occurs because the photolysis of bromine cleaves off single bromine radicals that then contribute directly

to the inorganic PG budget. Furthermore, the remaining organic PGs contain one fewer bromine atoms as a result. Meanwhile, reaction with OH leads to the abstraction of a hydrogen atom, which creates organic PGs containing either two or three bromine atoms. Given the idealized initialization and experimental design within Marécal et al. (2012), it is not possible to properly assess these conjectures here, but it would be of interest to evaluate more formally differences in PG speciation arising in different photochemical environments that display differences in the $CHBr_3$ lifetime with respect to

reaction with OH and to photolysis.

Marécal et al. (2012) showed that significant amounts of $Br_2$ were only released into the gas phase from R8 in cloud droplets when their idealized simulation was initialized with very high $CHBr_3$ (40 pptv) in the BL. In the more realistic case where boundary layer $CHBr_3$ was ~1.6 pptv, the formation and release of $Br_2$ via R8 was very limited, which is consistent with our findings here as shown in Fig. 15. Following the causal link we identify between low ozone and resulting low HBr and

HOBr, these similar results for $Br_2$ could be explained by the low background $O_3$ simulated over the two regions (Borneo and Darwin) in both studies. In Marécal et al. (2012) the background $O_3$ was 14 ppbv at 2 km as compared to the 10-15 ppbv range simulated over the inner domain at 2 km altitude in this study.

**5 Limitations**

There are a few limitations to this study. First, it is difficult to generalise these results to other tropical areas because we

studied a region with a relatively low ozone background in the troposphere, which impacts the tropospheric bromine chemistry. Furthermore, other tropical regions could have vastly different $CHBr_3$ emissions, and in the case where emissions are much higher, as explored in Marécal et al., (2012), we could expect more $Br_2$ formation resulting in an increased role for

Br$_2$ in the transport of bromine to the upper troposphere. Pertinent to this point, Hossaini et al. (2016) highlight the Indian sub-continent and southeast Asia as another region that is potentially of importance for transport of bromoform and its PGs to the upper troposphere within deep convection. They predict it to make a more minor contribution to the vertical transport of bromoform compared to the maritime continent, but the background tropospheric conditions are very different from the present study due to its proximity to large pollutant and ozone precursor sources, and its contrasting conditions could make it an interesting case study.

Another limitation is that we neglect other VSLS in our simulations and only focus on CHBr$_3$, and as a result our findings are specific for the case where CHBr$_3$ is the only VSLS. Including other VSLS would alter the relative contributions made by CHBr$_3$ and its PGs to the vertical transport of bromine to the UT.

Another limitation is that our analysis is performed on cross sections of the most mature convective systems, and so our analyses offer a snapshot of the most intense convective activity and its effects.

Lastly, the performance of the various top-down and bottom-up CHBr$_3$ inventories varies significantly by region and the divergent global emission estimates represent a significant source of uncertainty in estimates of strat-Br$_y$$^{VSLS}$ (Hossaini et al., 2013). Despite the difference between observed and simulated CHBr$_3$ mixing ratios that we identify in the background UT we argue that the choice in using the Ziska et al. (2013) emission inventory was the correct one. Here, we cite Hossaini et al. (2013) that determined that the Ziska et al. (2013) emissions gave the closest agreement to observations when evaluated in the TOMCAT CTM over the same region of Borneo. Indeed, our results show consistent CHBr$_3$ mixing ratios compared to the simulations of Hossaini et al. (2013) for the November 19[th] flight, and together these findings provide tentative model-based evidence that the Ziska et al. (2013) emissions provide useful estimates of CHBr$_3$ emissions in this region. Furthermore, Ziska et al. (2013) project a CHBr$_3$ emission climatology ranging between 1200-1600 pmol m$^{-2}$ hr$^{-1}$ within the region of interest along the north west coast of Borneo while the emissions measured locally (over several days) by Fuhlbrügge et al. (2016) range from 300-4300 pmol m$^{-2}$ hr$^{-1}$. We conclude from this that Ziska et al. (2013) provides a good estimate of CHBr$_3$ emissions over this specific area.

## 6 Summary and Conclusions

We used a convective scale model ($2 \times 2$ km resolution) in order to gain a better understanding of the effects of tropical deep convection on the transport of CHBr$_3$ and the speciation of its PGs in the troposphere. Until now, two modelling studies have been carried out at the convective scale but only for idealised cases (Krysztofiak et al. 2012, Marécal et al., 2012). Our objective was to go a step further by modelling a real case study of deep convection that occurred along the west coast of Borneo on the afternoon/early night of 19 November, 2011, during the SHIVA field campaign.

It was shown that the meteorological development of the convective systems in the model have general characteristics similar to those observed. To further evaluate the simulation, we compared our modelled CHBr$_3$ mean concentrations and convection transport efficiency to those derived by Krysztofiak et al. (2018) from SHIVA measurements. The comparison

showed an underestimation of the CHBr$_3$ background concentrations within the upper troposphere related to underestimates in the Ziska et al (2013) emissions to the East of Borneo. These findings are consistent with those of Keber et al. (2020) that showed similar underestimates in the background tropical UT and with Fuhlbrugge et al. (2016) who showed that local sources alone cannot account for the observed CHBr$_3$ levels in the UT. Nevertheless, the fraction $f$ of air from the BL driven in the UT by the convective systems is consistent with Krysztofiak et al. (2018): 0.18±0.14 to 0.33±0.23 for the model and 0.17±0.15 to 0.29±0.25 for the observations.

Despite variation in the timing, location, and availability of CHBr$_3$ for entrainment in the BL below each convective system, the same general behaviour is observed across all three simulated convective systems. Most of the bromine (>85%) transported to the UT in each convective system is in the form of CHBr$_3$. Within the convective systems, the remaining 1-2% of the total bromine present is mostly in the form of organic PGs, i.e., the insoluble brominated carbonyls CHBrO and CBr$_2$O (86% as a contribution to the total PG). This is despite the inorganic PGs making a larger contribution than the organic PGs to the total Br mixing ratios in the free troposphere (i.e., 45% versus only 5%). Falling hydrometeors within the convective column efficiently remove the inorganic PGs, whose tropospheric budget is dominated by the extremely soluble HBr gas. Overall, we conclude that organic PGs are more important than inorganic PGs for the vertical transport of bromine within the convective columns for the conditions that we study here.

The insoluble inorganic PGs, BrO and Br$_2$, are only present at negligible mixing ratios and play no significant role in the vertical transport of bromine. Our interpretation is that the lower tropospheric inorganic PG budget is shifted heavily in favour of HBr formation due to the low background O$_3$ mixing ratios simulated in this region. This limits the availability of lower tropospheric HOBr leading to only very limited formation of Br$_2$ within the cloud and rain droplets within the lower regions of the convective system resulting from the reaction between HBr and HOBr. More BrO and HOBr would form in cases with higher background O$_3$, which could potentially lead to enhanced Br$_2$ formation within other convective systems and a more important role of the inorganic PGs for the vertical transport of bromine.

In the future, it would be of interest to evaluate findings from global CTM studies of VSLS transport to the UT and stratosphere in light of our findings based on modelling at the convection scale. This additional work, however, is beyond the scope of this study since it would require considerable additional technical work to reconcile the differences in spatial scale and conditions since global CTMs derived grid scale mixing ratios representing the mean both within and outside of convective systems. We hope to reconcile these issues and test the hypotheses raised in this paper as part of future work. This future effort could provide insights into the processes represented within CTMs.

*Author contributions.*

**PDH:** co-designed the study, wrote the main text of the paper, ran the model, analysed the model results, and created many of the figures.

**VM:** co-designed the study, helped develop this version of C-CATT-BRAMS, wrote some of the text, and created some of the figures.

**RH:** ran the TOMCAT simulations and helped edit the manuscript

**MP:** co-designed the chemical mechanism and the study, helped develop this version of C-CATT-BRAMS

**GK:** co-designed the chemical mechanism, helped develop this version of C-CATT-BRAMS, and edited the manuscript.

**FZ:** developed the emissions and provided us with the emission data.

**AE:** headed the GHOST-MS team onboard the Falcon aircraft and helped to edit the manuscript.

**SS:** a member of the GHOST-MS team onboard the Falcon aircraft.

**TK:** a member of the GHOST-MS team onboard the Falcon aircraft.

**HB:** a member of the GHOST-MS team onboard the Falcon aircraft.

**EA:** collected the CHBr3 observations on the Sonne boat cruise during SHIVA.

**MC:** helped run the TOMCAT CTM

**VC:** helped to run instrumentation onboard the Falcon, assisted in the SHIVA campaign planning, and edited the manuscript.

**AAS:** helped in the interpretation of the meteorological results.

**MD:** helped in the planning and implementation of the SHIVA aircraft campaign

**PSM:** helped enable the SHIVA campaign to take place in Malaysia.

**LKP:** helped enable the SHIVA campaign to take place in Malaysia.

**HS:** helped in the planning and implementation of the SHIVA aircraft campaign

**KP:** headed the planning and implementation of the SHIVA aircraft campaign

*Competing interests*. The authors declare that they have no conflict of interest.

*Acknowledgements.* This work was supported by the EU Stratospheric Ozone: Halogen Impacts in a Varying Atmosphere (SHIVA) project (SHIVA-226224-FP7-ENV-2008-1). We are grateful to the support given by the National Oceanography Directorate (NOD) and the Economic Planning Unit (EPU) of Malaysia. In particular, we are grateful for the support given 805 by Prof. Dr. Nor Aieni Binti Haji Mokhtar (NOD) and Mrs. Munirah Abd Manan (EPU) without which the SHIVA campaign in the Western Pacific would not have been possible. We would like to acknowledge the use of computing hours on the FUXI high performance computer at the Laboratoire d'Aérologie, Toulouse, France. We thank B. Quack for substantial comments and advice on the manuscript preparation. CATT-BRAMS is a free software provided by CPTEC/INPE and distributed under the CC-GNU-GPL license. RH is supported by a NERC Independent Research 810 Fellowship (NE/N014375/1). EA was supported by funds from NASA Upper Atmosphere Program (Grant NNX17AE43G). For CJH.

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

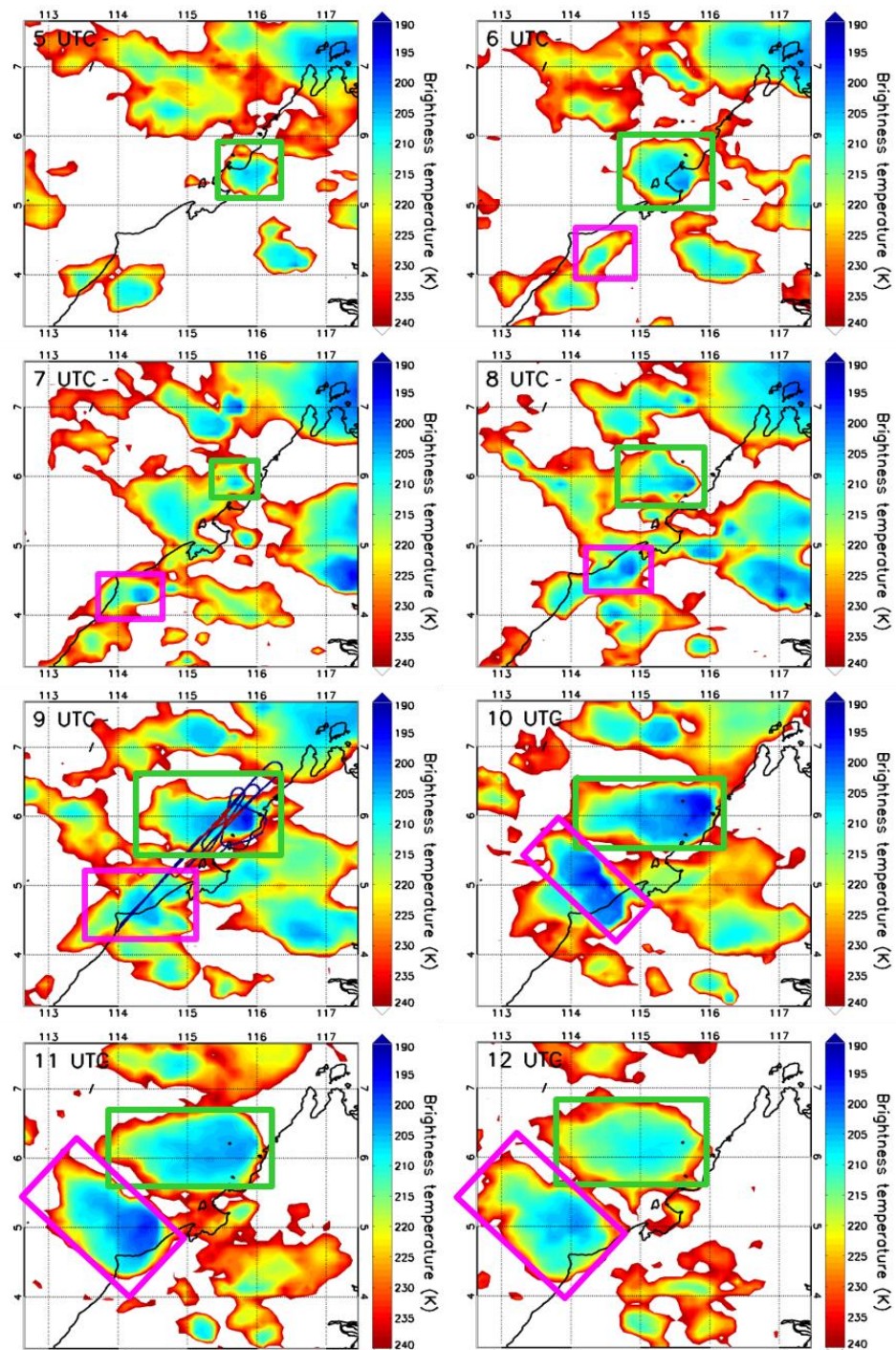

**Figure 1: The contours show the brightness temperatures from MTSAT-2 at (a) 5UTC, (b) 6UTC, (c) 7UTC, (d) 8 UTC, (e) 9UTC, (f) 10 UTC, (g) 11 UTC and (h) 12UTC on November 19 2011. The system called Obs_Conv1 is shown by a green rectangle and Obs_Conv2 by a pink rectangle. The black line shows the coast with the land to the east. The vertical axis is latitude (degrees north) and the horizontal axis is longitude (degrees east). In the 09 UTC panel, the Falcon aircraft trajectory on the afternoon of**


**November 19, 2011 is overplotted. The red crosses indicate the location where the aircraft crossed convective outflows as determined in Krysztofiak et al. (2018).**

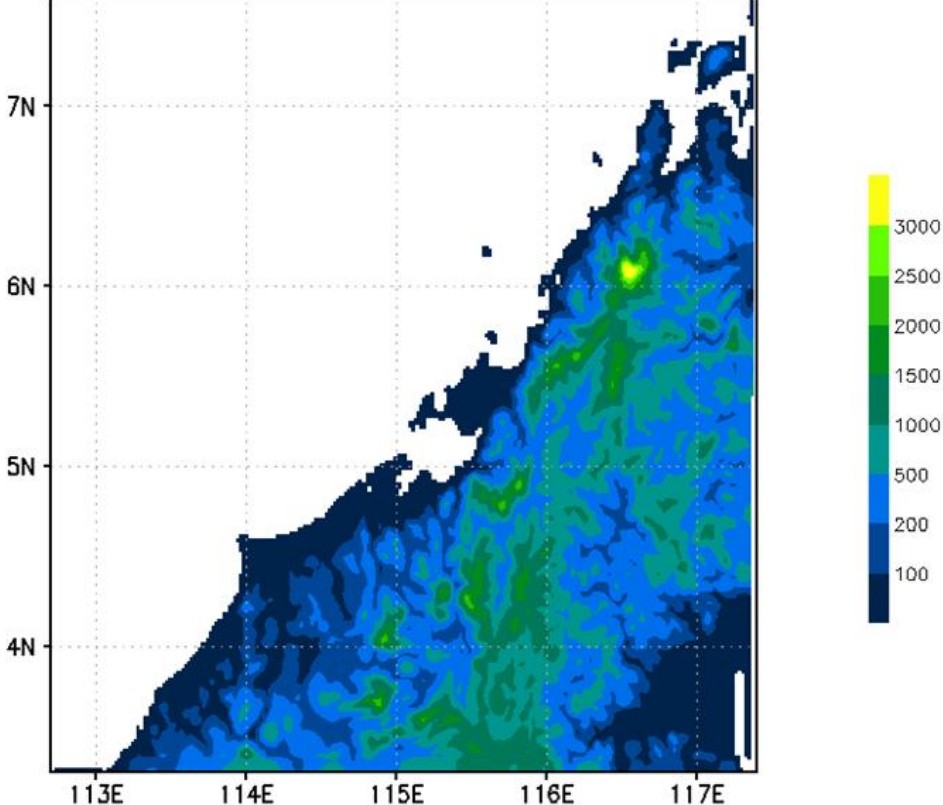

**Figure 2: Map of topography (in m) used in CCATT-BRAMS for the finest scale model grid. The vertical axis is latitude (degrees north) and the horizontal axis is longitude (degrees east).**

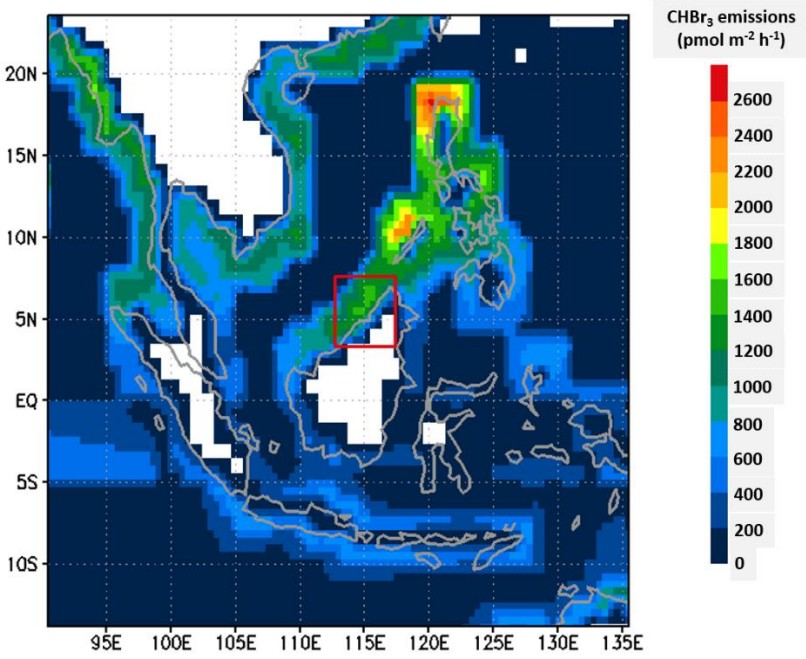


Figure 3: Map of the annual CHBr₃ emission distribution in pmol m⁻² h⁻¹ used in C-CATT-BRAMS in the largest model domain (Ziska et al., 2013). The vertical axis is latitude (degrees north) and the horizontal axis is longitude (degrees east). The red rectangle corresponds to the domain of the finest grid domain displayed in Figs. 1 and 2.

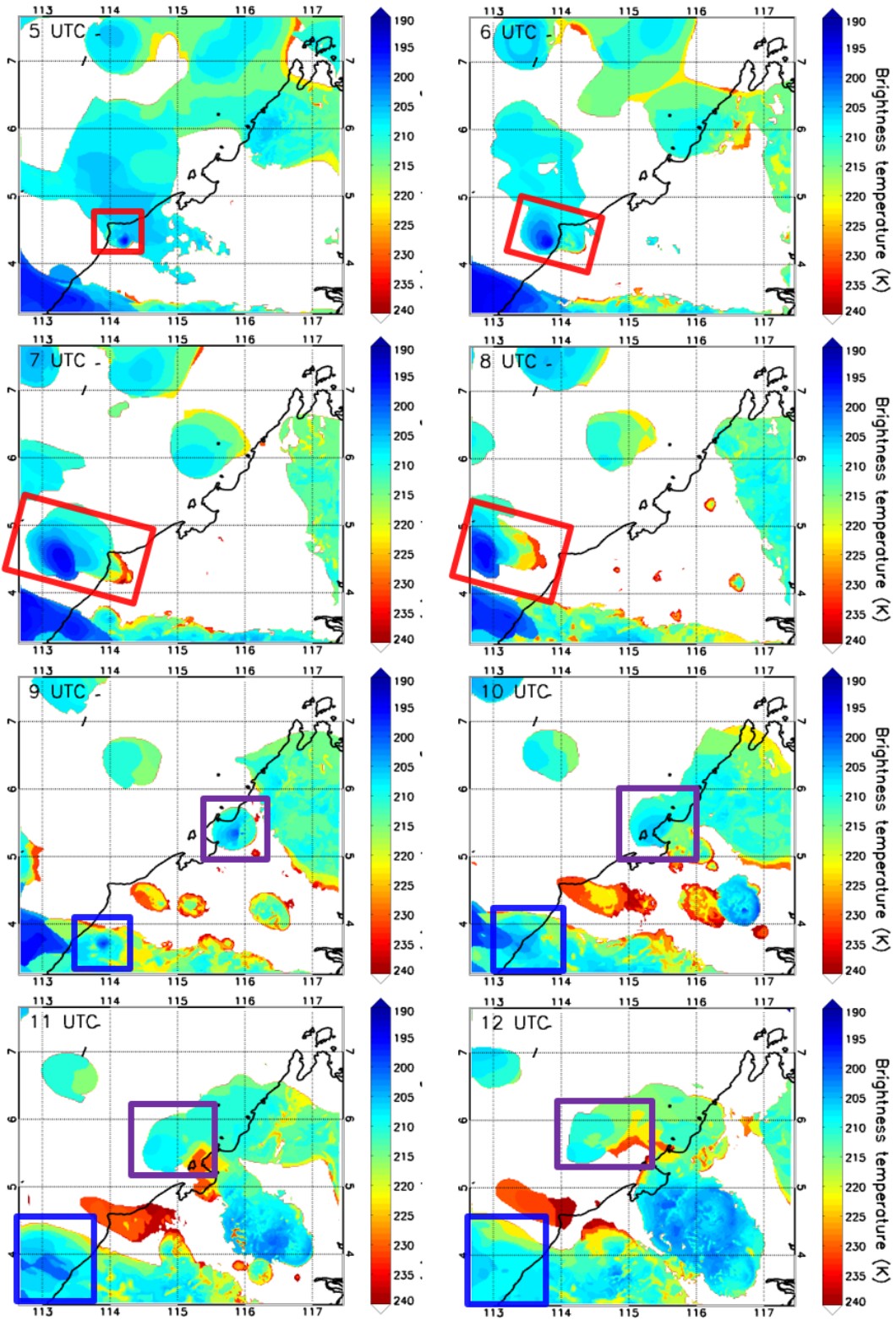

**Figure 4: Same as in Fig. 1 but for the brightness temperatures calculated using the simulation fields (see explanations in the text). (a) 5UTC, (b) 6UTC, (c) 7UTC, (d) 8 UTC, (e) 9UTC, (f) 10 UTC, (g) 11 UTC and (h) 12 UTC). The system called Mod_Conv_4.35N is shown by a red rectangle, Mod_Conv_3.75N by a blue rectangle, and Mod_Conv_5.4N by a purple rectangle. The vertical axis is latitude (degrees north) and the horizontal axis is longitude (degrees east).**

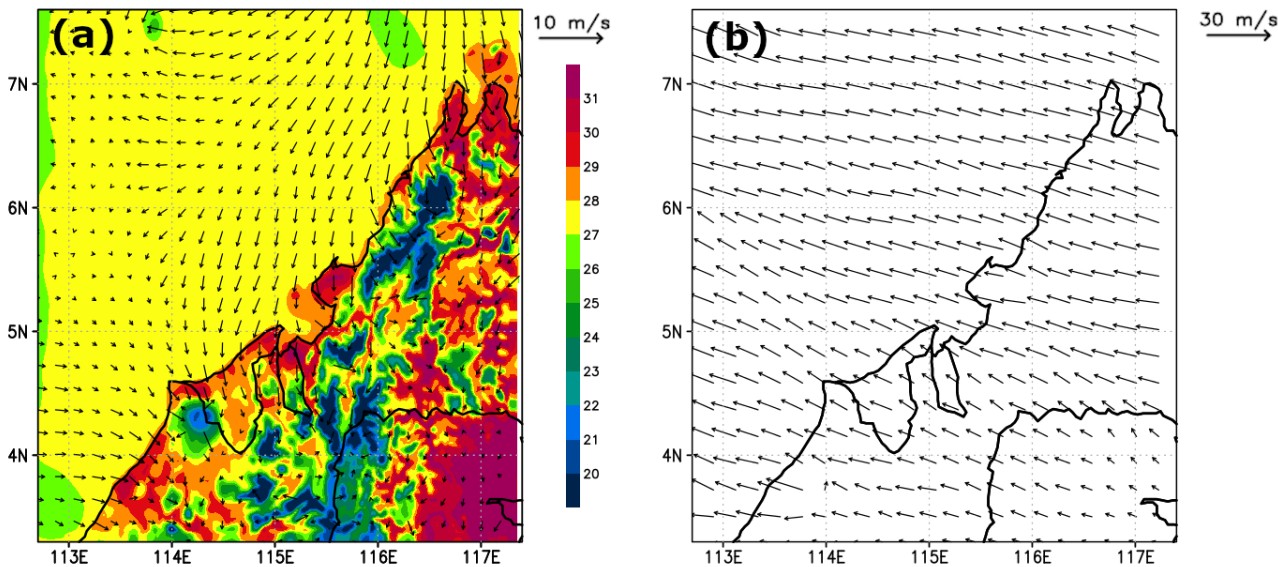

**Figure 5: Simulated (a) temperature and horizontal wind at the lowest model level (24 m) at 6 UTC. (b) horizontal wind at 11700 m altitude at 9UTC from the C-CATT-BRAMS model. The vertical axis is latitude (degrees north) and the horizontal axis is longitude (degrees east).**

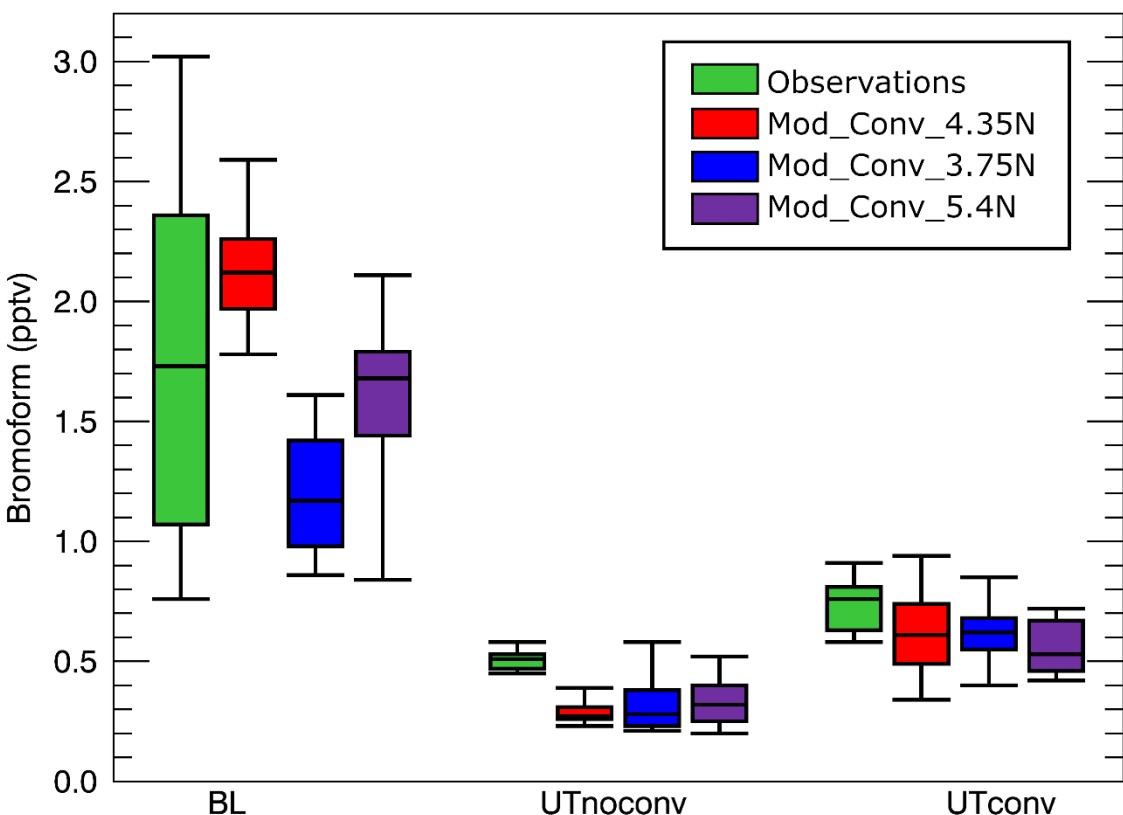

**Figure 6: Box and whiskers plots (5th, 25th percentile, median, 75th percentile, 95th) for the 3 simulated convective system and from the observations of CHBr₃ concentrations in pptv. The green bars show the observed mixing ratios, the red, those of Mod_Conv_4.35N, the blue those of Mod_Conv_3.75N, and the purple those of Mod_Conv_5.4N. From left to right the results as shown for the boundary layer, non-convective upper troposphere and convective troposphere.**

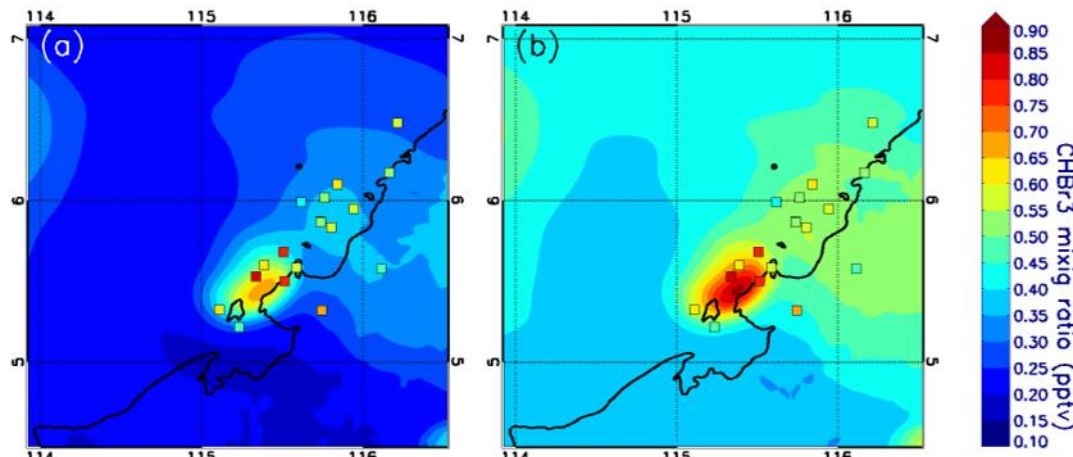

Figure 7: Map of the modelled CHBr$_3$ mixing ratios in pptv for Mod_Conv_5.4N at 10 UTC and 12.5 altitude (panel a). The squares represent the CHBr3 mixing ratios measured by the GHOST instrument within and in the vicinity of Obs_Conv1. 10 UTC corresponds to the time when Mod_Conv_5.4N is at the convective mature stage, i.e., the anvil is well developed. Because of the difference in location between Obs_Conv1 and Mod_Conv_5.4N, the observations are shifted in space to fit with the centre of Mod_Conv_5.4N anvil. Panel b) is similar to panel a) but with 0.17 pptv added to the modelled CHBr$_3$ to account for the underestimation of Mod_Conv_5.4N in the UT (conv and noconv) from Table 3.

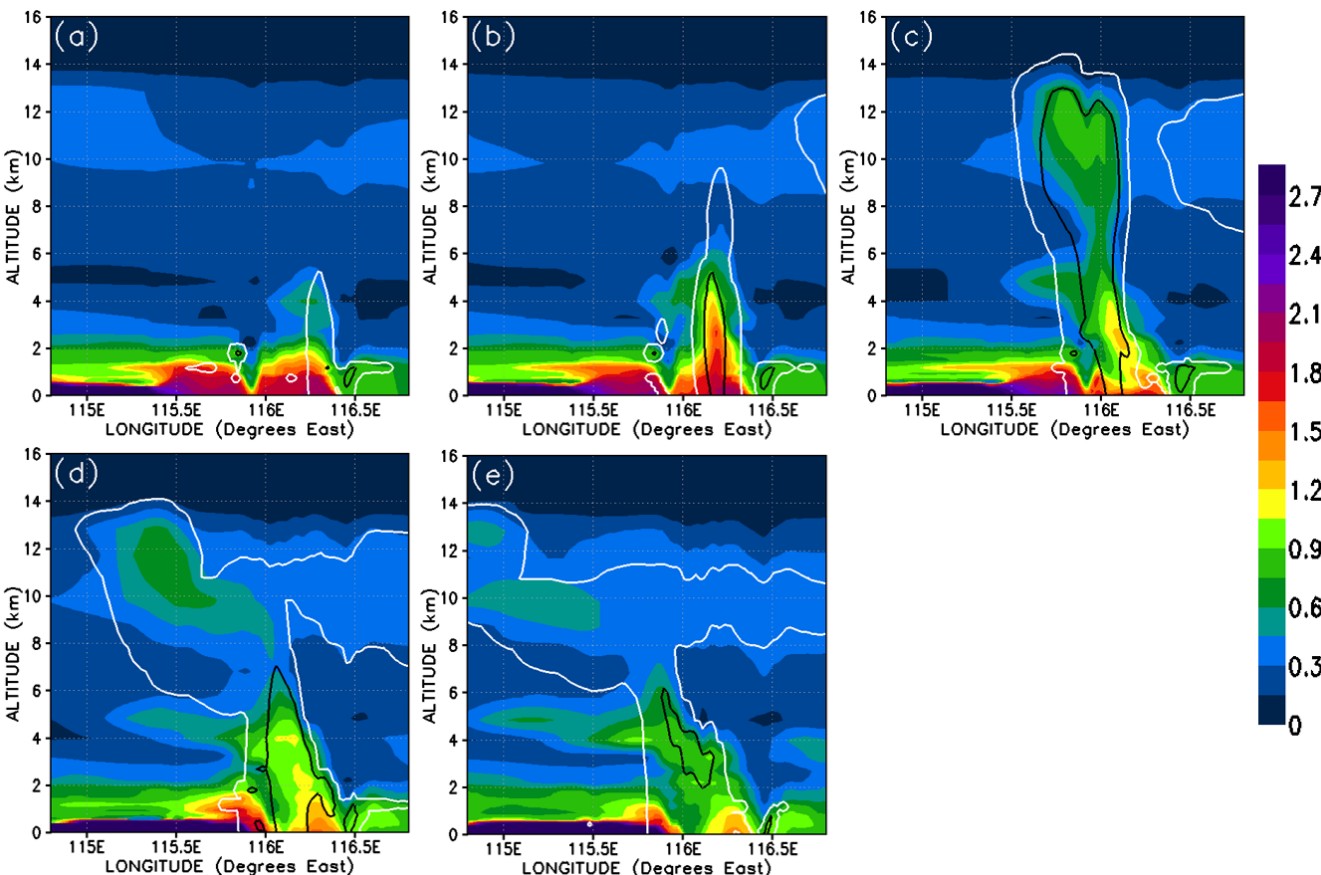

**Figure 8: Vertical cross section within the most active part of Mod_Conv_5.4N convective system (located at 5.4°N) showing time evolution of CHBr$_3$ mixing ratio in pptv for 7 UTC, 8 UTC, 9 UTC, 10 UTC, and 11 UTC. The white and black lines represent the 0.01g kg$^{-1}$ and 0.5 g kg$^{-1}$ contour of the simulated condensed water (cloud and precipitation in ice and liquid phase).**


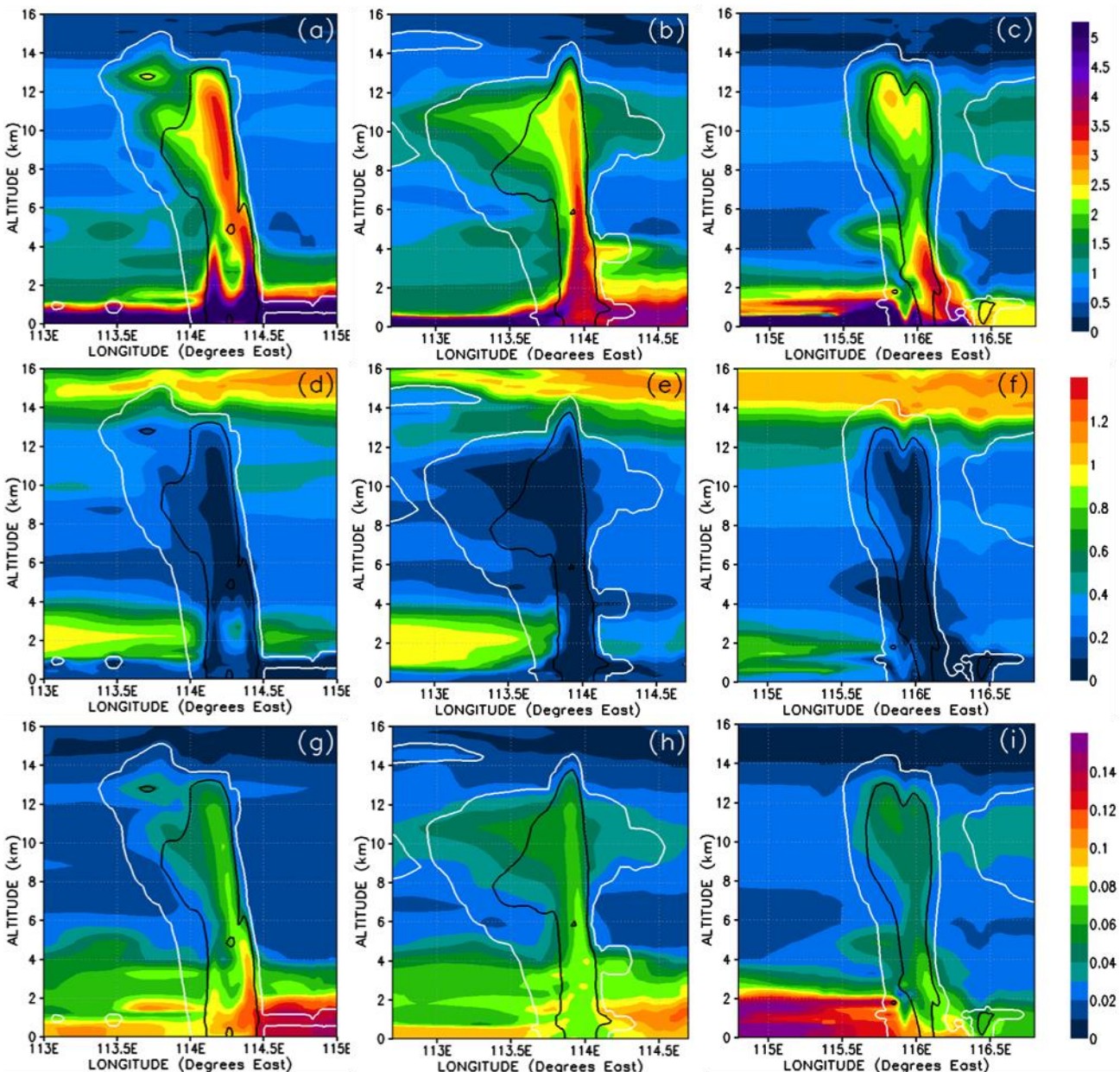

**Figure 9: Vertical cross-sections of mixing ratios (expressed in Br pptv) of CHBr₃ (first row), inorganic (middle row) and organic (bottom row) bromine compounds. The left, middle and right columns correspond to cross sections of Mod_Conv_4.35N at 4.35°N at 6 UTC, Mod_Conv_3.75N at 3.75°N at 9 UTC and Mod_Conv_5.4N at 5.4°N at 9 UTC. The white and black lines represent the 0.01g kg⁻¹ and 0.5 g kg⁻¹ contour of the simulated condensed water (cloud and precipitation in ice and liquid phase), respectively. Note that (c) is identical to Fig. 8 (c) except scaled upwards by 3 for the number of bromine atoms.**

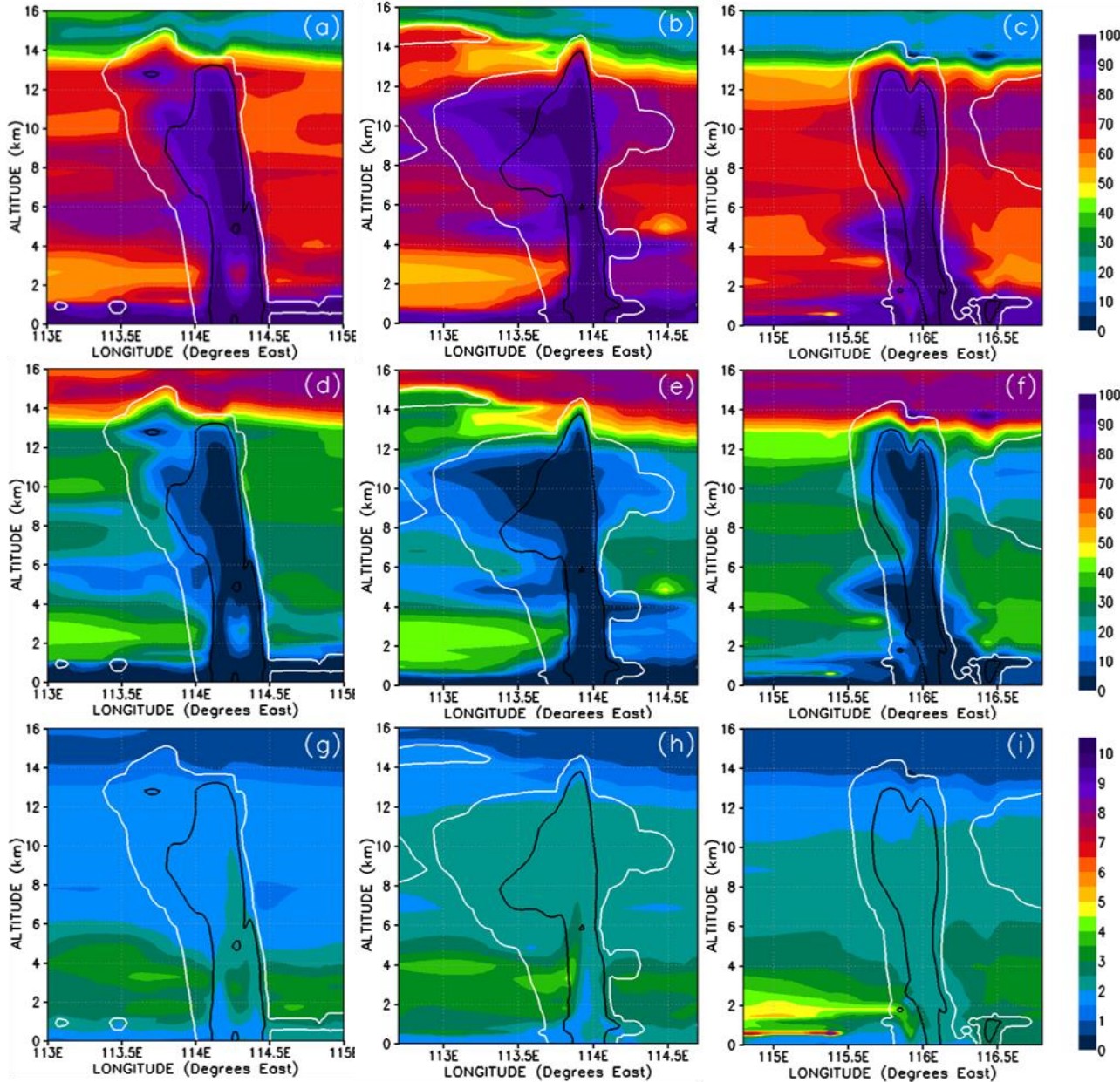

**Figure 10: Vertical cross-sections similar to Fig. 9 but for the percentage contributions from CHBr₃, inorganic and organic bromine compounds to the total Br mixing ratio. Note that for organic bromine the scale is from 0 to 10%.**

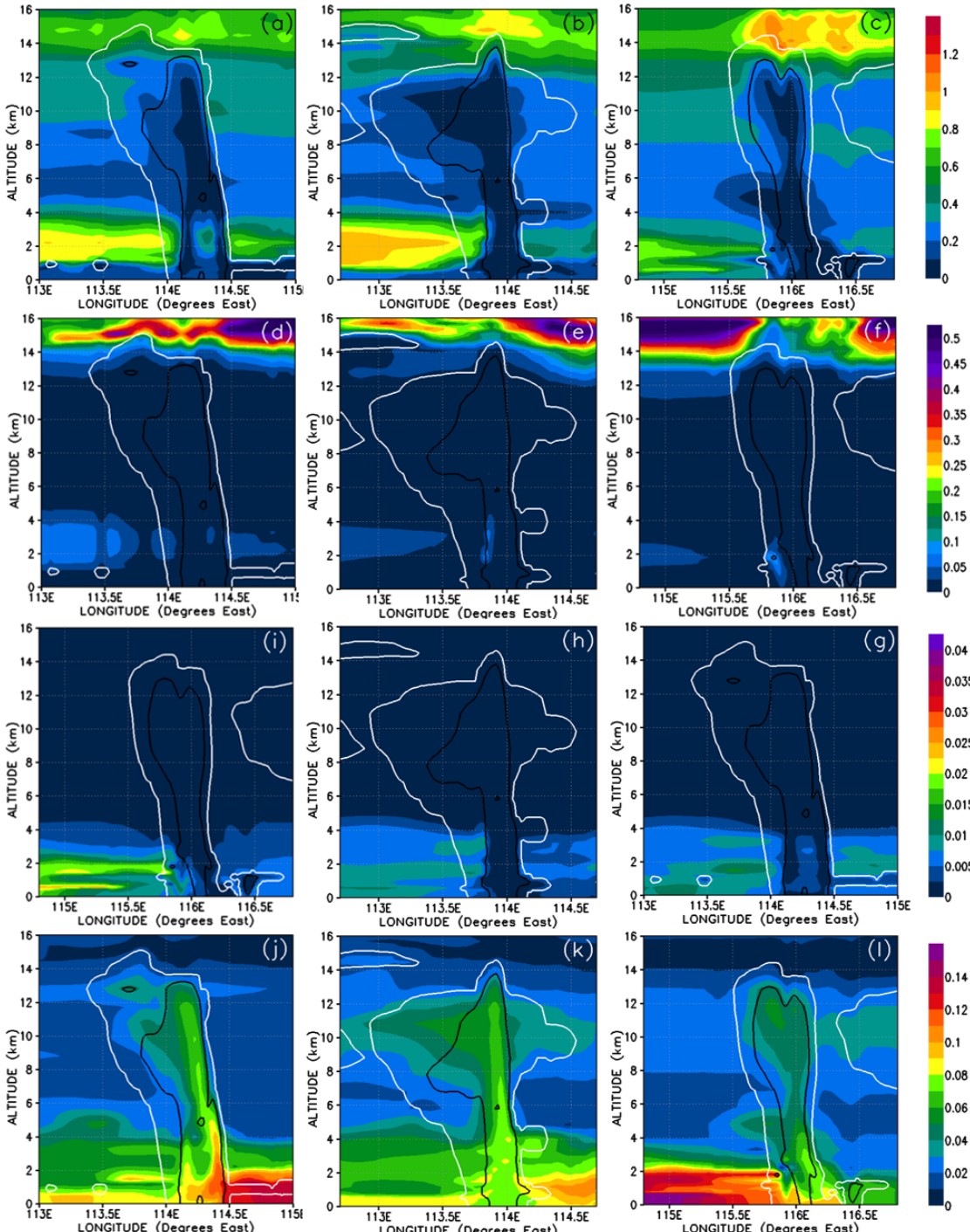

**Figure 11: Vertical cross-sections of mixing ratios (expressed in Br pptv) of soluble inorganic [HOBr, HBr, and BrONO₂] (first row), insoluble inorganic [Br, Br₂, and BrO] (second row), soluble organic [bromo-methyl peroxides] (third row) and insoluble organic [bromo-carbonyls] (bottom row) bromine compounds. The left, middle and right columns correspond to cross sections of Mod_Conv_4.35N at 4.35°N at 6 UTC, Mod_Conv_3.75N at 3.75°N at 9 UTC and Mod_Conv_5.4N at 5.4°N at 9 UTC.**

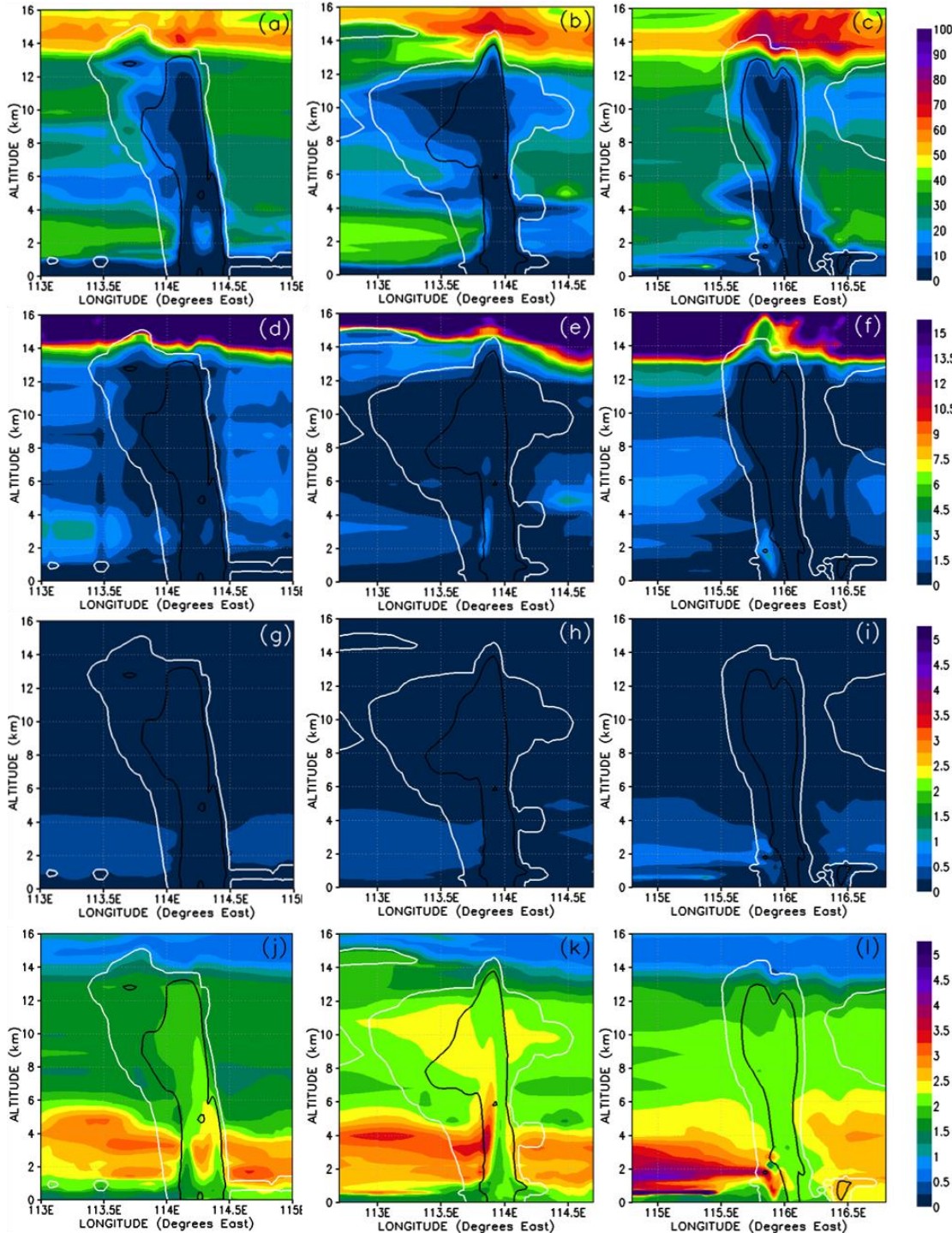

**Figure 12. Vertical cross-sections similar to Fig. 11 but for the percentage contributions from soluble inorganic (first row), insoluble inorganic (second row), soluble organic (third row) and insoluble organic (bottom row) bromine compounds to the total Br mixing ratio. Note that for insoluble inorganic and organic bromine, the scale is from 0 to 5%.**

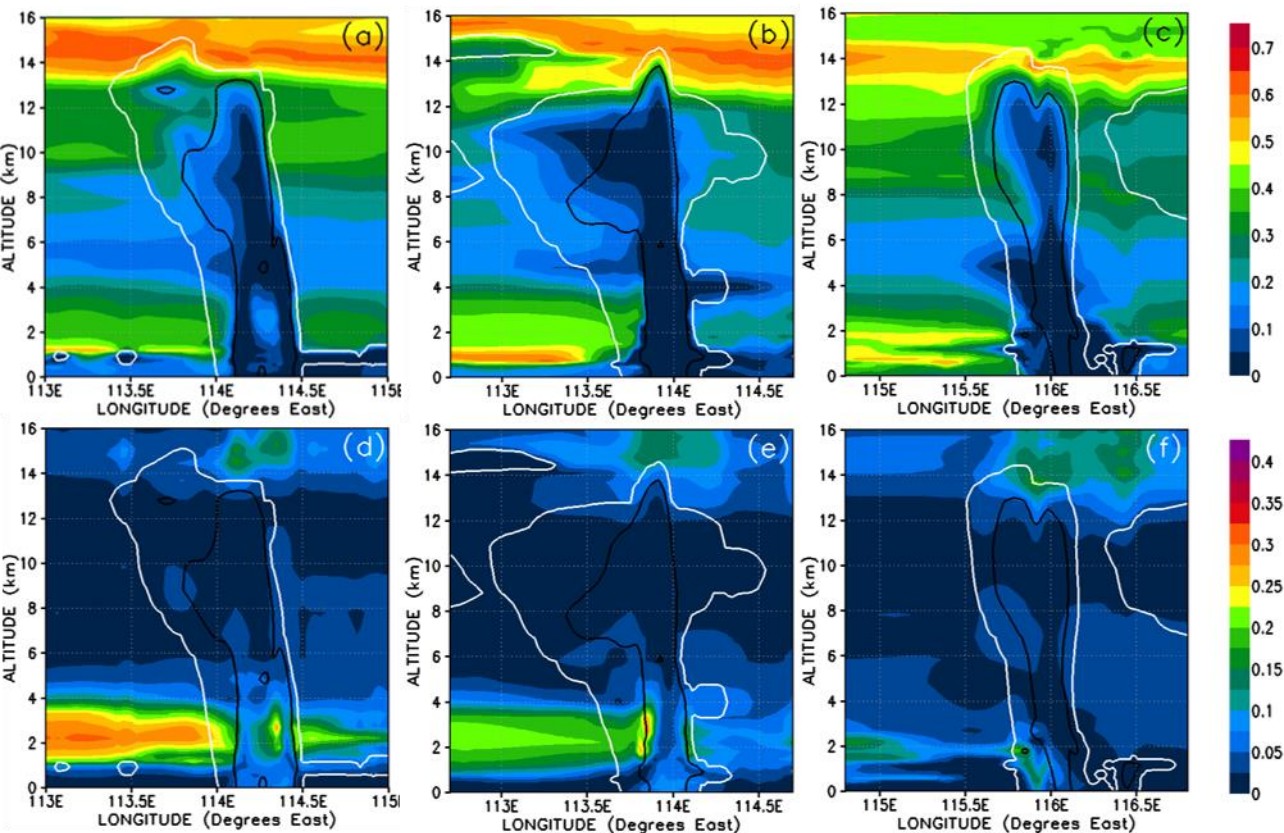

**Figure 13: Vertical cross-sections of mixing ratios (expressed in Br pptv) of HBr (top row) and HOBr (bottom row) bromine compounds. The left, middle and right columns correspond to cross sections of Mod_Conv_4.35N at 4.35°N at 6 UTC, Mod_Conv_3.75N at 3.75°N at 9 UTC and Mod_Conv_5.4N at 5.4°N at 9 UTC.**

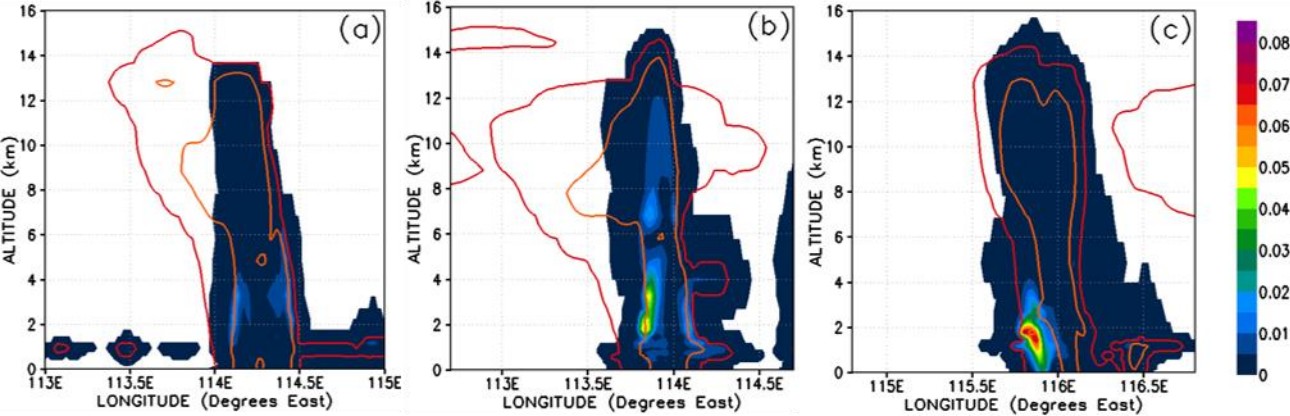

**Figure 14: Vertical cross-sections similar to Fig. 13 but for the percentage contributions of HBr (top row) and HOBr (bottom row) to the total inorganic Br mixing ratio.**

**Figure 15: Vertical cross-sections of mixing ratios (expressed in Br pptv) of Br₂. The left, middle and right columns correspond to cross sections of Mod_Conv_4.35N at 4.35°N at 6 UTC, Mod_Conv_3.75N at 3.75°N at 9 UTC and Mod_Conv_5.4N at 5.4°N at 9 UTC. Note that the lines representing the 0.01g kg⁻¹ and 0.5 g kg⁻¹ contours of the simulated condensed water are plotted in this figure as red and orange, respectively.**