# Peer review of "Cloud-scale modelling of the impact of deep convection on the fate of oceanic bromoform in the troposphere: a case study over the west coast of Borneo"

_Atmospheric Chemistry and Physics, 2020_

## Referee Comment (RC1) · Anonymous Referee #1 · 16 Sep 2020

The paper by Hamer et al. discusses a case study of convective uplift of bromoform over the west coast of Borneo. The authors first analyze the spatial and temporal evolution of two convective systems based on satellite cloud top temperatures, which have been probed in research flights during the SHIVA campaign. The same region is then simulated with the C-CATT-BRAMS atmospheric model and three similar convective systems are identified. The simulated CHBr3 is then compared in a statistical way with measurements of CHBr3 in observed convective systems. Further, convective systems are analyzed in cross sections regarding their uplift of CHBr3 and

its product gases.

This paper addresses an important topic which is certainly within the scope of ACP. However, the manuscript misses to make one consistent work out of several interesting studies. The authors should certainly better motivate each part of their study to illustrate how these different parts contribute to the scientific questions addressed by this paper. In addition, I have identified several major issues and specific comments that need to be addressed before resubmission. I recommend resubmission of this manuscript after major revisions have been made based on the listed issues and comments below.

Major issues:

- The connection between measurements and model study is not clear to me. Measurements from the SHIVA campaign are only used marginally and mostly are used to motivate to simulate the west coast of Borneo in the model study.

- One of the connections between measurements and model is the statistical comparison of measured and simulated $CHBr_3$ mixing ratios at BL, convective and nonconvective UT. Unfortunately, the usage of statistical quantities in comparison with measurements seems to be very arbitrary. The authors should first introduce each statistical quantity and tell the reader what they want to show by examining this specific statistical measure. Then they can proceed to discuss each quantity.

- I was quite disappointed that the only comparison to aircraft measurements was in a statistical way. As indicated by Fig. 2, there are $CHBr_3$ measurements from transects through the Obs_Conv1 convective system. The authors are right that a lat/lon-based comparison between measurement and model would be not reasonable due to the local displacement of the convective systems. But why do they

not show CHBr3 transects through the simulated Mod_Conv3 convective system on similar altitudes as measured? For a comparison, measured and simulated transects could be aligned in the center of the convective system. In my opinion, such a comparison would much better demonstrate a possible agreement between model and measurement compared to only the statistical comparison.

- Throughout the manuscript, but in particular in Section 4.3., figures are not well introduced in the text. Some parts of figures are not even mentioned at all in the manuscript. The authors should either remove these parts from the figures or introduce them in the text. See also in the specific comments.

- The discussion of the PGs is not well motivated. What is the aim of this very lengthy discussion? Do the authors want to explain mixing ratios of CHBr3 PGs in the convective systems, in the convective outflow or in the tropospheric background? With the given plots it is not demonstrated that PGs increase in the upper troposphere due to convective uplift of CHBr3 or PGs. How can the enhanced values of PGs be discriminated from enhanced values of PGs due to background CHBr3 that has been in the upper troposphere without convection? Are PGs transported upwards by convection or are they rather formed in the UT from transported or background CHBr3?

- All discussions about upward transport are limited to transport to the upper troposphere. It is mentioned by the authors, that CHBr3 and PGs are of high relevance in the stratosphere, but it is not even mentioned if brominated air masses reach the stratosphere through the convective systems (in fact, not even the tropopause is marked in any plot). Why is this possibility neglected in the study?

Specific comments:

- 60: "CHBr3, with 3 Br atoms per molecule, has the largest emissions among the different brominated VSLS.": Please quote a suitable reference for this statement.

- 82: "Convective transport and the associated chemistry and washout of all bromine containing species (Bry) cannot be simulated in detail with global 3-dimensional models because of their coarse resolution, and because of the complexity of the chemical processes (e.g. Hossaini et al., 2010).": Why are the authors mixing the two topics here in one sentence? The following sentences are only about convection, not about chemistry and washout, so I don't see any need to introduce this topic already here. In addition, I would talk about "current state-of-the-art global 3-dimensional models" not being able to resolve all convective events.

- 87: "Regarding chemical processes and their interactions with liquid and ice hydrometeors, global models have made progress (Hossaini et al. 2012, Aschmann and Sinnhuber, 2013, Liang et al. 2014), but they still need to compromise between complexity and computing resources.": This sentence is very vague. What are the progresses that have been made by the models? What are the relevant chemical processes and interactions with liquid and ice hydrometeors for CHBr3? Some more detail is missing here.

- 82-93: The whole paragraph should be restructured. After reading it multiple times it became clear to me that the authors are talking about chemical processes during the convective uplift process. This should be mentioned in the beginning of the paragraph, not in the last sentence. In addition: What are the most important reactions of CHBr3 that need modelling in kilometer-scale resolution?

- 97: "... within the optimised mechanism." I'm not a native speaker, but "within" sounds wrong to me. Maybe better use "based on"? But I certainly may be wrong here.

- 129 (and following occurrences): I find it hard to remember which of the convective systems was called "Obs_Conv1" and "Obs_Conv2". I don't see any reason why the authors would need to number these systems. For me it

would be much easier to call them either by their colors (e.g.: "Obs_Conv_blue" and "Obs_Conv_pink") or maybe according to their geographical positions (e.g. "Obs_Conv_northeast" and "Obs_Conv_southwest"). The same comment applies to the "Mod_Conv*" named convective systems later in the manuscript. In addition, the "blue box" in Fig. 1 looks not really blue to me. Maybe the authors could increase brightness and saturation of this color?

- 137 following: The authors could help the reader by either mentioning the panel character in Fig. 1 in the text (e.g.: "Obs_Conv1 was already well developed at 05 UTC (13h local time: 13 LT; Fig. 1a) ..." or by writing the UTC times directly in the panels of Fig. 1. Of course, it is stated in the caption, but for me it took some while to find the corresponding panel from the description in the text.

- Fig. 1: The colorbar used for the brightness temperature includes several maxima and minima in brightness, which distorts the perception of the plot. It is, for example, very difficult to see if the brightness temperature of the blue box convective system increases from panel (f) to (g). Please use a different colormap. Colormaps used for Fig. 8 and following are much better. In addition: A grid line for $117°E$ longitude is missing in the map.

- 144 following and Fig. 2: I'm not really sure what I should see in Fig. 2. The flight path was very complicated and even though the authors tried to mark certain points with time stamps it is not possible to reconstruct the flight path. If the purpose of Fig. 2 should be to prove that measurements have been taken inside Obs_Conv1 and Obs_Conv2, I would suggest to repeat the 9 UTC brightness temperature measurements of Fig. 1 in the background of Fig. 2. Or maybe it is even possible to integrate the flight path (using different colors) in Fig. 1?

- 152 following: "This scenario was confirmed ... CHBr3 measurements were performed by the GHOST ..." The last sentence of this paragraph seems out of

place and gives little information that could have been included in the previous sentence. Also, GHOST has not been defined yet.

- 159: The authors state that "This system is capable of resolving meteorological processes ...", but later it is described that it "includes various physical parameterizations to simulate sub-grid scale meteorological processes ...". The important question for this manuscript is: Is deep convection resolved by the model or parameterized?

- 167: Please define "MODIS NDVI"

- 190: Please define "Fast-TUV"

- 198: "For the model to simulate ..." This sentence sounds strange. Maybe: "Several important changes have been applied to the model to simulate chemical and physical processes associated with CHBr3 degradation chemistry and transport."

- Fig. 4: The authors could help the reader to find Borneo on the coarser map by marking the detailed map boundaries that have been used for Figures 1 2.

- 280 "Therefore, Fig. 5 is mainly used here to show the general temporal and spatial development of the simulated convective systems but does not provide a precise measure of the cloud top height and spread of the anvil.": If this is the purpose of Fig. 5, I do not understand why the authors chose to present cloud top altitudes and not an approximate conversion to brightness temperature. In the given representation of Fig. 5, it is almost impossible to compare to the temporal and spatial development of the measured convective systems in Fig. 1. I would strongly recommend to change Fig. 5 to brightness temperatures instead of cloud top altitudes.

- 283: Same comment regarding the names "Mod_Conv*" as for line 129.

- 299: "Table 2 shows a general agreement on times and altitudes between the observations and the model." This statement is not true for the timing of the convective systems. The simulated convective systems either are started later (8 UTC in the table, but 9 UTC in Fig. 5 compared to 5 and 7 UTC in the measurements) or dissipate earlier (8 UTC compared to 11 UTC in the measurements).

- 308: "the duration of the system of several hours and decay during early evening.": Same here. The duration of the system is always simulated considerably shorter than measured.

- 322: I would call this section differently - it is the part where the model finally is compared the measurements - so I would mention the comparison to the measurements in the title of the section.

- Table 3: Please give units if applicable (I guess it is pptv for all [X] quantities). Also, fraction f is given with a kind of uncertainty (+-...) but in contrast to the [X] quantities, it is not stated what kind of uncertainty is presented.

- 344: "However, this high fraction f is consistent with the average value calculated from all SHIVA aircraft data ...": The authors should introduce the fraction f better. Is it expected to be the same factor f for all convective systems or is it expected to vary between individual convective systems? In the latter case, it would mean that Mod_Conv2 is not comparable to Obs_Conv1 or Obs_Conv2 in terms of the fraction f. It is also worth noting that Mod_Conv2 has a higher uncertainty and agrees with the fraction f of Obs_Conv1 and Obs_Conv2 within the combined uncertainties.

- Figure 7: Please add a legend explaining the colors. It would be also helpful to use the same colors for Mod_Conv* regions as in Fig. 5

- 353 and following: From column [X]BL in Tab. 3, I would say that the model results agree to the measurements within their uncertainties. I don't see the

rationale of discussing single differences in percentage numbers here. For the whole following paragraph it is not clear to me why there are so many different statistical measures used. I would recommend to restructure this paragraph and first explain which statistical measure is used for what purpose. In the current state, the authors jump from a discussion of mean values (without mentioning the 1 sigma errors) to the median with 25 and 75 percentiles. What do I learn from these numbers? The same issue continues for the following paragraphs.

- 366: "If we consider the higher spatial resolution of our simulations and the smaller domain considered for the statistics compared to TOMCAT, these remaining differences appear consistent with one another.": I don't understand this sentence. Please give some context.

- 369 following: This paragraph and the following (starting at line 378) are very similar to the previous paragraph and discuss the differences in background CHBr3 in the UT. Some arguments are repeated in these paragraphs, some are new. Please restructure these three paragraphs to one without repetitions. Instead, a discussion of UTconv CHBr3 is missing completely here. Also median and 25 and 75 percentile information from Fig. 7 is not used here at all.

- 389: "We selected Mod_Conv3 since it corresponds mostly close in space to Obs_Conv1.": This seems a good choice, but unfortunately, the authors do not compare this simulated CHBr3 to observations.

- 393: "...naturally highest closest to the point of convective detrainment..." Please check the formulation. Maybe "... naturally highest, close to the time of convective detrainment ..."?

- Figure 8 and following figures: Please label at least one axis per column and row.

- Figure 9: A short notice would be helpful that Fig. 9c is the same as Fig. 8c scaled by the number of bromine atoms (3). It would be also helpful to repeat the

black and white lines from the first row of Fig. 9 to all other rows and the following figures to guide the eye in a comparison. In addition, these black and white lines would help to identify regions of convective outflow and tropospheric background. Also a tropopause would be helpful for all these kind of plots.

- 438: Missing ")" after "(i)"

- 452: Please define the gases that are summarized as "bromo-carbonyls" and "bromo-methyl peroxides" or use the same names for these groups as in the introduction.

- 453: "CHBr3 is insoluble relative to its PGs" -> "CHBr3 is less soluble than its PGs"

- 469: "These compounds contribute 86

- Figure 14 d-f: The representation of HOBr as percentage of total inorganic Br mixing ratio is dangerous here, because this plot suggests that HOBr has a large contribution to the convective system by showing relative contributions up to 100

- Section 4.3.3: There are plots for insoluble organic bromine compounds in figures 11 and 12, but these are not even mentioned in this section.

- 470 and following: In this paragraph, the "behaviour of inorganic bromine" is discussed. Earlier in this section inorganic bromine was introduced as a large number of species, but here only HBr and HOBr are mentioned. This selective discussion of only two gases needs to be motivated.

- 479: "A key finding is that inorganic bromine dominates the PG budget within the troposphere, yet despite this the inorganic PGs are almost entirely removed during convective transport by washout due to their solubility." This is a broad statement based on a case study. I think the authors should limit this statement to their case study and not leave it in a general sense.

- 480: "We here argue that the regional tropospheric composition present in our simulations is the underlying cause of this prevalence of HBr and in turn the washout of inorganic PGs that results from this.": I find this sentence very unclear and don't understand what the authors try to say here. Please rephrase. This makes it also very hard to understand the motivation for the whole Section

- 4.3.4. Why is so much discussion devoted to the chemical processes?

- 523 following: The comparison to the Marécal et al. (2012) study is very interesting but comes very abrupt here. The authors should consider giving this comparison its own subsection.

- 559: "First, it could be difficult to ..." –> "First, it is difficult to ..."

- 561: "Furthermore, other tropical regions could have vastly different CHBr3 emissions, and in the case of much higher emissions, as was explored in Marécal et al., (2012), we could expect a larger role for Br2 formation." Please check the grammar of this sentence. In addition: What exactly is expected to have a larger role for Br2 formation?

- 570: "Despite the difference in simulated CHBr3 mixing ratios ...": Differences to what?

- 573: "Indeed, our results show consistent CHBr3 mixing ratios compared to the simulations of Hossaini et al. (2013) for the November 19th flight.": To my understanding this consistency between two models using the same emission scenario only proves that both models work properly in terms of chemistry and transport, but it does not prove that the emission scenario is useful, as it is intended by the authors here.

- 593: "Most of the bromine (>85

- 595: Missing subscript in CBr2O

- 598: "Overall, we conclude that organic PGs are more important than inorganic PGs for the vertical transport of bromine within the convective columns for the conditions that we study here." This is not true, because it was stated earlier in this paragraph that most of bromine that was convectively transported was in the form of CHBr3. Maybe the authors want to limit their statement to the vertical transport of PGs.

- 606: "... more important role of the inorganic PGs for the vertical transport of bromine.": The authors have not shown that the enhancements of HBr in the upper troposphere are due to convective transport.

- 608: "Overall, these conclusions are valid in all parts of the convective system except for where the anvil detrains into residual convective outflow in the UT.": Such a statement is not covered by the findings discussed in the main part of this paper.

- 612: In my opinion, section 7 is very short and could be attached to section 6.

---

## Referee Comment (RC2) · Anonymous Referee #2 · 29 Sep 2020

Review of "Cloud-scale modelling of the impact of deep convection on the fate of oceanic bromoform in the troposphere: a case study over the west coast of Borneo" by Hamer et al.

The paper describes a cloud-scale modeling study to investigate convective transport of bromoform ($CHBr_3$) and its product gases. The model is applied to a case study along the west coast of Borneo that was sampled with aircraft measurements, which provide a means to evaluate the model. The major findings are that there is good agreement of $CHBr_3$ mixing ratios between model and observations in the boundary layer and reasonable agreement in convective outflow regions. Analysis of the bromine speciation in convective outflow shows ~85% of the Br is from $CHBr_3$, < 10% from inorganic product gases (HBr, BrO, HOBr, Br, $Br_2$, $BrONO_2$), ~2% from organic product gases (brominated peroxides and carbonyls). The paper finds that the inorganic product gases are dominated by HBr, which is highly soluble and quickly removed by the convection. The paper suggests that the high HBr is a result of the low $O_3$ environment ($O_3$ < 20 ppbv) that favors production of HBr from Br + HCHO and Br + $HO_2$ over production of BrO from Br + $O_3$. Without BrO production, HOBr also has low mixing ratios. Further, aqueous phase reaction between HOBr and HBr is fast, further limiting HOBr within convection.

This study provides fundamental knowledge on the processing of Br compounds in tropical convection, which is important to apply to global models that examine pathways of Br compounds to the stratosphere where Br plays a critical role in ozone chemistry. The paper covers several topics without fully justifying why each topic is addressed. My main concern is that the main points are not concisely given (e.g. in the abstract or results section) but are written along with other points that cause a loss of clarity in the story. For example, why do we need to know the convective transport efficiency? Secondly, why is it important to know if $CHBr_3$, HBr, BrO, or organic Br is the bromine compound transported to the upper troposphere?

In addition, the discussion focuses on the regional chemistry that the convection forms in. The discussion makes sense but is not supported with figures showing rates of reactions.

Specific Science Comments

1. The abstract needs to be improved by making the existing text be more concise, stating more clearly that low $O_3$ levels lead to HBr gas-phase production, and adding comments on the role of aqueous chemistry.
2. Line 102 in Introduction. I do not think it was explained what the limitations are of the previous studies. There should be a few sentences clearly stating these limitations in the Introduction and again in the Discussion.
3. Line 153 states that the $CHBr_3$ measurements were performed by the GHOST instrument. Please add information on what the technique for these measurements is, as well as its detection limits and uncertainties.
4. Since the GHOST instrument is a GC/MS technique, are there other trace gases that it measured that are useful for this analysis, such as other Br-containing trace gases?

5. Line 223. I noticed that the chemistry listed in the supplement does not include Br-Cl reactions, e.g. $Cl + CHBr_3 \rightarrow HCl + CBr_3$. What impact would Cl chemistry have on the results provided in this paper?
6. Does the model include direct uptake of Br compounds onto ice? It may not be important for the species considered (except $BrONO_2$; Fernandez et al., 2014, ACP), but I suggest mentioning this process.
7. Lines 245-249. Could you provide a little more information on the emissions? Do the emissions vary temporally? If not, what time-span is the average for?
8. Line 298. What is the vertical resolution of the model in the upper troposphere? Does the vertical resolution affect the cloud top height estimate? Since the results show a fairly good comparison, I guess not, but I suggest thinking about the uncertainties in the model cloud top height if dz > 500 m.
9. Line 328. Are there any temporal uncertainties to consider? I imagine the statistical approach removes effects of poor timing of the convection, but does it also reduce uncertainties in relation to timing of emissions (if there is a diurnal profile) or photochemistry?
10. Line 330. I first saw the equation given on line 333 in Cohan et al. (1999) JGR (please cite them). This equation assumes that there is no mixing of air at different altitudes between cloud base and cloud top, i.e. it is a two-component mixture model. During the past decades, others have used modified versions of this type of equation. For example, Borbon et al. (2012) JGR applied a three-component mixture model, while Yang et al. (2015) JGR applied a four-component mixture model, and Fried et al. (2016) JGR applied a 10-layer mixture to account for entrainment of air between cloud base and cloud top. The multi-layer approach is useful for conditions when the vertical profiles vary with altitude in the clear air. It would be good to see the clear-sky vertical profile for $CHBr_3$ to show its variability with altitude. Is $CHBr_3$ sufficiently close to zero or non-varying with altitude such that entrainment of air between cloud base and cloud top can be ignored?
11. Lins 388-395. Could you specify where the vertical cross section is located. Is it an average across the anvil, or a line down the center of the anvil?
12. Section 4.3.2. Figures 9 and 10 show values based on Br pptv and percent contribution to total Br. Somehow the text gets a bit confusing. Is it possible to organize the text a little differently, such as by region, which would better align with the Conclusions section. For example, In the UT convective outflow $CHBr_3$ contributes 85%, inorganic PGS < 10%, and organic PGs ~2% to total Br. In the boundary layer, $CHBr_3$ contributes ..... In the 1-4 km layer, $CHBr_3$ contributes .....
13. Line 488. It is suggested that reactions 2 and 3 could be important routes for HBr formation. Reaction 2 is Br + HCHO. Does HCHO come from $CH_4$ or are there VOCs, such as isoprene, contributing to its mixing ratio? What is the typical reactivity for reactions 1 to 3 for the study area? It would help to see reaction rate constants and typical $O_3$, HCHO, and $HO_2$ mixing ratios.
14. Line 505. Again, it would help to know the reaction rate constant and reactivity of reaction 7. Marecal et al. (2012) report $1.6 \times 10^{10}$ $M^{-1}$ $s^{-1}$, which is indeed high but is the rate high as well when compared to competing gas-phase reactions? It would be good to see figures of reaction rates supporting the discussion on what reactions dominate.

15. Line 521. It mentions the very short residence time due to falling hydrometeors. Can you calculate the residence time? I would suggest looking at Bela et al. (2018) JGR who discuss time an air parcel spends in contact with liquid water in convection (ranging from weak convection to severe convection). They also showed via vertical profiles that highly soluble trace gases (like $H_2O_2$) are depleted rapidly below the freezing level, whereas less soluble gases will be lofted higher.

16. Lines 524-557 discuss how the current results compare to those by Marecal et al. (2012). Could the authors make clear what are the differences between the model configuration by Marecal et al. and this paper? I found two comments on the differences, 1) running real meteorology (and thus global model generated initial conditions) and running ideal meteorology, and 2) running a case for Darwin versus Borneo where emissions, photolysis rates, OH mixing ratios can be different. Are there other differences? What about the information in the "new chemistry" section?

17. Line 542. For the discussion of differences due to Darwin and Borneo settings, the lifetime of $CHBr_3$ is discussed. These lifetimes are > 15 days. How do the changes in chemical lifetime affect the results of convective transport, which occurs in < 1 hour and is the topic of the paper?

18. Line 557. Isn't $O_3$ low in both Darwin and Borneo? I do not see why low $O_3$ would cause differences between the results from Marecal et al. and this paper.

19. Section 4. How do the results from this paper fit in the context of findings from the CONTRAST field campaign? I am not sure it is possible to have a thorough discussion, but noticed that Chen et al. (2016) JGR discuss convective transport (or not) of BrO and HOBr + $Br_2$.

20. Section 5. Can results from global model studies (e.g. from TOMCAT, or other CCMI models) help identify other tropical regions that should be studied?

Organization, Clarity, Technical Comments

1. Line 60, I suggest starting the sentence as, "Bromoform ($CHBr_3$), with 3 Br atoms …."
2. Line 589, Suggest: Fuhlbrugge et al. (2016) who showed that ….
3. Line 505, Could you make the negative ion sign more obvious. Perhaps via a symbol or equation editor.
4. I recommend that the Appendix be placed in the supplement.

Figures and Table

1. In Table 1, is the $H_x$ for $Br_2$ correct? A positive value is given for $a_H$ yet $H_x$ is less than $H_{298}$.
2. Is the vertical cross section in Figure 8 in a different location than the other vertical cross sections?

---

## Referee Comment (RC3) · Anonymous Referee #1 · 27 Oct 2020

I am very sorry that I made a mistake in the LaTeX formatting of my previous review. All points containing a "%" somewhere in the text were truncated, because LaTeX uses "%" as indication of comments. Here are the missing points:

Specific comments:

- 469: "These compounds contribute 86% of the PG bromine total." I'm confused by the number of 86%. Where does this number come from?

- Figure 14 d-f: The representation of HOBr as percentage of total inorganic Br

[Figure]

mixing ratio is dangerous here, because this plot suggests that HOBr has a large contribution to the convective system by showing relative contributions up to 100% inside the convective system. From Fig. 13 it is clear that HOBr only has very low background Br pptv above 4 km (and below even a minimum) inside the convective system, but is scaled up in Fig 14 due to the relative minimum in HBr. In my opinion this should be noted in the figure caption to avoid wrong interpretations of Fig. 14.

- 593: "Most of the bromine (>85%) transported to the UT in each convective system is in the form of CHBr3.": This is a very important statement that was not mentioned in the discussions of Section 4.3
* * *

---

## Author Comment (AC1) · 7 Sep 2021

The comment was uploaded in the form of a supplement:
https://acp.copernicus.org/preprints/acp-2020-655/acp-2020-655-AC1-supplement.pdf

---

## Author Response (AR1)

**Response to Reviewer 1**

We thank Reviewer 1 for his/her comments leading to a significant improvement of the manuscript. Note that this response includes the additional comments posted by the reviewer in RC3.

This response is organised with the reviewer's comments in black bold italic font, the authors' answers in black normal font and the text added in the revised manuscript in red font.

Note that following reviewer 2's comments, section 4.3.2 has been organised differently, i.e. with an analysis by regions (BL, within convection, …) to be consistent with the presentation of the conclusion. To improve the clarity of the manuscript, the discussion section (4.3.4) is now split into two sections with the comparison to Marécal et al. (2012) being now in a separate section (new section 4.3.5).

***The paper by Hamer et al. discusses a case study of convective uplift of bromoform over the west coast of Borneo. The authors first analyze the spatial and temporal evolution of two convective systems based on satellite cloud top temperatures, which have been probed in research flights during the SHIVA campaign. The same region is then simulated with the C-CATT-BRAMS atmospheric model and three similar convective systems are identified. The simulated CHBr3 is then compared in a statistical way with measurements of CHBr3 in observed convective systems. Further, convective systems are analyzed in cross sections regarding their uplift of CHBr3 and its product gases. This paper addresses an important topic which is certainly within the scope of ACP. However, the manuscript misses to make one consistent work out of several interesting studies. The authors should certainly better motivate each part of their study to illustrate how these different parts contribute to the scientific questions addressed by this paper. In addition, I have identified several major issues and specific comments that need to be addressed before resubmission. I recommend resubmission of this manuscript after major revisions have been made based on the listed issues and comments below.***

***Major issues:***

***The connection between measurements and model study is not clear to me. Measurements from the SHIVA campaign are only used marginally and mostly are used to motivate to simulate the west coast of Borneo in the model study.***

As explained in section 2, the west coast of Borneo is particularly interesting for studying the chemistry and transport to the upper troposphere of bromoform and motivated the location of the SHIVA campaign. By sampling convective outflows under the influence of high CHBr₃ coastal emission zones, several of SHIVA flights were particularly well designed to document the impact of deep convection on the bromoform distribution in the upper troposphere. This is what motivated the choice of the 19[th] of November case study.

The reason why we have only made use of the SHIVA measurements so far for statistics was that we aimed at capturing the general features of the bromoform distribution. As suggested by the reviewer, the bromoform observations have also been compared to the model on maps (new figure added) to complement the statistics (see more details in the responses below).

***One of the connections between measurements and model is the statistical comparison of measured and simulated CHBr3 mixing ratios at BL, convective and non-convective UT. Unfortunately, the usage of statistical quantities in comparison with measurements seems to be very arbitrary. The authors should first introduce each statistical quantity and tell the reader what they want to show by examining this specific statistical measure. Then they can proceed to discuss each quantity.***

For the comparison, we have chosen to present the results using the same statistical indicators as in Krysztofiak et al. (2018), i.e. the mean +/- σ and f. Additionally, the box and whisker plots are used to characterise the variability of the observed and modelled bromoform mixing ratios. We agree that the

percentile information from figure 7 (now figure 6) were not very much used in the original manuscript. We have modified section 4.2 to make a clearer use of the box and whisker plot.

"We now examine the magnitude of the CHBr$_3$ mixing ratios in the BL and in the UT in both inside and outside of the convection using Table 3 and Fig. 6. The box and whisker plots in Fig. 6 giving the median (i.e. 50$^{th}$ percentile) and the 5$^{th}$, 25$^{th}$, 75$^{th}$ and 95$^{th}$ percentiles provide complementary statistical information to Table 3 on the variability of the observed and simulated CHBr$_3$ mixing ratios. Differences between the median and the mean is a measure of the skewness of the distribution of points. The 5$^{th}$, 25$^{th}$, 75$^{th}$ and 95$^{th}$ percentiles give additional information characterising the low and high values of the distribution of the bromoform mixing ratios.

In the BL, there is a large spread in the observations because of the very large local variability of the emissions that the model cannot capture due the resolution of the CHBr$_3$ emission inventory we used. Nevertheless, the median BL mixing ratios of all three lie within the 25$^{th}$ and 75$^{th}$ percentile of the observed BL mixing ratios, and the 5$^{th}$-95$^{th}$ percentile range of simulated mixing ratios across all three simulated systems lie within the 5$^{th}$-95$^{th}$ percentile range of observed mixing ratios in the BL."

*I was quite disappointed that the only comparison to aircraft measurements was in a statistical way. As indicated by Fig. 2, there are CHBr3 measurements from transects through the Obs_Conv1 convective system. The authors are right that a lat/lon-based comparison between measurement and model would be not reasonable due to the local displacement of the convective systems. But why do they not show CHBr3 transects through the simulated Mod_Conv3 convective system on similar altitudes as measured? For a comparison, measured and simulated transects could be aligned in the center of the convective system. In my opinion, such a comparison would much better demonstrate a possible agreement between model and measurement compared to only the statistical comparison.*

In response to this comment and subsequent comment 389, we have added a figure showing the comparison between the observations and the model. As suggested, we have drawn a map of the model CHBr$_3$ field at 12.5 km altitude which is the average altitude of the upper tropospheric measurements used. We make the comparison between the observations in Obs_Conv1 and Mod_Conv_5.4N (ex-Mod_Conv3) at the convective mature stage when the anvil is well developed corresponding to 10UTC for Mod_Conv_5.4N. Because of the difference in location, we also shift the observations that we plot on the map to fit with the center of Mod_Conv_5.4N anvil (left panel). We also add 0.17pptv to the modelled CHBr$_3$ in the right panel to account for the underestimation of Mod_Conv_5.4N in the UT (UTconv and UTnoconv) as given in Table 3. The comparison is provided in a new Fig 7 (see below) and shows a good agreement when the 0.17 pptv shift is taken into account.

[Figure]

**Figure 7: Map of the modelled CHBr$_3$ mixing ratio in pptv for Mod_Conv_5.4N at 10 UTC and 12.5 altitude (panel a). The squares represent the CHBr$_3$ mixing ratios measured by the GHOST instrument within and in the vicinity of Obs_Conv1. 10 UTC corresponds to the time when Mod_Conv_5.4N is at the convective mature stage, i.e., the anvil is**

**well developed. Because of the difference in location between Obs_Conv1 and Mod_Conv_5.4N, the observations are shifted in space to fit with the centre of Mod_Conv_5.4N anvil. Panel b is similar to panel a but with a 0.17 pptv added to the modelled CHBr$_3$ to account for the underestimation of Mod_Conv_5.4N in the UT (conv and noconv) from Table 3.**

*Throughout the manuscript, but in particular in Section 4.3., figures are not well introduced in the text. Some parts of figures are not even mentioned at all in the manuscript. The authors should either remove these parts from the figures or introduce them in the text. See also in the specific comments.*

We have addressed these specific comments in order to introduce the figures more carefully and to describe the results in all figures.

*The discussion of the PGs is not well motivated. What is the aim of this very lengthy discussion? Do the authors want to explain mixing ratios of CHBr3 PGs in the convective systems, in the convective outflow or in the tropospheric background? With the given plots it is not demonstrated that PGs increase in the upper troposphere due to convective uplift of CHBr3 or PGs. How can the enhanced values of PGs be discriminated from enhanced values of PGs due to background CHBr3 that has been in the upper troposphere without convection? Are PGs transported upwards by convection or are they rather formed in the UT from transported or background CHBr3?*

We agree that the motivations of the discussion were not explained clearly enough and this obviously mislead Reviewer 1.

The aim of the discussion is to analyse the most important processes that control the speciation of the PGs in the lower troposphere not affected by convection. This speciation determines the mix of bromine compounds that can be entrained by convection from the low levels and thereby can be transported into the upper troposphere. Additionally, in this mix, only the insoluble/low solubility species can reach the upper troposphere because of washout, as illustrated by Figs. 11 and 12 (section 4.3.2).

This discussion in section 4.3.4 goes a step further in the analysis of the chemical processes leading to the modelled speciation of bromine compounds in the lower troposphere and is not dealing with the processes occurring in the convective uplift and anvil. These latter processes (transport and washout) are analysed in previous sections 4.3.2 and 4.3.3.

The introduction of section 4.3.4 has been modified as follows:

'We now present a discussion of the most important processes that control the overall speciation of the PGs in the background lower troposphere unaffected by convection in our simulation. This is important because this speciation determines the starting mix of chemical compounds present in the surrounding environment prior to convection and then ultimately what is available for entrainment into the convective system. The speciation of PGs is relevant for determining their potential transport to the upper troposphere since they individually have different solubilities.'

Note that the discussion in section 4.3.4 has been split into two sections. The part of the discussion concerning the comparison with Marécal et al. (2012) has been moved to the new section 4.3.5.

We also specifically address the comments related to distinguishing PGs transported in the convective systems versus those formed from photochemically aged CHBr$_3$ in the UT. We have re-written sections 4.3.2 and 4.3.3, so that these points come through clearer now, but we reiterate the basic arguments underpinning the interpretations and conclusions here:

The reported enhancements or deficits in the CHBr$_3$ and PG mixing ratios in the convective column and UT are based on a set of inferences. These involve comparing the mixing ratios in each region of the atmosphere involved in the convective system, i.e., from BL, to vertical component of convective system, and to outflow, within Figs 9 to 15. We use the contours defining the mass mixing ratios of condensed water to define what is within and what is outside of the convective systems and outflow.

This allows us to conclude in each case the enhancement or deficit of the different chemical components by tracing their path during the convection process. We now explain this basic premise at the beginning of Section 4.3.

"We now briefly explain the underlying methodology used to interpret the model results (in sections 4.3.1, 4.3.2, and 4.3.3) and to conclude whether the convective transport leads to an enhancement or deficit in the mixing ratios for $CHBr_3$ and its PGs within the UT. The enhancements or deficits in the $CHBr_3$ and PG mixing ratios in the convective column and UT that we report are based on comparisons of the simulated mixing ratios in each region of the atmosphere involved in the convective system, i.e., from BL, to vertical component of convective system, and to outflow. We use the simulated mass mixing ratios of condensed water as a metric to define what is within and what is outside of the convective systems and outflow. Mass mixing ratios of 0.5 g kg$^{-1}$ and 0.01 g kg$^{-1}$ are used to define the most intense and outer limits of the convection systems, respectively."

In addition to the changes above we also try to provide a further explanation and summary below of the analysis performed. In both the case of $CHBr_3$ and insoluble organic PGs we see an enhancement 'anomaly' in their mixing ratios within the regions of the convective systems (as defined by the 0.5 g kg$^{-1}$ condensed water contour) above 8 km relative to the mixing ratios outside the core of convective system. The enhancement is even stronger when comparing mixing ratios of both $CHBr_3$ and the insoluble organic PGs within the 0.5 g kg$^{-1}$ contour to the mixing ratios outside of the 0.01 g kg$^{-1}$ contour (above 8 km), which highlights the effect of the dilution of the convective outflow with background UT air.

We conclude the presence of an inorganic PG 'deficit anomaly' in the UT convective outflow due to the fact that inorganic PGs display a range of minima with mixing ratios approaching zero ppt within the convective system (as defined by the 0.5 g kg$^{-1}$ condensed water contour line). Background inorganic PG mixing ratios are higher in the boundary layer than in the convective system itself, which, due to their high solubility, allows us to conclude that washout has taken place in the convective system. This washout deficit anomaly of low inorganic PG mixing ratios is then transported vertically and then detrained into the UT as outflow. The net effect of this is to dilute the higher background UT inorganic PG mixing ratios with air depleted of the PGs and lower the UT inorganic PG levels. This interpretation is strongly supported by the increasing mixing ratios that we see moving from within the 0.5 g kg$^{-1}$ condensed water contour to the region between the 0.5 g kg$^{-1}$ and 0.01 g kg$^{-1}$ contours, and outwards beyond the 0.01 g kg$^{-1}$ condensed water contour into the background UT.

***All discussions about upward transport are limited to transport to the upper troposphere. It is mentioned by the authors, that CHBr3 and PGs are of high relevance in the stratosphere, but it is not even mentioned if brominated air masses reach the stratosphere through the convective systems (in fact, not even the tropopause is marked in any plot). Why is this possibility neglected in the study?***

As explained in the introduction, air masses that reach the stratosphere from the troposphere in the tropics follow two pathways. For the convective systems overshooting in the stratosphere (first pathway), there is a direct transport from the low troposphere to the stratosphere, but these events are rare. Our case study corresponds to the most common case, in which deep convection reaches the upper troposphere. From there, air is lifted to the stratosphere if above the level of zero radiative heating, around 15 km altitude in clear sky conditions in the tropics and lower in deep convective cloudy (anvil cirrus) conditions (Corti et al. 2005 2006). This is the second pathway. In this case, the air masses in the upper troposphere and their composition, here bromoform and PGs, impact the stratospheric composition after slow radiatively driven ascent over a few months. Over such a time period, bromoform would be expected to age photochemically and entirely into PGs. However, that longer term transport and photochemical ageing is beyond the scope of this paper. Here we focus on just the initial vertical transport and processes in the convective system on the timescale of a few hours. Studying the detailed processes (chemistry associated to meteorology) and associated bromine partitioning (bromoform, organic and inorganic PGs) in the upper troposphere linked to deep

convection is therefore relevant for the stratospheric composition. These arguments were not explicitly given. The description of the aims of the paper in the introduction (second to last paragraph) has been completed by the following sentences:

"The case study corresponds to tropical deep convection reaching the upper troposphere that is far more common than overshooting convection. Because part of the air masses within clouds resulting from convective outflow in the tropical upper troposphere reach the stratosphere by radiative ascent, this paper is relevant for global stratospheric studies, in particular with global models that are only representing this type of pathway and not overshooting convection."

**Specific comments:**

*60: "CHBr3, with 3 Br atoms per molecule, has the largest emissions among the different brominated VSLS.": Please quote a suitable reference for this statement.*

Thank you, we have added a reference now.

*82: "Convective transport and the associated chemistry and washout of all bromine containing species (Bry) cannot be simulated in detail with global 3-dimensional models because of their coarse resolution, and because of the complexity of the chemical processes (e.g. Hossaini et al., 2010).": Why are the authors mixing the two topics here in one sentence? The following sentences are only about convection, not about chemistry and washout, so I don't see any need to introduce this topic already here. In addition, I would talk about "current state of- the-art global 3-dimensional models" not being able to resolve all convective events.*

*87: "Regarding chemical processes and their interactions with liquid and ice hydrometeors, global models have made progress (Hossaini et al. 2012, Aschmann and Sinnhuber, 2013, Liang et al. 2014), but they still need to compromise between complexity and computing resources.": This sentence is very vague. What are the progresses that have been made by the models? What are the relevant chemical processes and interactions with liquid and ice hydrometeors for CHBr3? Some more detail is missing here.*

*82-93: The whole paragraph should be restructured. After reading it multiple times it became clear to me that the authors are talking about chemical processes during the convective uplift process. This should be mentioned in the beginning of the paragraph, not in the last sentence. In addition: What are the most important reactions of CHBr3 that need modelling in kilometer-scale resolution?*

Comments 82, 87 and 82-93 have been addressed together since related. See responses below.

The paragraph has been restructured so that the focus on processes occurring in deep convection is clearer, starting from fine scale modelling before discussing global scale modelling. Details have been given on the simplifications considered in the global model studies that were cited. About km-scale modelling, the most important processes to be included have been listed. The revised version of the paragraph is the following.

"Since $CHBr_3$ transport by deep convection occurs at the local scale and involves complex chemistry, the analysis of the detailed $CHBr_3$ and PG processes occurring within deep convection and in its vicinity requires fine scale modelling at the kilometer-resolution with detailed chemistry (e.g., Barth et al., 2001; Marécal et al., 2006). These processes are the convective-scale transport and mixing, the full bromoform degradation scheme in the gaseous phase, the speciation of the resulting PGs into organic and inorganic forms, the partitioning of PGs across the gas-aqueous phases due to their solubilities and interactions with hydrometeors during formation, mature and decaying convective stages, hydrolysis of $BrONO_2$ within cloud and rain droplets, and aqueous phase chemistry of dissolved gases in cloud and rain droplets. This knowledge, gained from studies at the convective scale, may then improve the representation of the fate of chemical species in global models. Because of their coarse resolution, current state-of-the-art global 3-dimensional models use sub-grid scale parameterization of deep convection and are not able to resolve all convective events. These parameterizations are a known

source of uncertainty for tracer transport, including CHBr$_3$, from the boundary layer to the upper troposphere (e.g. Hoyle et al. 2011, Liang et al., 2014; Hossaini et al., 2016; Butler et al., 2018). Also, because global models need to compromise between complexity and computing resources, they include simplifications of CHBr$_3$ chemical processes and their interactions with hydrometeors. For instance, in Hossaini et al. (2012), CHBr$_3$ degradation is assumed to release Br$_y$ immediately. Liang et al. (2014) study is based on a stratospheric model and uses a OH climatology in the troposphere. Aschmann and Sinnhuber (2013) represent in their stratospheric model the partitioning between inorganic species and HBr uptake on ice but no organic products and no explicit tropospheric chemistry is included."

*97: "... within the optimised mechanism." I'm not a native speaker, but "within" sounds wrong to me. Maybe better use "based on"? But I certainly may be wrong here.*

We have changed this to "included within" to be a bit clearer.

*129 (and following occurrences): I find it hard to remember which of the convective systems was called "Obs_Conv1" and "Obs_Conv2". I don't see any reason why the authors would need to number these systems. For me it would be much easier to call them either by their colors (e.g.: "Obs_Conv_blue" and "Obs_Conv_pink") or maybe according to their geographical positions (e.g. "Obs_Conv_northeast" and "Obs_Conv_southwest"). The same comment applies to the "Mod_Conv*" named convective systems later in the manuscript. In addition, the "blue box" in Fig. 1 looks not really blue to me. Maybe the authors could increase brightness and saturation of this color?*

We have changed the modelled convective system names to Mod_Conv_'Latitude', i.e., Mod_Conv_4.35N Mod_Conv_3.75N Mod_Conv_5.4N. We have also changed all of the colours of the boxes to be consistent between Fig.1, Fig. 5 (new figure 4), and Fig. 7 (new figure 6).

*137 following: The authors could help the reader by either mentioning the panel character in Fig. 1 in the text (e.g.: "Obs_Conv1 was already well developed at 05 UTC (13h local time: 13 LT; Fig. 1a) ..." or by writing the UTC times directly in the panels of Fig. 1. Of course, it is stated in the caption, but for me it took some while to find the corresponding panel from the description in the text.*

The times have been added to figures 1 and 5 (Fig. 5 being the new figure 4).

*Fig. 1: The colour bar used for the brightness temperature includes several maxima and minima in brightness, which distorts the perception of the plot. It is, for example, very difficult to see if the brightness temperature of the blue box convective system increases from panel (f) to (g). Please use a different colormap. Colormaps used for Fig. 8 and following are much better. In addition: A grid line for 117_E longitude is missing in the map.*

We have been able to modify the colour bar to be a blue to red colour bar corresponding to colder to warmer temperatures. It was not possible to match the colour maps in Figure 8 because we used different plotting software in each case.

*144 following and Fig. 2: I'm not really sure what I should see in Fig. 2. The flight path was very complicated and even though the authors tried to mark certain points with time stamps it is not possible to reconstruct the flight path. If the purpose of Fig. 2 should be to prove that measurements have been taken inside Obs_Conv1 and Obs_Conv2, I would suggest to repeat the 9 UTC brightness temperature measurements of Fig. 1 in the background of Fig. 2. Or maybe it is even possible to integrate the flight path (using different colors) in Fig. 1?*

Following this suggestion, Figure 2 has been removed and the flight path has been plotted in Figure 1 on the 9UTC panel since the measurements were gathered around this time.

***152 following: "This scenario was confirmed ... CHBr3 measurements were performed by the GHOST ..." The last sentence of this paragraph seems out of place and gives little information that could have been included in the previous sentence. Also, GHOST has not been defined yet.***

We have merged the two sentences and removed the last sentence referring to GHOST measurements that was not needed there.

"Since CHBr$_3$ emissions and its marine boundary layer (BL) mixing ratios are large close to Borneo's west coast (Ziska et al., 2013; Fuhlbrügge et al., 2016), CHBr$_3$ was transported from the BL aloft by the Obs_Conv_4.35N and Obs_Conv_3.75N systems as confirmed by observations of elevated CHBr$_3$ mixing ratios relative to the background conditions during the flight (Sala et al., 2014; Krysztofiak et al., 2018)."

***159: The authors state that "This system is capable of resolving meteorological processes ...", but later it is described that it "includes various physical parameterizations to simulate sub-grid scale meteorological processes ...". The important question for this manuscript is: Is deep convection resolved by the model or parameterized?***

Depending on the resolution chosen by the user, deep convection needs to be parameterized or not. In our case, deep convection is explicitly resolved in the finest grid since its 2 km resolution does not require to use a sub-grid scale parameterization. We have added the following sentence to make this clearer.

"Note that for km-scale simulations, convection is resolved explicitly and thus subgrid-scale convective parameterizations are not needed."

***167: Please define "MODIS NDVI"***

Done.

***190: Please define "Fast-TUV"***

Done.

***198: "For the model to simulate ..." This sentence sounds strange. Maybe: "Several important changes have been applied to the model to simulate chemical and physical processes associated with CHBr3 degradation chemistry and transport."***

Done.

***Fig. 4: The authors could help the reader to find Borneo on the coarser map by marking the detailed map boundaries that have been used for Figures 1 2.***

A red square has been added to figure 4 (new figure 3) to show the boundaries of Fig. 1 that correspond to the domain of the finest model grid.

***280 "Therefore, Fig. 5 is mainly used here to show the general temporal and spatial development of the simulated convective systems but does not provide a precise measure of the cloud top height and spread of the anvil.": If this is the purpose of Fig. 5, I do not understand why the authors chose to present cloud top altitudes and not an approximate conversion to brightness temperature. In the given representation of Fig. 5, it is almost impossible to compare to the temporal and spatial development of the measured convective systems in Fig. 1. I would strongly recommend to change Fig. 5 to brightness temperatures instead of cloud top altitudes.***

We agree that it is better to make a comparison in brightness temperatures. As proposed, we have applied a simple method to convert the model cloud top to brightness temperatures. We have used a calculation in RTTOV v12.3 (Radiative Transfer of TOVS, https://nwpsaf.eu/site/software/rttov/, Saunders et al. 2018) to derive the brightness temperature from the cloud top pressure. Figure 5 showing cloud top altitude has been replaced by brightness temperatures. The analysis of the results shown in the new figure 4 (see below with caption) has hardly been changed since the maps of modelderived brightness temperatures are consistent with the cloud top altitude maps that were shown in the original version of the paper.

[Figure]

**Figure 4: Same as in Fig. 1 but for the brightness temperatures calculated using the simulation fields (see explanations in the text). (a) 5UTC, (b) 6UTC, (c) 7UTC, (d) 8 UTC, (e) 9UTC, (f) 10 UTC, (g) 11 UTC and (h) 12 UTC). The system called Mod_Conv_4.35N is shown by a red rectangle Mod_Conv_3.75N by a blue rectangle, and Mod_Conv_5.4N by a purple rectangle. The vertical axis is latitude (degrees north) and the horizontal axis is longitude (degrees east).**

Saunders, R., Hocking, J., Turner, E., Rayer, P., Rundle, D., Brunel, P., Vidot, J., Roquet, P., Matricardi, M., Geer, A., Bormann, N., and Lupu, C., 2018: An update on the RTTOV fast radiative transfer model (currently at version 12), Geosci. Model Dev., 11, 2717-2737, https://doi.org/10.5194/gmd-11-2717-2018.

"To show the evolution of the modelled convective systems, we plot in Fig. 4 the model-derived brightness temperatures at 11 mm (IR108 channel wavelength) estimated from cloud top pressures from 05 UTC to 12 UTC on November 19th, 2011 using RTTOV v12.3 (Radiative Transfer of TOVS, https://nwpsaf.eu/site/software/rttov/, last access: 23 June 2021, Saunders et al. 2018)."

**283: Same comment regarding the names "Mod_Conv*" as for line 129.299: "Table 2 shows a general agreement on times and altitudes between the observations and the model." This statement is not true for the timing of the convective systems. The simulated convective systems either are started later (8 UTC in the table, but 9 UTC in Fig. 5 compared to 5 and 7 UTC in the measurements) or dissipate earlier (8 UTC compared to 11 UTC in the measurements).**

**308: "the duration of the system of several hours and decay during early evening.": Same here. The duration of the system is always simulated considerably shorter than measured.**

We answer comments 283 and 308 together since they are related.

Regarding the time of the modelled convective systems, we agree that the timing is not exactly the same as observed. In the observations, we see that the two convective systems originating on the west coast of Borneo do not first reach the upper troposphere at the same time (5UTC and 7 UTC) and the duration of the system before they start to dissipate also varies (5h and 4h). In the model, the three convective systems show variations of these two parameters which are close to those observed. The time when convection first reaches the upper troposphere is off by 1h maximum compared to observations. Regarding the dissipation time, the notation for the two systems that leave the model domain (using '>') might not have been clear enough. For these two systems, there is an uncertainty since we can only estimate that the dissipation stage starts at least at the last time when they are in the model domain but it might be a bit later.

We also want to stress (as explained in section 4.1) that the state-of-the-art literature shows that even km-scale simulations often fail to provide exact time of tropical convection. This is why we do not aim at reproducing exactly the timing of the convective systems but what is important for the bromoform chemistry which is that the convective transport to the upper troposphere takes place in afternoon conditions when photolysis is active. This is the case of all three modelled systems.

We have changed the notation '>' by 'at least' in Table 2 to make it clearer. We have added more details in the analysis of Table 2.

"Regarding the timing, the two observed convective systems originating on the west coast of Borneo do not first reach the upper troposphere at the same time (5UTC for Obs_Conv1 and 7 UTC for Obs_Conv2) and the duration before they start to dissipate also varies (5h for Obs_Conv1 and 4h for Obs_Conv2). In the model, the three convective systems also show variations of these two parameters which are close to those observed. The time when convection first reaches the upper troposphere in the model is only off by 1h maximum (08 UTC) compared to observations (07 UTC) and still occurs in the afternoon. Regarding the dissipation time, there is an uncertainty because two of the systems leave the model domain. However, the model simulates anvils that last at least 3h before they start to dissipate."

**322: I would call this section differently - it is the part where the model finally is compared the measurements - so I would mention the comparison to the measurements in the title of the section.**

We have now modified the title of this section to: "4.2 Comparison of the Measured and Modelled Bromoform Statistics and Convective Transport Efficiency"

***Table 3: Please give units if applicable (I guess it is pptv for all [X] quantities). Also, fraction f is given with a kind of uncertainty (+-...) but in contrast to the [X] quantities, it is not stated what kind of uncertainty is presented.***

It is now explained in the table caption that [X] is in pptv and that the error on $f$ is calculated by propagating the errors on [X] in the calculation of $f$. This step is now explained in Supplement S3.

***344: "However, this high fraction f is consistent with the average value calculated from all SHIVA aircraft data ...": The authors should introduce the fraction f better. Is it expected to be the same factor f for all convective systems or is it expected to vary between individual convective systems? In the latter case, it would mean that Mod_Conv2 is not comparable to Obs_Conv1 or Obs_Conv2 in terms of the fraction f. It is also worth noting that Mod_Conv2 has a higher uncertainty and agrees with the fraction f of Obs_Conv1 and Obs_Conv2 within the combined uncertainties.***

We have added information on $f$ and values for $f$ from the literature in the revised text. $f$ varies from one convective system to another. By comparing $f$ from the simulation to the observed ones we evaluate the model ability to transport air masses from the boundary layer to the upper troposphere. Observed and modelled values of $f$ are consistent (within the same range) indicating that the simulation provides realistic convective transport. We agree with your last remark that we have include in the revised text.

Reviewer #2 has also raised questions regarding $f$. The changes shown below address the comments from both reviewers on the introduction of section 4.2 and the explanation of $f$.

"Before discussing the results of the simulated chemistry in detail (section 4.3), we evaluate if the simulation gives reasonable results for $CHBr_3$ concentrations and for convective transport efficiency compared to the aircraft observations.

We firstly use statistical characteristics for this comparison. We choose this approach for two reasons. First because of differences in location and timing between the observed and simulated convection events, and, second, because of spatial uncertainties in the emission inventory used in the simulation. This approach allows a clearer comparison of the observations and simulation by removing effects arising from inherent time and spatial uncertainties.

In order to compare the convective transport efficiency between the observed and simulated systems we follow the approach proposed by Cohan et al. (1999) and used by Bertram et al. (2007). To estimate the air fraction, $f$, originating from the boundary layer (BL) and transported by convection we use the relationship from Cohan et al. (1999):

$$[X]_{UTconv} = f \, . \, [X]_{BL} + (1-f) \, . \, [X]_{UTnoconv}$$

where the mean mixing ratios in the boundary layer, the upper troposphere within the convective systems, and the upper troposphere in the vicinity but outside the convective systems are represented by $[X]_{BL}$, $[X]_{UTconv}$, and $[X]_{UTnoconv}$, respectively. $f$ ranges from 0 to 1 with large values corresponding to an efficient convective transport of air masses from the boundary layer to the upper troposphere. This formulation of $f$ is chosen because it was recently applied to the SHIVA aircraft data (Krysztofiak et al., 2018). It relies on the assumption of a low variability of background concentrations with altitude, which is fulfilled for $CHBr_3$ in our case study (not shown). Previous studies based on observations and reported in Krysztofiak et al (2018) provides estimates of $f$ in the range 0.17 to 0.36."

***Figure 7: Please add a legend explaining the colors. It would be also helpful to use the same colors for Mod_Conv* regions as in Fig. 5***

We have now added a legend and made all of the colours consistent between Figs. 1, 5 (new figure 4), and 7 (new figure 6).

***353 and following: From column [X]BL in Tab. 3, I would say that the model results agree to the measurements within their uncertainties. I don't see the rationale of discussing single differences in***

*percentage numbers here. For the whole following paragraph it is not clear to me why there are so many different statistical measures used. I would recommend to restructure this paragraph and first explain which statistical measure is used for what purpose. In the current state, the authors jump from a discussion of mean values (without mentioning the 1 sigma errors) to the median with 25 and 75 percentiles. What do I learn from these numbers? The same issue continues for the following paragraphs.*

The box and whiskers plots add complementary information to Table 3 on the variability and the distribution of the observations and the model by showing the median and percentiles. We agree that this information was not used enough. Following this comment and the comment 369, we have revised and restructured the three paragraphs discussing Table 3 and Figure 7 (new Figure 6), avoiding repetitions and making more use of Figure 7 (new Figure 6) statistics in our comments. We also include a short introduction of each statistic to explain what it is and how it is used. The modified text is included in a response above.

*366: "If we consider the higher spatial resolution of our simulations and the smaller domain considered for the statistics compared to TOMCAT, these remaining differences appear consistent with one another.": I don't understand this sentence. Please give some context.*

We agree that the text was not clear. We believe that the differences occur due to differences in model and observation sampling approaches in the two studies. The -0.08 pptv bias in TOMCAT is calculated for the whole flight and ours only using the observations in proximity to the convective system in the upper troposphere (between ~08.20 UTC and ~09.40 UTC). From Figure 11 in Hossaini et al. (2012), it is easy to see that the average bias in this time/altitude range is more negative than for the full flight. We have changed the manuscript.

"TOMCAT's negative bias is slightly smaller than in our case because our sampling focuses on a smaller spatio-temporal domain in proximity to the convective system (between ~08.20 UTC and ~09.40 UTC and in the upper troposphere) while TOMCAT's bias is calculated for the full flight. In the smaller domain, TOMCAT negative biases were larger than the 0.08 pptv average for the full flight."

*369 following: This paragraph and the following (starting at line 378) are very similar to the previous paragraph and discuss the differences in background CHBr3 in the UT. Some arguments are repeated in these paragraphs, some are new. Please restructure these three paragraphs to one without repetitions. Instead, a discussion of UTconv CHBr3 is missing completely here. Also median and 25 and 75 percentile information from Fig. 7 is not used here at all.*

See answers to comment for line 353.

*389: "We selected Mod_Conv3 since it corresponds mostly close in space to Obs_Conv1.": This seems a good choice, but unfortunately, the authors do not compare this simulated CHBr3 to observations.*

To respond to this and to earlier comments about only carrying out a statistical comparison we have now included a comparison in a new map figure (new figure 7) between Mod_Conv_5.4N and Obs_Conv1 showing the CHBr$_3$ observed by the Falcon compared to the model level at 12.5 km in altitude.

*393: "...naturally highest closest to the point of convective detrainment..." Please check the formulation. Maybe "... naturally highest, close to the time of convective detrainment ..."?*

We have changed the formulation.

"The concentrations in the anvil are naturally at their highest at the time and location of convective detrainment and reach up to 0.9 pptv at 09 UTC (17 LT) and begin to decrease after one hour (up to 0.75 pptv) as the anvil is advected north westward by the high-altitude winds."

*Figure 8 and following figures: Please label at least one axis per column and row.*

We have now added the axis labels for longitude and altitude.

*Figure 9: A short notice would be helpful that Fig. 9c is the same as Fig. 8c scaled by the number of bromine atoms (3). It would be also helpful to repeat the black and white lines from the first row of Fig. 9 to all other rows and the following figures to guide the eye in a comparison. In addition, these black and white lines would help to identify regions of convective outflow and tropospheric background. Also a tropopause would be helpful for all these kind of plots.*

We have added a note that Fig. 8c is the same as Fig. 9c except for the bromine atom scaling. We have also added the black and white lines present in Fig. 8 to all other relevant figures. We thank the reviewer for this suggestion which helps the description and the understanding of the figures.

*438: Missing ")" after "(i)"*

Thank you, we have fixed this.

*452: Please define the gases that are summarized as "bromo-carbonyls" and "bromo-methyl peroxides" or use the same names for these groups as in the introduction.*

We have added the chemical formulae in brackets for each class of compounds.

*453: "CHBr3 is insoluble relative to its PGs" -> "CHBr3 is less soluble than its PGs"*

We have modified the text according to the reviewers' advice.

*469: "These compounds contribute 86% of the PG bromine total." I'm confused by the number of 86%. Where does this number come from?*

We had performed some calculations and done some plotting offline and not shown this in the paper. We recognize that the origin of this sentence is not explained. We have now added a short explanation as well as adding a wider range of values to be more precise.

"These compounds contribute a maximum of between 75-95% of the PG bromine total within each of the three the convective columns, which is based on figures (not shown) of the relative contribution of the insoluble bromo-carbonyls to the total PG mixing ratio. We see across all three convective systems that the relative contribution of the insoluble bromo-carbonyls is at a maximum at the point PG mixing ratios are at a minimum. This indicates that these compounds survive complete washout during vertical ascent but since their mixing ratio is small, they play a small role in the vertical transport of bromine within the convection systems."

*Figure 14 d-f: The representation of HOBr as percentage of total inorganic Br mixing ratio is dangerous here, because this plot suggests that HOBr has a large contribution to the convective system by showing relative contributions up to 100 % inside the convective system. From Fig. 13 it is clear that HOBr only has very low background Br pptv above 4 km (and below even a minimum) inside the convective system, but is scaled up in Fig 14 due to the relative minimum in HBr. In my opinion this should be noted in the figure caption to avoid wrong interpretations of Fig. 14.*

We have modified the existing sentence highlighting that these high proportions are only in areas where inorganic bromine levels are depleted to near zero to be clearer.

"Note that the very high HOBr and HBr relative contributions of up to 100 % within the most active part of the convective systems shown in Figs. 14 (d) (e) (f) are not meaningful since the total inorganic bromine mixing ratios are negligible there."

***Section 4.3.3: There are plots for insoluble organic bromine compounds in figures 11 and 12, but these are not even mentioned in this section.***

We agree that we had not drawn attention to the soluble organic PG plots either in Figs. 11 and 12 (g), (h) and (i). We have now added reference to both these figures, and to the ones referring to the insoluble organic PGs in Figs. 11 and 12 (j), (k), and (l).

"The contribution to the total Br mixing ratio from the soluble organic bromine PGs (Figs. 11 and 12 (g), (h) (i)) is also negligible (at a maximum of ~1% in the low troposphere not affected by convection). The bulk of organic PG species are instead in the form of insoluble species (Figs. 11 and 12 (j), (k), and (l)). The enhancements of the organic PGs we see in Figs. 9 (g), (h), and (i) within the convective system are due to the insoluble bromo-carbonyls, i.e., up to 0.08 pptv compared to the background free troposphere 0.02 pptv (Figs. 11 and 12 (j), (k), and (l))."

***470 and following: In this paragraph, the "behaviour of inorganic bromine" is discussed. Earlier in this section inorganic bromine was introduced as a large number of species, but here only HBr and HOBr are mentioned. This selective discussion of only two gases needs to be motivated.***

We did not make it clear enough that HBr and HOBr are the main inorganic bromine species that contribute significantly to the total amount of inorganic bromine in the troposphere. Following reviewer 2's comment, we have made a more detailed analysis and we have found that $BrONO_2$ has also a non-negligible to inorganic bromine even though it is less that HBr and HOBr. We have now changed the text to reflect this.

"We now present a discussion of the most important processes that control the overall speciation of the PGs in the background lower troposphere unaffected by convection in our simulation. This is important because this speciation determines the starting mix of chemical compounds present in the surrounding environment prior to convection and then ultimately what is available for entrainment into the convective system. The speciation of PGs in the lower troposphere is relevant for determining their potential transport to the upper troposphere since they individually have different solubilities.

A key finding is that inorganic bromine dominates the PG budget within the background lower troposphere during this case study. However, during convective transport the inorganic PGs present in the low tropospheric background air are almost entirely removed by washout since HBr is by far its most prevalent component in the marine boundary layer and the next two most abundant inorganic PGs (HOBr and $BrONO_2$) are also highly soluble. The regional tropospheric composition present in our simulations is the underlying cause of this prevalence of HBr and this in turn causes the efficient washout of inorganic PGs."

***479: "A key finding is that inorganic bromine dominates the PG budget within the troposphere, yet despite this the inorganic PGs are almost entirely removed during convective transport by washout due to their solubility." This is a broad statement based on a case study. I think the authors should limit this statement to their case study and not leave it in a general sense.***

This is addressed together with the next comment.

***480: "We here argue that the regional tropospheric composition present in our simulations is the underlying cause of this prevalence of HBr and in turn the washout of inorganic PGs that results from this.": I find this sentence very unclear and don't understand what the authors try to say here. Please rephrase. This makes it also very hard to understand the motivation for the whole Section***

"A key finding is that inorganic bromine dominates the PG budget within the background lower troposphere during this case study. However, during convective transport the inorganic PGs present in the low tropospheric background air are almost entirely removed by washout since HBr is by far its most prevalent component in the marine boundary layer and the next two most abundant inorganic PGs (HOBr and $BrONO_2$) are also highly soluble. The regional tropospheric composition present in our simulations is the underlying cause of this prevalence of HBr and this in turn causes the efficient washout of inorganic PGs."

***4.3.4. Why is so much discussion devoted to the chemical processes?***

We have now added a short introduction to section 4.3.4 to explain the emphasis placed on discussing the chemical processes (see new text added in the answer to the major comments related to section 4.3.4).

***523 following: The comparison to the Marécal et al. (2012) study is very interesting but comes very abrupt here. The authors should consider giving this comparison its own subsection.***

We have now changed this to be its own sub section.

***559: "First, it could be difficult to ..." –> "First, it is difficult to ..."***

We have made this change.

***561: "Furthermore, other tropical regions could have vastly different CHBr₃ emissions, and in the case of much higher emissions, as was explored in Marécal et al., (2012), we could expect a larger role for Br2 formation." Please check the grammar of this sentence. In addition: What exactly is expected to have a larger role for Br2 formation?***

We mean a larger role in terms of vertical bromine transport within convective systems. We recognize the current text is unclear and we are modifying the sentence to read:

"Furthermore, other tropical regions could have vastly different $CHBr_3$ emissions, and in the case where emissions are much higher, as explored in Marécal et al., (2012), we could expect more $Br_2$ formation resulting in an increased role for $Br_2$ in the transport of bromine to the upper troposphere."

***570: "Despite the difference in simulated CHBr3 mixing ratios ...": Differences to what?***

We now make this clearer by stating the difference "between observed and" simulated CHBr3 mixing ratios.

***573: "Indeed, our results show consistent CHBr3 mixing ratios compared to the simulations of Hossaini et al. (2013) for the November 19th flight.": To my understanding this consistency between two models using the same emission scenario only proves that both models work properly in terms of chemistry and transport, but it does not prove that the emission scenario is useful, as it is intended by the authors here.***

We partially agree with the reviewer here. While our results, and those of Hossaini et al., do not provide absolute proof that the Ziska et al. emissions are useful and fully correct, our results cannot be used to refute that statement either. Until further evidence, we would argue that our results provide tentative evidence that these emissions are "useful" for this region. Useful meaning here, of use as a basis for studying bromoform and it chemistry and transport. We have therefore added the following text after the statement identified by reviewer 1: ", and together these findings provide tentative model-based evidence that the Ziska et al. (2013) emissions provide useful estimates of $CHBr_3$ emissions in this region".

***593: "Most of the bromine (>85%) transported to the UT in each convective system is in the form of CHBr3.": This is a very important statement that was not mentioned in the discussions of Section 4.3.***

This statement is based on the discussion that was in section 4.3.1 line 411, and in the new manuscript section 4.3.1 line 482. We have now made it clearer how this statement is reached by discussing more generally the analysis and methodological approach in section 4.3 in some new text in section 4.3 line 445 (see responses to earlier comments). Simply put here, it is possible to trace the enhancement anomalies in CHBr3 vmr in the boundary layer to the convective outflow in the UT. Figure 10 highlights that CHBr3 contributes to >85 of the simulated bromein in the boundary layer, convective column, and convective outflow in the UT.

***595: Missing subscript in CBr2O***

Thank you. We have fixed this issue.

***598: "Overall, we conclude that organic PGs are more important than inorganic PGs for the vertical transport of bromine within the convective columns for the conditions that we study here." This is not true, because it was stated earlier in this paragraph that most of bromine that was convectively transported was in the form of CHBr3. Maybe the authors want to limit their statement to the vertical transport of PGs.***

Within the terminology used by WMO to describe the different brominated gases, bromoform is classified as a source gas while its photo-oxidation products are described as product gases, i.e., PGs. Thus, bromoform is not an organic PG it is an organic "SG". We therefore determine that the original description in the manuscript is correct as it is.

***606: "... more important role of the inorganic PGs for the vertical transport of bromine.": The authors have not shown that the enhancements of HBr in the upper troposphere are due to convective transport.***

We are not sure we fully understand this comment. Our assumption is that there has perhaps been a misunderstanding. We fully agree that we have not shown the UT enhancements in HBr have a link to convective transport. We have actually tried to argue the contrary, which is that HBr is efficiently removed within the convective column. The sentence highlighted by the reviewer was meant to convey that inorganic bromine might be transported more within other convective systems where background ozone levels were higher. We have tried to clarify the text as it also may not have been clear that preceding sentences were referring to higher HBr in the lower troposphere and not the UT. We would also like to highlighted that we believe the higher levels of HBr simulated in the UT are as a result of photochemical ageing of CHBr$_3$ lofted by prior convection.

"The insoluble inorganic PGs, BrO and Br$_2$, are only present at negligible mixing ratios and play no significant role in the vertical transport of bromine. Our interpretation is that the lower tropospheric inorganic PG budget is shifted heavily in favour of HBr formation due to the low background O$_3$ mixing ratios simulated in this region. This limits the availability of lower tropospheric HOBr leading to only very limited formation of Br$_2$ within the cloud and rain droplets within the lower regions of the convective system resulting from the reaction between HBr and HOBr. More BrO and HOBr would form in cases with higher background O$_3$, which could potentially lead to enhanced Br$_2$ formation within other convective systems and a more important role of the inorganic PGs for the vertical transport of bromine."

***608: "Overall, these conclusions are valid in all parts of the convective system except for where the anvil detrains into residual convective outflow in the UT.": Such a statement is not covered by the findings discussed in the main part of this paper.***

We have removed the entire paragraph discussing this point. The reviewer is correct that this issue is not discussed prior to this point and so it makes sense to remove it from our conclusions.

***612: In my opinion, section 7 is very short and could be attached to section 6.***

We have now merged section 7 with section 6 and made it clear the final paragraph presents a short outlook.

We thank Reviewer #2 for his/her comments leading to a significant improvement of the manuscript.

This response is organised with the reviewer's comments in black bold font, the authors' answers in black normal font and the text added in the revised manuscript in red font.

Note that following reviewer #1's comment:

- figure 2 has been removed but the flight trajectory is now included in figure 1
- a new figure 7 have been included showing a direct comparison between the simulation and the measurements on a map.
- the naming of the modelled systems has been changed to include their latitude (Mod_Conv1, Mod_Conv2 and Mod_Conv3 replaced by Mod_Conv_4.35N, Mod_Conv_3.75N, and Mod_Conv_5.4N, respectively)
- figure 5 showing the model cloud top has been replaced by an estimate of the brightness temperature from the cloud top pressure (new figure 4) to provide a direct comparison to the observed brightness temperatures (figure 1)
- because section 7 was very short, we have now merged section 7 with section 6 and made it clear the final paragraph presents a short outlook

To improve the clarity of the manuscript, the discussion section (4.3.4) is now split into two sections with the comparison to Marécal et al. (2012) being now in a separate section (new section 4.3.5).

**The paper describes a cloud-scale modeling study to investigate convective transport of bromoform (CHBr3) and its product gases. The model is applied to a case study along the west coast of Borneo that was sampled with aircraft measurements, which provide a means to evaluate the model. The major findings are that there is good agreement of CHBr3 mixing ratios between model and observations in the boundary layer and reasonable agreement in convective outflow regions. Analysis of the bromine speciation in convective outflow shows ~85% of the Br is from CHBr3, < 10% from inorganic product gases (HBr, BrO, HOBr, Br, Br2, BrONO2), ~2% from organic product gases (brominated peroxides and carbonyls). The paper finds that the inorganic product gases are dominated by HBr, which is highly soluble and quickly removed by the convection. The paper suggests that the high HBr is a result of the low O3 environment (O3 < 20 ppbv) that favors production of HBr from Br + HCHO and Br + HO2 over production of BrO from Br + O3. Without BrO production, HOBr also has low mixing ratios. Further, aqueous phase reaction between HOBr and HBr is fast, further limiting HOBr within convection.**

**This study provides fundamental knowledge on the processing of Br compounds in tropical convection, which is important to apply to global models that examine pathways of Br compounds to the stratosphere where Br plays a critical role in ozone chemistry. The paper covers several topics without fully justifying why each topic is addressed. My main concern is that the main points are not concisely given (e.g. in the abstract or results section) but are written along with other points that cause a loss of clarity in the story. For example, why do we need to know the convective transport efficiency? Secondly, why is it important to know if CHBr3, HBr, BrO, or organic Br is the bromine compound transported to the upper troposphere?**

The paper has been revised to make the analysis of the results more organized and clearer (see answers to specific comments below). We have made clear that the comparison of the transport efficiency is a mean to evaluate the simulation compared to observations. We have also largely improved the discussion on the bromine partitioning in particular making clearer its aims.

**In addition, the discussion focuses on the regional chemistry that the convection forms in. The discussion makes sense but is not supported with figures showing rates of reactions.**

As suggested, we have now given details on the reaction rates to support our analysis (see answers to specific comments).

**Specific Science Comments**

**1. The abstract needs to be improved by making the existing text be more concise, stating more clearly that low O3 levels lead to HBr gas-phase production, and adding comments on the role of aqueous chemistry.**

We have now shortened the abstract to make it more concise by removing unnecessary details repeated later in the paper. We have also highlighted the link between low ozone and HBr in a clearer way.

**2. Line 102 in Introduction. I do not think it was explained what the limitations are of the previous studies. There should be a few sentences clearly stating these limitations in the Introduction and again in the Discussion.**

The previous studies were only idealized cases that used induced convection in the simulations by introducing artificial atmospheric perturbations and used various simplifying assumptions. We have now highlighted this as their main weakness and limitation in the introduction:

"The previous studies of VSLS chemistry and transport at the convective scale were only idealised cases that used a set of simplifying assumptions (e.g., no emissions, constant vertical profiles for initial conditions, and no synoptic scale meteorological forcing) and artificial perturbations to the modelled atmosphere to induce their simulated convection. Thus, these cases were not realistic and it would not have been relevant to compare them to observations. We wish to expand upon that previous work, e.g., Marécal et al. (2012), by carrying out a real-world case study."

We then repeat some of this information in the discussion in more detail within the new section 4.3.5. This also covers responses to comments made by reviewer 1.

**3. Line 153 states that the CHBr3 measurements were performed by the GHOST instrument. Please add information on what the technique for these measurements is, as well as its detection limits and uncertainties.**

We have now added text in section 4.2 explaining that "The GHOST instrument is a gas-chromatograph mass spectrometer and had an error of ±17.7% that was primarily driven by uncertainties in the gas standard."

**4. Since the GHOST instrument is a GC/MS technique, are there other trace gases that it measured that are useful for this analysis, such as other Br-containing trace gases?**

The GHOST instrument does measure other bromocarbons, but these compounds are not included in our chemical mechanism and so these additional observations cannot help our analysis. Beyond the GHOST instrument, although it was initially planned in SHIVA to measure other bromine species, logistical problems with aircraft availability created a tight limitation on the payload for the mission. As a result, it was not possible to take chemical ionization mass spectrometry instrumentation onboard, which would have been able to measure some of the inorganic components.

**5. Line 223. I noticed that the chemistry listed in the supplement does not include Br-Cl reactions, e.g. Cl + CHBr3 → HCl + CBr3. What impact would Cl chemistry have on the results provided in this paper?**

A sensitivity analysis on the impact of Cl chemistry was done in Marécal et al. (2012). They showed that chlorine chemistry has only a negligible impact on the production of $Br_x$. This is why we have not

included Br-Cl reactions in the chemistry scheme used in the paper. This information is added in the revised manuscript.

*"Note that bromine-chlorine reactions have not been included in the chemistry scheme since Marécal et al. (2012) showed that it has a small impact on the production of $Br_x$."*

**6. Does the model include direct uptake of Br compounds onto ice? It may not be important for the species considered (except BrONO2; Fernandez et al., 2014, ACP), but I suggest mentioning this process.**

There is no direct uptake of Br compounds onto ice. Br is transferred to ice through liquid-to-ice processes (like riming) using retention coefficient as explained in section 3.1. These processes are dominant in the formation of ice hydrometeors in deep convective systems.

*"This approach represents liquid-to-ice processes like riming that are the dominant process for the formation of ice hydrometeors in convective clouds."*

Furthermore, we find that HBr is almost entirely removed at an altitude in the convection column where hydrometeors exist only in solid form. Therefore, we do not expect an important effect of direct uptake onto ice as also discussed in Marécal et al. (2012) Sect. 3.2.1.

*"The uptake of bromine species onto ice hydrometeors is not represented as it was found in Marecal et al. (2012) to not have an important effect on bromine removal."*

**7. Lines 245-249. Could you provide a little more information on the emissions? Do the emissions vary temporally? If not, what time-span is the average for?**

We have used a diurnal variability on the emissions linked to solar zenith angle such that emissions peak at solar noon. The mean emission in pmol $m^2$ $h^{-1}$ is equal to that shown in Fig. 3. We have added this information in the text.

*"A diurnal variability linked to solar zenith angle is applied to these emissions such that they peak at solar noon."*

**8. Line 298. What is the vertical resolution of the model in the upper troposphere? Does the vertical resolution affect the cloud top height estimate? Since the results show a fairly good comparison, I guess not, but I suggest thinking about the uncertainties in the model cloud top height if dz > 500 m.**

At the altitudes near the cloud top height the model vertical resolution is ~300 m. This does mean that the precision of the model's cloud top height estimates were equivalent to this resolution. Note that, on the advice of reviewer #1 we now use a model estimated cloud brightness temperature as a measure of cloud top height. This was preferred because we show satellite observations of cloud top brightness temperature. This does not change the results that still show a good agreement between the model and the observations.

**9. Line 328. Are there any temporal uncertainties to consider? I imagine the statistical approach removes effects of poor timing of the convection, but does it also reduce uncertainties in relation to timing of emissions (if there is a diurnal profile) or photochemistry?**

We think the main effect is in removing effects of different timing of the convection. It is harder to say anything concrete about the emissions uncertainties since little is known about the short term variability of the emissions. Temporal emission errors would mainly effect the accumulation of bromoform in the boundary layer prior to convective entrainment and the amount of bromoform

lofted in the convective column. The statistical approach has more an effect of temporally smoothing the detrainment and so this choice does not remove emission error.

**10. Line 330. I first saw the equation given on line 333 in Cohan et al. (1999) JGR (please cite them). This equation assumes that there is no mixing of air at different altitudes between cloud base and cloud top, i.e. it is a two-component mixture model. During the past decades, others have used modified versions of this type of equation. For example, Borbon et al. (2012) JGR applied a three-component mixture model, while Yang et al. (2015) JGR applied a four-component mixture model, and Fried et al. (2016) JGR applied a 10-layer mixture to account for entrainment of air between cloud base and cloud top. The multi-layer approach is useful for conditions when the vertical profiles vary with altitude in the clear air. It would be good to see the clear-sky vertical profile for CHBr3 to show its variability with altitude. Is CHBr3 sufficiently close to zero or non-varying with altitude such that entrainment of air between cloud base and cloud top can be ignored?**

Reference to Cohan et al. 1999 added. The reason we have used the approach based on a two-component mixture model is that we wanted to evaluate the simulation by making a comparison with Krysztofiak et al. (2018) study that chose this approach. We agree that a multi-layer approach is better for vertical profiles varying with altitude. Looking at the background CHBr3, we find that there is very little variability from 2 km to the upper troposphere ($CHBr_3$ 0-0.5 pptv). Between 400m and 2km altitude, $CHBr_3$ concentrations are slightly affected by the lowest levels but are still below 1 pptv.

Also following reviewer #1's comments we have revised the introduction of section 4.2 and added more explanation of $f$ (air fraction transported from the BL to the UT).

"Before discussing the results of the simulated chemistry in detail (section 4.3), we evaluate if the simulation gives reasonable results for $CHBr_3$ concentrations and for convective transport efficiency compared to the aircraft observations.

We firstly use statistical characteristics for this comparison. We choose this approach for two reasons. First because of differences in location and timing between the observed and simulated convection events, and, second, because of spatial uncertainties in the emission inventory used in the simulation. This approach allows a clearer comparison of the observations and simulation by removing effects arising from inherent time and spatial uncertainties.

In order to compare the convective transport efficiency between the observed and simulated systems we follow the approach proposed by Cohan et al. (1999) and used by Bertram et al. (2007). To estimate the air fraction, $f$, originating from the boundary layer (BL) and transported by convection we use the relationship from Cohan et al. (1999):

$$[X]_{UTconv} = f \cdot [X]_{BL} + (1 - f) \cdot [X]_{UTnoconv}$$

where the mean mixing ratios in the boundary layer, the upper troposphere within the convective systems, and the upper troposphere in the vicinity but outside the convective systems are represented by $[X]_{BL}$, $[X]_{UTconv}$, and $[X]_{UTnoconv}$, respectively. $f$ ranges from 0 to 1 with large values corresponding to an efficient convective transport of air masses from the boundary layer to the upper troposphere. This formulation of $f$ is chosen because it was recently applied to the SHIVA aircraft data (Krysztofiak et al., 2018). It relies on the assumption of a low variability of background concentrations with altitude, which is fulfilled for $CHBr_3$ in our case study (not shown). Previous studies based on observations and reported in Krysztofiak et al (2018) provides estimates of $f$ in the range 0.17 to 0.36."

**11. Lines 388-395. Could you specify where the vertical cross section is located. Is it an average across the anvil, or a line down the center of the anvil?**

The cross section is taken as close to the centreline of the convective cell as possible. We have now explained this in the text.

"The cross sections in each case run as close to the centreline of each convective system as possible."

**12. Section 4.3.2. Figures 9 and 10 show values based on Br pptv and percent contribution to total Br. Somehow the text gets a bit confusing. Is it possible to organize the text a little differently, such as by region, which would better align with the Conclusions section. For example, In the UT convective outflow CHBr3 contributes 85%, inorganic PGS < 10%, and organic PGs ~2% to total Br. In the boundary layer, CHBr3 contributes ….. In the 1-4 km layer, CHBr3 contributes …..**

This section has been organised following this suggestion making it clearer

"Figs. 9 (d), (e), and (f) show that there are relatively low levels of inorganic bromine (Br, $Br_2$, BrO, HOBr, HBr, $BRONO_2$) concentrations in the boundary layer even in the areas not directly affected by convective precipitation with values typically in the range of 0 to 0.4 pptv Br, i.e., <5 % contribution to the total Br mixing ratio (Figs. 10 (d), (e), and (f)). The highest simulated inorganic bromine mixing ratios (0.3-0.4 pptv Br) in the boundary layer occur to the west of the Mod_Conv_5.4N system still only contributes <10 % to the total boundary layer pptv Br (see Fig. 10 (f)). This spatial variability in the boundary layer inorganic bromine mixing ratios around each convective system arises due to differences in precipitation location and timing over the course of the simulation prior to November 19[th], 2011. Precipitation events occurring in the two preceding days deplete the boundary layer of inorganic bromine due to washout (analysis not shown). In the boundary layer, organic PGs (CHBrO, $CBr_2O$, $CHBr_2OOH$, and $CBr_3COOH$) concentrations are up to 0.2 pptv Br (up to 10% contribution to total bromine but very locally) and are formed due to $CHBr_3$ photochemical loss (Figs. 9 (g), (h) and (i) and 10 (g), (h) and (i)).

Air masses in the convective column itself and convective outflow are almost entirely depleted of inorganic bromine with mixing ratios of <0.1 pptv Br and with contributions to the total Br mixing ratio well below 5%. There, organic compounds are being driven from the low levels up to the upper troposphere in the main ascent and the outflow and show enhanced mixing ratios within the convective column compared to the free troposphere (Figs. 9 (g), (h), (i)). Still organic PGs have a contribution to the total bromine only up to ~4%.

In the free troposphere, inorganic and organic bromine concentrations are enhanced between 1 and 4 km to the west of each convective system (Figs. 9 (d), (e), (f), (g), (h), (i) and 10 (d), (e), (f), (g), (h), (i)). There, total inorganic (resp. organic) PGs peak up to 1 pptv Br (resp. 0.2 pptv Br), which constitutes a portion up to 45% (resp. 5%) of the total Br mixing ratio. Among the three convective systems, Mod_Conv_5.4N exhibits the highest concentrations of the organic PGs.

Above 4 km altitude in convection-free areas, inorganic bromine is mainly in the 0.2-0.4 pptv Br range (15-35% contribution to total Br) that is higher than within convection (Figs. 9 (d), (e), (f) and 10 (d), (e), (f)). There, organic PGs have low concentrations (0.02-0.03 pptv Br) and contributes only to 1-3% (Figs. 9 (g), (h), (i) and 10 (g), (h), (i))."

**13. Line 488. It is suggested that reactions 2 and 3 could be important routes for HBr formation. Reaction 2 is Br + HCHO. Does HCHO come from CH4 or are there VOCs, such as isoprene, contributing to its mixing ratio? What is the typical reactivity for reactions 1 to 3 for the study area? It would help to see reaction rate constants and typical O3, HCHO, and HO2 mixing ratios.**

There are other VOCs present beyond methane, and biogenic isoprene emissions contribute towards the HCHO mixing ratio. We agree that it would be useful to list the reactivity of reactions 1-3. We have added table 4 now to show the typical $HO_2$, $O_3$, and HCHO mixing ratios along with the rate constants for these reactions and the resulting reaction rates. We include a new paragraph discussing this table and the implications of these results. Having examined the rates of the various equations in more detail we have now added a discussion of a reaction that was not previously mentioned (BrO + NO),

which plays a role in helping to suppress BrO levels. We considered plotting the cross-section structure of the reaction rates for each compound, but this rapidly becomes very complicated to analyse and requires extensive explanation. Table 4 and the accompanying explanation offer a compromise that conveys the essential information in a condensed form. Additionally, we used mixing ratios in the lowest part of the troposphere (200 m) to construct Table 4 because it is the mixing ratios in these air masses that become entrained within the convective systems.

**14. Line 505. Again, it would help to know the reaction rate constant and reactivity of reaction 7. Marecal et al. (2012) report 1.6x1010 M-1 s-1, which is indeed high but is the rate high as well when compared to competing gas-phase reactions? It would be good to see figures of reaction rates supporting the discussion on what reactions dominate.**

Following this query we have compiled all of the relevant reaction rate constants and reaction rates discussed within section 4.3.4 into a new table (Table 4). The data in Table 4 are selected for a point in the boundary layer at 200 m altitude in an area to the north of Mod_conv5.4N. This location is part of an air mass that becomes entrained in Mod_Conv5.4N and is representative of the typical air masses entrained in the three systems. Note that we have added three more reactions into the discussion to give a more complete explanation (BrO + NO, and the photolysis of both HOBr and BrONO$_2$), and as a result R7 is now numbered as R10. As the reviewer points out, the reaction rate constant we have used for R10 (HOBr+HBr) is calculated in the same way as in Marecal et al. (2012). Indeed, R10 has a very fast rate constant, which has implications for HOBr lifetime within cloud and rain droplets (see response to next query). However, despite the fast rate constant, the overall rate of R10 in the example given in Table 4 is low and we have determined this is because R10 is strongly limited by the availability of HOBr. R10 only becomes important in a few areas where background levels HOBr is elevated as a result of higher background levels of BrO. The new discussion highlights importance of R2 (Br + HCHO) for leading to HBr formation, R4 (BrO + HO2) for removing BrO, and R9 (HOBr + hv) for recycling HOBr back to Br and for keeping HOBr mixing ratios very low.

Since we substantially rewrote section 4.3.4 amounting to nearly 3.5 pages of text we have not copied the full text here and quote the most relevant passages along with Table 4.

[revised manuscript text omitted]

**15. Line 521. It mentions the very short residence time due to falling hydrometeors. Can you calculate the residence time? I would suggest looking at Bela et al. (2018) JGR who discuss time an air parcel spends in contact with liquid water in convection (ranging from weak convection to severe convection). They also showed via vertical profiles that highly soluble trace gases (like H2O2) are depleted rapidly below the freezing level, whereas less soluble gases will be lofted higher.**

Following this query (and the one above) from the reviewer we spent some time trying to understand the residence time versus the reaction rate for the HOBr + HBr reaction (now R10). As a result, we realized that the original explanation in the paper, i.e., that the residence time was shorter than the lifetime of HOBr or HBr in aqueous solution, was incorrect. While unfortunately we cannot calculate the residence time, we were able to determine that the lifetime of HOBr in the presence of 0.1 g kg-1 cloud and 0.5 g kg-1 (corresponding to the active convective center in the BL) and 0.2 pptv HBr (dissolved) was very short at $2 \times 10^{-2}$ s such we can be confident it is much shorter than the residence times of rain or cloud droplets. We have added the following text though to draw a contrast with Bela et al. (2018):

"The residence time of cloud and rain droplets in the atmosphere can impact aqueous phase chemistry in cloud and rain droplets. For other aqueous chemistry systems occurring in cloud and rain droplets the rate of chemical reactions could occur slower than it takes for cloud or rain droplets to fall leading to wet scavenging of the chemical species involved. Bela et al. (2018) report an example of aqueous phase chemistry in convective cloud and rain where the wet scavenging removal of H2O2 was found to be much faster than its production via aqueous phase chemistry in cloud and rain. By contrast, the rate constant for R10 is approximately $10^7$ times faster than the rate constants involved in the formation of $H_2O_2$ described by Bela et al. (2018), the lifetime of HOBr in solution is approximately $2 \times 10^{-2}$ s as a result. Thus, the residence time of cloud and rain droplets is not a limiting factor for R10."

**16. Lines 524-557 discuss how the current results compare to those by Marecal et al. (2012). Could the authors make clear what are the differences between the model configuration by Marecal et al. and this paper? I found two comments on the differences, 1) running real meteorology (and thus global model generated initial conditions) and running ideal meteorology, and 2) running a case for Darwin versus Borneo where emissions, photolysis rates, OH mixing ratios can be different. Are there other differences? What about the information in the "new chemistry" section?**

We have now added the following text to explain the differences with the model configuration in Marecal et al. (2012).

"The C-CATT-BRAMS model configuration in Marecal et al. (2012) differs from the setup used in this study in the following ways:

- The model domain was set to be at the same latitude and longitude as Darwin (Australia).
- There were no emissions of $CHBr_3$ or any other VSLS. The only source of $CHBr_3$ was from a 2 km homogenous layer above the surface set as an initial mixing ratio of either 1.6 pptv or 40 pptv in the two scenarios.
- Marecal et al. (2012) used a chemical mechanism that did not include a representation of non-methane hydrocarbon chemistry. This chemistry was added in this study by including the ReLACS chemical mechanism.
- Apart from CHBr3 set in the lowermost 2 km, the model initial and boundary conditions for the chemical species were defined from a single vertical profile from the MOCAGE CTM from over Darwin. The mixing ratios of all PGs were initialized at 0 pptv in all model grids and layers.
- The meteorology in Marécal et al. (2012) involved an initial setup that applied a single vertical profile of meteorological conditions throughout the entire horizontal domain. The vertical profile was defined from a radiosonde profile obtained above Darwin corresponding to November just prior to the main wet season. Convection was artificially forced in the simulation by introducing a perturbation in the lower model layers of increased temperature and humidity."

**17. Line 542. For the discussion of differences due to Darwin and Borneo settings, the lifetime of CHBr3 is discussed. These lifetimes are > 15 days. How do the changes in chemical lifetime affect the results of convective transport, which occurs in < 1 hour and is the topic of the paper?**

The C-CATT-BRAMS simulation is run for almost 48 hours prior to the onset of convective activity while the TOMCAT simulation used for background and initial conditions was run for 3 years prior to the 19th November 2011. The lifetime of $CHBr_3$ during this preceding period affects the abundance of it and its PGs as well defining the relative composition of the different PG components. So while the typical transport time within the convective systems is less than one hour, the lifetimes with respect to photolysis and reaction with OH, both do play an important role in defining the tropospheric composition of $CHBr_3$ and its PGs prior to convection. We now make this point a bit clearer by adding the following text:

"The $CHBr_3$ lifetimes with respect to photolysis and OH are important for defining the relative partitioning of bromine between $CHBr_3$ and its different PGs prior to the convective activity on the 19th November."

**18. Line 557. Isn't O3 low in both Darwin and Borneo? I do not see why low O3 would cause differences between the results from Marecal et al. and this paper.**

The intention of this sentence was to draw a parallel with the ozone levels along the coast of Borneo and in the simulation near Darwin in Marécal et al., 2012. However, due to the placement of the sentence and the overall context of the paragraph, we realise this is not very clear. We have therefore modified this entire section of text to make it clearer.

"Marécal et al. (2012) showed that significant amounts of $Br_2$ were only released into the gas phase from R7 in cloud droplets when their idealized simulation was initialized with very high $CHBr_3$ (40 pptv) in the BL. In the more realistic case where boundary layer $CHBr_3$ was ~1.6 pptv, the formation and release of $Br_2$ via R7 was very limited, which is consistent with our findings here as shown in Fig. 15. Following the causal link we identify between low ozone and resulting low HBr and HOBr, these similar results for $Br_2$ could be explained by the similarly low background $O_3$ simulated over the two regions (Borneo and Darwin) in both studies. In Marécal et al. (2012) the background $O_3$ was 14 ppbv at 2 km as compared to the 10-15 ppbv range simulated over the inner domain at 2 km altitude in this study."

**19. Section 4. How do the results from this paper fit in the context of findings from the CONTRAST field campaign? I am not sure it is possible to have a thorough discussion, but noticed that Chen et al. (2016) JGR discuss convective transport (or not) of BrO and HOBr + Br2.**

We thank the reviewer for highlighting this relevant paper. This study offers some qualitative support for the hypothesis linking both the low simulated BrO and HOBr levels and also the limited production of $Br_2$ to the low background tropospheric ozone levels. Indeed, they only observed elevated BrO and HOBr+$Br_2$ in biomass plumes where ozone was significantly elevated and for the remainder of the time the observed levels of BrO and HOBr+$Br_2$ in the troposphere were below the limit of detection. We have therefore added some text in Sect 4.3.4 to discuss this.

"Indeed, there may be some observational support for this in data collected during the CONTRAST field campaign. Chen et al. (2016) showed that the observed levels of BrO and the sum of HOBr+$Br_2$ were both below the limit of detection (0.6-1.3 pptv and 1.5-3.5 pptv, respectively) in background tropical tropospheric air where ozone levels were relatively low (< 50 ppbv). They found that BrO and HOBr+Br2 were only above the limit of detection in biomass burning plumes where ozone levels were significantly elevated (> 50 ppbv). Their findings are at least consistent with our expectations of a link between ozone and the bromine speciation between HBr, BrO, HOBr, and $Br_2$."

**20. Section 5. Can results from global model studies (e.g. from TOMCAT, or other CCMI models) help identify other tropical regions that should be studied?**

The work of Hossaini et al. (2016) highlights the Indian sub-continent and South-East Asia during the summer monsoon as two regions for importance for vertical transport of bromoform linked to deep convection. Their work shows overall lower levels of bromoform being transported to the upper troposphere, but the background tropospheric composition is likely very different due to the proximity to large sources of anthropogenic pollutants and ozone precursors. We have now highlighted this region in section 5 with:

"Hossaini et al. (2016) highlight the Indian sub-continent and southeast Asia as another region that is potentially of importance for transport of bromoform and its PGs to the upper troposphere within deep convection. They predict it to make a more minor contribution to the vertical transport of bromoform compared to the maritime continent, but the background tropospheric conditions are very different from the present study due to its proximity to large pollutant and ozone precursor sources, and its contrasting conditions could make it an interesting case study."

**Line 60, I suggest starting the sentence as, "Bromoform (CHBr3), with 3 Br atoms …."**

We have modified this sentence as suggested.

**Line 505, Could you make the negative ion sign more obvious. Perhaps via a symbol or equation editor.**

Thank you. We have tried to fix this by increasing the font size.

**Line 589, Suggest: Fuhlbrugge et al. (2016) who showed that ….**

Thank you. We have changed the text to correct this.

**4. I recommend that the Appendix be placed in the supplement.**

We have moved the appendix to Supplement S2.

**In Table 1, is the Hx for Br2 correct? A positive value is given for aH yet Hx is less than H298.**

Correct. This was an error. We have modified Hx to the correct value for Br2 in this case, which is 0.97.

**Is the vertical cross section in Figure 8 in a different location than the other vertical cross sections?**

This is the same location as shown in Figure 9c etc, which corresponds to Mod_Conv 3 located at 5.4°N. Figure 8c corresponds to the same time (9 UTC) as in Figure 9c etc.